# Les Houches lectures on non-perturbative Seiberg-Witten geometry

Loïc Bramley, Lotte Hollands, Subrabalan Murugesan

November 2024

## Abstract

In these lectures we detail the interplay between the low-energy dynamics of quantum field theories with four supercharges and the exact WKB analysis. This exposition may be the first comprehensive account of this connection, containing various novel arguments and illustrative examples.

The lectures start with the introduction of massive two-dimensional $\mathcal{N} = (2,2)$ theories and their spectra of BPS solitons. We place these theories in a two-dimensional cigar background with supersymmetric boundary conditions labelled by a phase $\zeta = e^{i\vartheta}$, while turning on the two-dimensional $\Omega$-background with parameter $\epsilon$. We show that the resulting partition function $\mathcal{Z}_{2d}^{\vartheta}(\epsilon)$ can be characterized as the Borel-summed solution, in the direction $\vartheta$, to an associated Schrödinger equation. The partition function $\mathcal{Z}_{2d}^{\vartheta}(\epsilon)$ is locally constant in the phase $\vartheta$ and jumps across phases $\vartheta_{\mathrm{BPS}}$ associated with the BPS solitons. Since these jumps are non-perturbative in the parameter $\epsilon$, we refer to $Z_{2d}^{\vartheta}(\epsilon)$ as the *non-perturbative partition function* for the original two-dimensional $\mathcal{N} = (2,2)$ theory. We completely determine this partition function $\mathcal{Z}_{2d}^{\vartheta}(\epsilon)$ in two classes of examples, Landau-Ginzburg models and gauged linear sigma models, and show that $\mathcal{Z}_{2d}^{\vartheta}(\epsilon)$ encodes the well-known vortex partition function at a special phase $\vartheta_{\mathrm{FN}}$ associated with the presence of self-solitons. This analysis generalizes to four-dimensional $\mathcal{N} = 2$ theories in the $\frac{1}{2}\Omega$-background.

EMPG

# 1 Introduction and summary

These lecture notes have evolved from a series of lectures given by the middle author at the Les Houches school on Quantum Geometry in August 2024. In these notes we have tried to preserve the expository character of the lectures, but also included various detailed arguments and new examples. Just as in the lectures themselves, these notes contain many cross-references to other lectures of the school, and the interested reader can consult the Les Houches webpage for further information.

In particular, at the time of these lectures students had already learned about various aspects of BPS states in four-dimensional $\mathcal{N} = 2$ theories (in particular those of class S) from Andrew Neitzke, based on the impressive papers [1, 2]. Furthermore, Nikita Nekrasov had introduced instanton counting in (the complementary class of) four-dimensional quiver $\mathcal{N} = 2$ theories, based on the influential papers [3, 4]. Moreover, Marcos Mariño simultaneously gave inspiring lectures on non-perturbative aspects of topological string theory, available as [5], and Kohei Iwaki delivered beautiful lectures on the exact WKB analysis and Painleve equations. All of these are very relevant to these lectures.

The aim of these notes is to combine our understanding of BPS states in theories with four supercharges, together with our ability to compute supersymmetric partition functions, to construct a new **non-perturbative partition function**. This partition function was introduced in the 4d $\mathcal{N} = 2$ setting in [6], evolving from [7, 8, 9, 10] and inspired by the Gaiotto-Moore-Neitzke papers [1, 2]. It is the 4d analogue of the non-perturbative topological string partition function introduced around the same time in [11]. The approach we take is closely related to various other perspectives and results in the literature, such as the topics of non-perturbative topological string theory [12, 13], isomonodromic tau functions [14], BPS/CFT correspondence [15], analytic/geometric Langlands [16], Riemann-Hilbert problems [17], holomorphic Floer theory [18], etc.[1]

The adjective "non-perturbative" might be confusing, as you may argue that the instanton partition function is already analytic, and thus non-perturbative, in the parameters of the $\Omega$-background. Yet, we want to argue that it is natural to introduce a new partition function, and associated Seiberg-Witten geometry, that depends in a locally constant way on an additional phase, in such a way that it naturally reproduces the instanton partition function when this phase coincides with the phase of $W$-bosons in the underlying 4d $\mathcal{N} = 2$ theory. We call the new partition function non-perturbative since its jumps have a non-perturbative dependence on the parameters of the $\Omega$-background, and encode the spectrum of 4d BPS states. Moreover, it turns out that the new partition function, at a phase opposite to that of the W-bosons, is closely related to the so-called non-perturbative topological string partition function.

---

[1]We apologize in advance for the small collection of papers that we cite in these lecture notes. It is simply impossible to do justice to all the exciting papers on these topics. Rather, we make the choice to give the reader a gateway to the many interesting papers out there.

In these lecture notes we follow the structure of the four lectures in the school, albeit spreading them out over two papers. We start the notes off in two dimensions, where the relation between supersymmetric field theory and the exact WKB analysis is cleanest. The class of two-dimensional Landau-Ginzburg models that we choose as a recurring example, is moreover closely related to minimal models, integrable hierarchies of KdV type, as well as matrix models, making a connection to earlier lectures in the school.

- In §2 we begin with gently reviewing 2d massive $\mathcal{N} = (2,2)$ theories and their vacuum structure. We find that the 2d $\mathcal{N} = (2,2)$ vacuum structure is mathematically encoded in a spectral curve $\Sigma$ together with a differential $\lambda$.

- In §3 we introduce BPS solitons as field configurations which tunnel between two vacua. We show how they are encoded as trajectories in spectral networks $\mathcal{W}^\vartheta$, and as special Lagrangian discs in the spectral geometry, and we derive celebrated wall-crossing formulae.

- In §4 we introduce BPS vortices as field configurations that have a non-trivial winding at infinity. We turn on the $\Omega$-background with parameter $\epsilon$ and compute the corresponding vortex and (closely related) Higgs branch partition function. We find that the Higgs branch partition function is annihilated by a differential operator $d_\epsilon$ that quantizes the spectral geometry. We further define the dual Coulomb branch partition function.

- In §5 we make contact with the exact WKB analysis. We define the non-perturbative Higgs branch partition function $\mathcal{Z}^\vartheta(\mathbf{z}, \epsilon)$ as the partition function on the $\Omega$-deformed cigar relative to boundary conditions labelled by the phase $\zeta = e^{i\vartheta}$. We show that this non-perturbative partition function may be computed by the exact WKB analysis with respect to the differential operator $d_\epsilon$, and that the vortex partition function is encoded in the non-perturbative partition function at a distinguished phase $\vartheta$, corresponding to the presence of self-solitons. We also analyse the dual non-perturbative Coulomb branch partition function.

This outline is depicted in Figures 1 and 2. Although most material in the first sections is well-known, and reviewed in other places, we believe that this is the first paper to give a complete account of the relationship of 2d $\mathcal{N} = (2,2)$ theories and the exact WKB analysis. We note that similar a partition function in the cigar background, with boundary conditions labeled by phase $\zeta = e^{i\vartheta}$, was studied in the context of the tt* geometry in [19] and [20]. Also, the four-dimensional analogue of $\mathcal{Z}^\vartheta(\mathbf{z}, \epsilon)$ was studied in [9, 10, 6]. Yet, the two-dimensional story was not yet spelled out in as much detail as we do in §5. Specifically, the statement that the vortex partition function is encoded in the non-perturbative (Higgs branch) partition function at a distinguished phase is new (albeit similar to an analogous such statement in four dimensions [8, 9]). Finally, other original contributions are

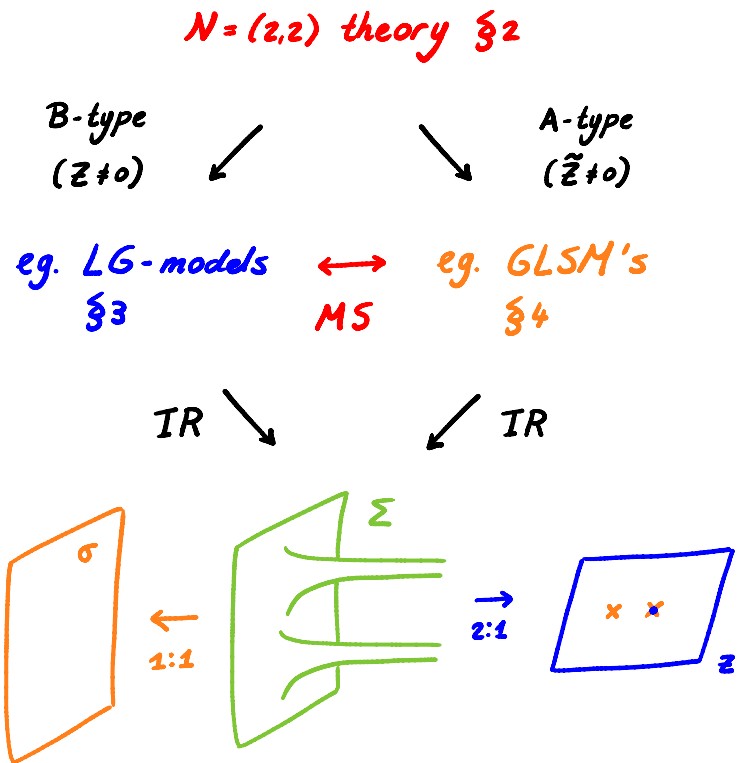

Figure 1: Schematic summary of sections 2, 3 and 4.

the characterisation of the non-perturbative Higgs branch partition function for the $\mathbb{P}^1$-model in §5.8 (although the main statement is part of [10]), as well as its dual interpretation as a non-pertubative Coulomb branch partition function. The duality between non-perturbative Higgs and Coulomb branch partition functions will be described in more detail in [21].

This analysis can be repeated for four-dimensional $\mathcal{N} = 2$ theories, where it is possible to make a connection with non-perturbative string theory, and in particular the TS/ST correspondence (reviewed in [5] and [13], respectively). This will be part of [22].

## Acknowledgements

We would like to thank the Les Houches School of Physics for an amazing learning environment and LH would like to think the organizers of the summer school on Quantum Geometry for the invitation to give this series of lectures. LH also thanks Andrew Neitzke for a long collaboration that has paved the way for many of the results presented here, Tudor Dimofte for countless enlightening discussions, Ahsan Khan and Tudor Dimofte for detailed comments on previous versions of these notes, and many others, including Murad Alim, Tom Bridgeland, Alba Grassi, Qianyu Hao, Marcos Mariño, Greg Moore, Takuya Okuda, Sasha Shapiro and Joerg Teschner, for

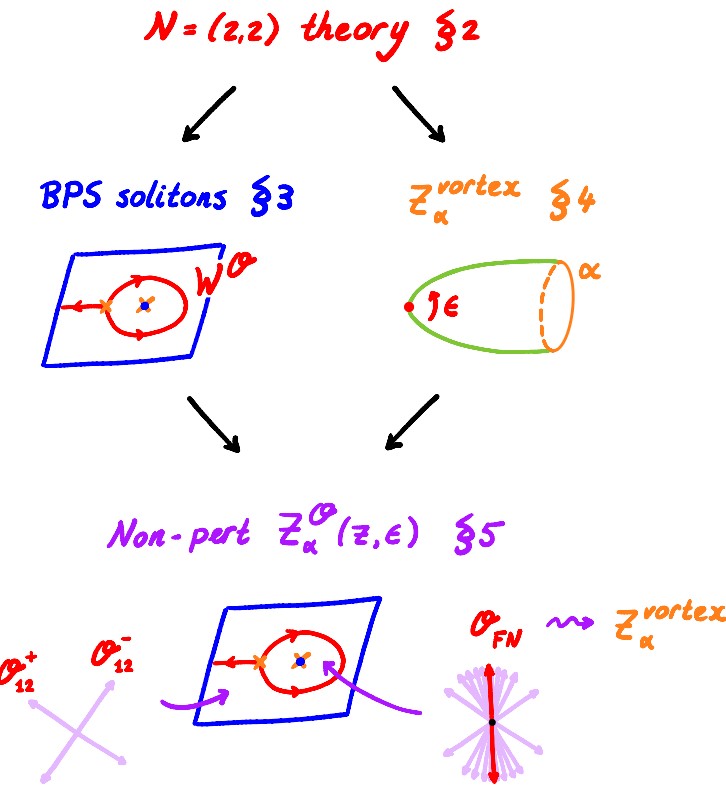

Figure 2: Schematic summary of sections 3, 4 and 5.

related collaborations and discussions.

# 2 2d $\mathcal{N} = (2, 2)$ theories and spectral geometry

In this section we analyse the vacuum structure of 2d quantum field theories that are invariant under the extended $\mathcal{N} = (2, 2)$ supersymmetry algebra.

## 2.1 Syntax of $\mathcal{N} = (2, 2)$ theories

To be self-contained, we start with a summary of some of the basic ingredients that go into defining 2d $\mathcal{N} = (2, 2)$ theories. More details can, for instance, be found in the Mirror Symmetry book [23] or in the lecture notes by Marcel Vonk [24]. Note that most of these ingredients have their origin in 4d $\mathcal{N} = 1$ theories, with the notable exception of twisted masses.

Let us make a few general remarks before we start. In most of this section, we adopt the traditional approach to describe a $d$-dimensional quantum field theory in terms of a Hilbert space together with an algebra of self-adjoint operators acting on it. This is the picture that one obtains after making a choice of space-like foliation of the $d$-dimensional space-time. Given such a choice, the Hilbert space is constructed

by considering the quantum fields on a space-like slice and imposing canonical quantization relations on them. The supersymmetry algebra is then an odd extension of the Poincaré algebra that leaves the Hilbert space invariant. Note that, even though the physics is independent of the particular space-like slice we choose in a given space-like foliation, because the quantum field theory is assumed to be time-independent, a different choice of foliation can change the fundamental structure of the theory.

It is good to keep in mind, though, that in a modern perspective, it is not necessary to make a choice of space-like slicing to define an algebra of local operators in a quantum field theory.[2] In particular, it is not necessary to represent a local operator in a quantum field theory as a self-adjoint operator acting on a particular Hilbert space. For instance, the supercharges $Q$ are intrinsically contour integrals of the so-called supercurrents $J^\mu$, and act on point-like operators $\mathcal{O}$ by enclosing their point of insertion with a small $(d-1)$-sphere $S^{d-1}$:

$$Q(\mathcal{O}) = \int_{S^{(d-1)}} J \cdot \mathcal{O}, \tag{1}$$

where $J$ is the $(d-1)$-form corresponding to the vector $J^\mu$. Only when choosing a distinguished space-like slice, the operation $Q(\mathcal{O})$ becomes the commutator $[Q, \mathcal{O}]$.[3] All algebraic relations in this section can be formulated in this more general sense.

### 2.1.1 Supersymmetry algebra

In the Lorentzian signature, the $\mathcal{N} = (2,2)$ **algebra** has four odd generators, the supercharges $Q_\pm$ and their Hermitian conjugates $\overline{Q}_\pm$. These supercharges obey the (non-zero) commutation relations

$$\{Q_\pm, \overline{Q}_\pm\} = H \pm P, \tag{2}$$
$$[iM, Q_\pm] = \mp Q_\pm, \qquad [iM, \overline{Q}_\pm] = \mp \overline{Q}_\pm,$$

with the Hamiltonian $H$, the momentum $P$ and the angular momentum $M$. Remember that $H$, $P$ and $M$ are the Noether charges for time translations $\partial_t$, spatial translations $\partial_\sigma$, and Lorentz rotations $t\partial_\sigma - \sigma\partial_t$, respectively, whereas the supercharges are the Noether charges of supersymmetry transformations.

As is common in (extended) supersymmetry algebras, we may enrich the $\mathcal{N} = (2,2)$ algebra with **central charges**, which are operators that commute with every other operator in the algebra. For the $\mathcal{N} = (2,2)$ algebra there are two possible complex central charges, commonly denoted by $Z$ and $\tilde{Z}$, as well as their complex conjugates $Z^*$ and $\tilde{Z}^*$, that enter in the anti-commutation relations

$$\{Q_+, Q_-\} = Z^* \qquad \{\overline{Q}_+, \overline{Q}_-\} = Z \tag{3}$$
$$\{Q_+, \overline{Q}_-\} = \tilde{Z}^* \qquad \{Q_-, \overline{Q}_+\} = \tilde{Z}.$$

---

[2]Formally, this algebra can be described using the concept of a factorization algebra (see the orginal books [25, 26], as well as for instance [27] for a good introduction).

[3]This aligns with the modern perspective on global symmetries (see the original paper [28], and for instance [29] for accessible lecture notes).

The $\mathcal{N} = (2,2)$ algebra may also have an internal R-symmetry $U(1)_L \times U(1)_R$, which rotates the supercharges. If we define

$$U(1)_V = \text{diag}\,(U(1)_L \times U(1)_R)\,, \tag{4}$$
$$U(1)_A = \text{anti-diag}\,(U(1)_L \times U(1)_R)\,,$$

which are known as the **vector** and the **axial R-symmetry**, respectively, then their generators $F_V$ and $F_A$ act on the supercharges as

$$[F_V, Q_\pm] = -Q_\pm, \qquad [F_A, Q_\pm] = \mp Q_\pm, \tag{5}$$
$$[F_V, \overline{Q}_\pm] = +\overline{Q}_\pm, \qquad [F_A, \overline{Q}_\pm] = \pm \overline{Q}_\pm.$$

From the relations (3) it then follows that $Z$ has to be zero when the $U(1)_V$ symmetry is conserved, while $\tilde{Z}$ has to be zero when the $U(1)_A$-symmetry is conserved. In these notes, we consider $\mathcal{N} = (2,2)$ theories for which (at least) one of the $U(1)$ R-symmetries is preserved. We sometimes denote this R-symmetry by $U(1)_R$.

The $\mathcal{N} = (2,2)$ algebra is invariant under the $\mathbb{Z}_2$ automorphism

$$Q_- \leftrightarrow \overline{Q}_-,$$
$$F_V \leftrightarrow F_A, \tag{6}$$
$$Z \leftrightarrow \tilde{Z},$$

with all other generators kept intact. This is **mirror symmetry** on the level of the supersymmetry algebra.[4]

In **Euclidean** signature, the $\mathcal{N} = (2,2)$ algebra has a slightly different form. The supercharges $Q_\pm$ and $\overline{Q}_\pm$ are now complex and independent, whereas the Lorentzian time coordinate $t$ is Wick rotated into the Euclidean time coordinate $\tau = -it$. This implies that the Hamiltonian, the momentum and the angular momentum can be expressed in terms of the complex coordinate $z = \sigma + i\tau$ and its complex conjugate. In particular, the Hamiltonian in Euclidean signature equals $H_E = iH$ whereas the rotation operator equals $M_E = iM$.

Later in these notes, we will also need to know about $\mathcal{N} = 2$ **subalgebras** of the $\mathcal{N} = (2,2)$ algebra. Such subalgebras are of the form

$$\{Q, \overline{Q}\} = 2H, \quad \{Q, Q\} = \{\overline{Q}, \overline{Q}\} = 0. \tag{7}$$

For instance, either of the linear combinations

$$Q_A^\xi = Q_- + \xi \overline{Q}_+, \tag{8}$$
$$Q_B^\xi = \overline{Q}_- + \xi \overline{Q}_+, \tag{9}$$

---

[4]It is important in mirror symmetry between LG models and GLSM's that whether the twisted central charge $\tilde{Z}$ for GLSM's picks up non-trivial quantum corrections (both perturbatively and at instanton level), because the twisted central charge $\tilde{Z}$ depends on a Kähler metric (that will be introduced in equation (24)), the central charge $Z$ for LG models can be written in terms of purely holomorphic quantities, which are protected under the renormalization flow [30].

together with their Hermitian conjugate, generates an $\mathcal{N} = 2$ subalgebra with the relations

$$\{Q_A^\xi, \overline{Q}_A^\xi\} = 2H + 2\text{Re}(\xi Z), \quad (Q_A^\xi)^2 = (\overline{Q}_A^\xi)^2 = \tilde{Z} = 0, \tag{10}$$

$$\{Q_B^\xi, \overline{Q}_B^\xi\} = 2H + 2\,\text{Re}(\xi \tilde{Z}), \quad (Q_B^\xi)^2 = (\overline{Q}_B^\xi)^2 = Z = 0, \tag{11}$$

in the Lorentzian signature. We refer to the $\mathcal{N} = 2$ subalgebra generated by $Q_A^\xi$ and $\overline{Q}_A^\xi$ (versus the one generated by $Q_B^\xi$ and $\overline{Q}_B^\xi$) as the **A-type** (and the **B-type**) subalgebra with phase $\xi$.

Note that $Z$ (and not $\tilde{Z}$) appears in the $Q_A^\xi$ commutator (10), whereas $\tilde{Z}$ appears in the $Q_B^\xi$-commutator (11). It could not have been the other way around, since the A-type subalgebra preserves the axial R-symmetry, implying that the central charge $\tilde{Z}$ is zero, while the B-type subalgebra preserves the vector R-symmetry, in turn implying that the central charge $Z$ is zero.

The above $\mathcal{N} = 2$ subalgebras are generated by a single time translation, and we therefore sometimes refer to them as being one-dimensional. They will play an important role when we study BPS solitons as well as BPS boundary conditions in the two-dimensional $\mathcal{N} = (2, 2)$ theory. More precisely, in this context we will require the $\mathcal{N} = 2$ subalgebras in **Euclidean** signature. In the latter signature the anti-commutators (10) and (11) read

$$\{Q_A^\xi, \overline{Q}_A^\xi\} = -2iH_E + 2\text{Re}(\xi Z), \quad (Q_A^\xi)^2 = (\overline{Q}_A^\xi)^2 = \tilde{Z} = 0, \tag{12}$$

$$\{Q_B^\xi, \overline{Q}_B^\xi\} = -2iH_E + 2\,\text{Re}(\xi \tilde{Z}), \quad (Q_B^\xi)^2 = (\overline{Q}_B^\xi)^2 = Z = 0. \tag{13}$$

Note that it follows from these equations, while remembering that $H_E \sim \partial_\tau \sim \text{Re}(i\partial_z)$, that rotations $z \mapsto e^{i\phi}z$ in two-dimensional space-time are correlated with rotations $Z \mapsto e^{-i\phi}Z$ in the central charge plane. In particular, this shows that $\xi$ can be thought of as a space-time rotation.

### 2.1.2 Supersymmetric fields

The two-dimensional $\mathcal{N} = (2, 2)$ fields are representations of the $\mathcal{N} = (2, 2)$ algebra. They are usually defined as functions on the $\mathcal{N} = (2, 2)$ superspace, which is an extension of two-dimensional space-time with four odd directions, parametrised by the fermionic coordinates

$$\theta^\pm, \overline{\theta}^\pm. \tag{14}$$

All $\theta$'s are anti-commuting coordinates that are related by complex conjugation,

$$(\theta^\pm)^* = \overline{\theta}^\pm, \tag{15}$$

where the $\pm$-index stands for the spin under a Lorentz transformation. Because the fermionic coordinates are anti-commuting, superfields can be Taylor expanded as monomials in the $\theta^\pm$ and $\overline{\theta}^\pm$.

Some particularly interesting classes of superfields are defined in terms of the supersymmetric covariant derivatives $D_\pm$ and $\overline{D}_\pm$. The latter are derivatives on $\mathcal{N} = (2,2)$ superspace that are defined in such a way that

$$\{D_\pm, \overline{D}_\pm\} = 2i\partial_\pm. \tag{16}$$

We have the

- **chiral superfields** $\Phi$, with $\overline{D}_\pm\Phi = 0$ and the expansion

$$\Phi = \phi + \theta^+\psi_+ + \theta^-\psi_- + \theta^+\theta^- F, \tag{17}$$

- analogously, anti-chiral superfields $\overline{\Phi}$ with $D_\pm\overline{\Phi} = 0$,

- **twisted chiral superfields** $\tilde{\Phi}$, with $\overline{D}_+\tilde{\Phi} = D_-\tilde{\Phi} = 0$ and the expansion

$$\tilde{\Phi} = \tilde{\phi} + \theta^+\tilde{\psi}_+ + \overline{\theta}^-\tilde{\psi}_- + \theta^+\overline{\theta}^- G, \tag{18}$$

- and analogously, twisted anti-chiral superfields $\overline{\tilde{\Phi}}$ with $\overline{D}_-\overline{\tilde{\Phi}} = D_+\overline{\tilde{\Phi}} = 0$.

There is some flexibility in assigning R-charges to these superfields, but usually we take the $U(1)_V$ charge of an (anti-)chiral superfield to be 1 and its $U(1)_A$-charge to be 0. This is the other way around for twisted (anti-)chiral superfields.

Implementing gauge symmetry requires the introduction of an additional vector superfield $V$, which encodes the gauge connection and its superpartners

$$\begin{aligned}
V = {} &\theta^-\bar{\theta}^-(v_0 - v_1) + \theta^+\bar{\theta}^+(v_0 + v_1) - \theta^-\bar{\theta}^+\sigma - \theta^+\bar{\theta}^-\bar{\sigma} \\
&+ \sqrt{2}i\,\theta^-\theta^+\left(\bar{\theta}^-\bar{\lambda}_- + \bar{\theta}^+\bar{\lambda}_+\right) + \sqrt{2}i\,\bar{\theta}^+\bar{\theta}^-\left(\theta^-\lambda_- + \theta^+\lambda_+\right) + 2\,\theta^-\theta^+\bar{\theta}^+\bar{\theta}^- D.
\end{aligned} \tag{19}$$

The **vector superfield** $V$ transforms as

$$V \mapsto V + \Lambda + \overline{\Lambda} \tag{20}$$

under gauge transformations parametrized by a chiral superfield parameter $\Lambda$. The vector superfield $V$ is forced to be neutral under both the axial and the vector R-symmetry.

It is then natural to define the gauge-covariant superderivatives

$$\begin{aligned}
\mathcal{D}_\alpha &= e^{-V} D_\alpha\, e^V, \\
\overline{\mathcal{D}}_{\dot{\alpha}} &= e^V \overline{D}_{\dot{\alpha}}\, e^{-V},
\end{aligned} \tag{21}$$

as well as the twisted chiral superfield

$$\Sigma = \frac{1}{\sqrt{2}}\{\overline{\mathcal{D}}_+, \mathcal{D}_-\}, \tag{22}$$

which encodes the two-dimensional field strength $F_{01}$ in its auxiliary component

$$\Sigma = \sigma + \theta^+\tilde{\lambda}_+ + \overline{\theta}^-\lambda_- + \theta^+\overline{\theta}^-(D - iF_{01}). \tag{23}$$

The twisted chiral superfield $\Sigma$ has $U(1)_A$ charge 2 and $U(1)_V$ charge 0.

### 2.1.3 Supersymmetric Lagrangian

$\mathcal{N} = (2,2)$ supersymmetry places severe constraints on the form of the Lagrangian. We give a full list of the allowed terms below. Let $\Phi$ (resp. $\tilde{\Phi}$) be the collection of (resp. twisted) chiral superfields in the theory, and let $V$ and $\Sigma$ be the vector and twisted chiral superfields as defined above.

- Matter field kinetic terms take the form

$$\int d^2\theta d^2\overline{\theta}\, K(\Phi, \tilde{\Phi}, \Phi^\dagger, \tilde{\Phi}^\dagger) = \int d^2\theta d^2\overline{\theta}\, g_{i\bar{j}}(\Phi^i, \overline{\Phi}^j)\, \overline{\Phi^j}\, \Phi^i, \qquad (24)$$

where $\Phi^i$ represents both the chiral and twisted chiral superfields. Because of $\mathcal{N} = (2,2)$ supersymmetry, these fields define local coordinates on a Kähler manifold $X$ with Kähler metric $g_{i\bar{j}}$. The function $K$ is the so-called **Kähler potential** for this Kähler metric. To incorporate gauge symmetry we replace

$$K(\Phi, \Phi^\dagger) \to K(e^V \Phi, \Phi^\dagger e^V), \qquad (25)$$

where $V$ acts in the appropriate representation. This simply replaces derivatives in the action by gauge covariant derivatives.

- Gauge kinetic terms take the form

$$-\int d^4\theta\, \frac{1}{2e^2}\, \bar{\Sigma}\Sigma. \qquad (26)$$

where $e$ is the gauge coupling.

- Superpotential terms take the form

$$\int d^2\theta\, W(\Phi) + h.c. \qquad (27)$$

where the **superpotential** $W(\Phi)$ is given by a (a priori) holomorphic function on (exclusively) the chiral superfields $\Phi$. Twisted superpotential terms similarly take the form

$$\int d^2\tilde{\theta}\, \widetilde{W}(\tilde{\Phi}) + h.c. \qquad (28)$$

where the **twisted superpotential** $\widetilde{W}(\tilde{\Phi})$ is given by a (a priori) holomorphic function on (exclusively) the twisted chiral superfields $\tilde{\Phi}$. In a gauge theory the (twisted) superpotential $W$ must evidently be gauge invariant.

(Twisted) superpotential terms contribute a factor proportional to

$$\int_{\mathbb{R}^{1,1}} d\sigma d\tau\, g^{\bar{i}j}\, \overline{\partial_i W}\, \partial_j W \qquad (29)$$

to the potential energy (and similar for the twisted superpotential). This suggests that one can generalise the (twisted) superpotential into a closed (but not necessarily exact) holomorphic 1-form

$$dW = \partial_i W \, d\Phi^i \tag{30}$$

on $X$ (see §3 of [31] for more details). This will be important in §4 when we introduce GLSM's.

Superpotential terms are always invariant under the axial $R$-symmetry, but only invariant under (the full) vector $R$-symmetry when it is possible to assign vector $R$-charges to the chiral superfields so that the each term in the superpotential carries total vector $R$-charge 2. For twisted superpotentials it is the other way around.

- We can introduce **complex masses** for the chiral superfields via superpotential terms. For instance, in a non-abelian gauge theory with chiral fields $\Phi^i$ in the fundamental representation as well as chiral fields $\hat{\Phi}_{\hat{j}}$ in the anti-fundamental representation, we can write down the interaction term[5]

$$\int d^2\theta \, m_i^{\hat{j}} \, \hat{\Phi}_{\hat{j}} \Phi^i + \text{h.c.,} \tag{31}$$

where the complex masses $m_i^{\hat{j}}$ are really the expectation value of a background chiral superfield. The same is true for twisted chiral superfields, for which the complex masses should instead be viewed as the expectation value of a background twisted chiral superfield.

- We can also introduce **twisted masses** $\widetilde{m}$ for any chiral superfield $\Phi$ via the Kähler potential term[6]

$$\int d^4\theta \, \Phi^\dagger e^{2V_{\text{bg}}} \, \Phi. \tag{32}$$

where $V_{\text{bg}}$ is an abelian *background* vector superfield whose only nonzero components are

$$V_{\text{bg}} = -\theta^-\bar{\theta}^+\widetilde{m} - \theta^+\bar{\theta}^-\widetilde{m}^\dagger. \tag{33}$$

The precise procedure for obtaining these interaction terms is to gauge the (maximal abelian subgroup of the) flavour symmetry, take the weak coupling

---

[5]This interaction term orginates in four dimensions .

[6]This twisted mass term does not have an analogue in four dimensions. Yet, one can introduce it through a four-dimensional construction: Start with a 4d $\mathcal{N} = 1$ theory of gauged chiral superfields and dimensionally reduce it to two dimensions, while setting the vector potential in the $3-4$ directions (the ones we are reducing along) to a constant value. This simultaneously breaks rotational symmetry along the $3-4$ directions (which corresponds to the axial R-symmetry in the resulting 2d $\mathcal{N} = (2,2)$ theory) and turns on a purely two-dimensional mass term.

limit ($e \to \infty$) to make the gauge kinetic term disappear, and then give the resulting background vector superfield(s) $V_{\text{bg}}$ an expectation value as in (33). The same procedure introduces twisted masses for twisted chiral superfields.

Turning on twisted masses breaks the axial R-symmetry. As we will see in §4 turning on twisted masses is mirror symmetric to turning on a 1-form superpotential, which in turn breaks the vector R-symmetry.[7]

Throughout these notes we will see many different combinations and explicit examples of these ingredients.

## 2.2 Vacuum structure

To understand the vacuum structure of 2d $\mathcal{N} = (2, 2)$ theories, we consider a particular type of operators in the theory that preserve half of the supersymmetry.

### 2.2.1 Chiral ring and twisted chiral ring

Operators $\mathcal{O}$ in the $\mathcal{N} = (2, 2)$ theory may be thought of as operator-valued products of fields. They are called:[8]

- **chiral** if $[\overline{Q}_\pm, \mathcal{O}] = 0$

- anti-chiral if $[Q_\pm, \mathcal{O}] = 0$,

- **twisted chiral** if $[\overline{Q}_+, \mathcal{O}] = [Q_-, \mathcal{O}] = 0$,

- twisted anti-chiral if $[Q_+, \mathcal{O}] = [\overline{Q}_-, \mathcal{O}] = 0$.

Since half of the supercharges act trivially on such operators, they are "half-BPS" operators. Similar to a conformal field theory, where one can define a product structure between primary operators, the (twisted) chiral operators defined above form a ring, the so-called **(twisted) chiral ring**. This is because if any two operators $\mathcal{O}_\alpha$ and $\mathcal{O}_\beta$ are (twisted) chiral, then their product $\mathcal{O}_\alpha \mathcal{O}_\beta$ is also (twisted) chiral. Note that the the chiral ring is graded by the $U(1)_V$-symmetry, whereas twisted chiral ring is graded by the $U(1)_A$-symmetry, These rings are discussed further in §2.2.3.

The (twisted) chiral ring is closely related to the cohomology of the nilpotent supercharges[9]

---

[7]Turning on a superpotential breaks $U(1)_V$ because it introduces a non-vanishing central charge $Z$.

[8]As discussed in the introduction to §2, remember that the condition $[Q, \mathcal{O}]$ can be phrased in more generality (without having to make a choice of space-like slice) as $Q(\mathcal{O}) = 0$.

[9]To be precise, the supercharges $Q_A$ (or $Q_B$) are only nilpotent on the nose if $\tilde{Z} = 0$ (or $Z = 0$), see equations (12) and (13). Yet, it is common to consider the $Q_A$-cohomology (or $Q_B$-cohomology) in the preserve of non-trivial $\tilde{Z}$ (or $Z$), for instance, when we turn on twisted masses or a (twisted) superpotential. Such deformations of the theory are tied with global symmetries, and the supercharge $Q$ should in this case be thought of as an equivariant differential instead.

$$Q_A^\xi = Q_- + \xi\overline{Q}_+ \quad \text{and} \quad Q_B^\xi = \overline{Q}_- + \xi\overline{Q}_+. \tag{34}$$

Indeed,

$$\text{twisted chiral} \implies Q_A^\xi\text{-closed}, \tag{35}$$

$$\text{chiral} \implies Q_B^\xi\text{-closed}. \tag{36}$$

It is easy to show that there is a one-one correspondence between (twisted) chiral operators and $Q_{A,B}$−cohomology, if we further assume that both the axial *as well as* the vector R-symmetry are conserved (with the implication that both central charges $Z = \tilde{Z} = 0$ vanish). The key observation is that the components of $Q_A$ transform with opposite charges under the $U(1)_V$-symmetry, whereas the components of $Q_B$ transform with opposite charges under the $U(1)_A$-symmetry. Consider the $Q_B$-charge and the equation $[Q_B, \mathcal{O}] = 0$ for now; the argument for $Q_A$ is similar. Also, assume (without loss of generality) that the operator $\mathcal{O}$ has a definite $F_A$-charge. Then, acting on the equation $[Q_B, \mathcal{O}] = 0$ with $F_A$, and taking linear combinations of this and the original equation, we conclude that both $[\bar{Q}_\pm, \mathcal{O}] = 0$.

In case one of the central charges does not vanish, the space of (twisted) chiral operators may only be a proper subset of the $Q_{A,B}$-cohomology. Yet, it turns out that we do not need to worry about such instances in these lecture notes. In all our examples we can identify the (twisted) chiral ring with the $Q_{A,B}$-cohomology[10].

As an example, let us determine the $Q_B$-cohomology of an $\mathcal{N} = (2,2)$ theory formulated in terms of free chiral superfields $\Phi$ (see for instance [32] for more details). Since

$$Q_B\phi = 0, \tag{37}$$

while

$$Q_B\overline{\phi} = -(\overline{\psi}_+ + \overline{\psi}_-) \quad \text{and} \quad Q_B(\overline{\psi}_+ + \overline{\psi}_-) = 0, \tag{38}$$

we can think of the fields $\phi$ and $\overline{\phi}$ as coordinates on a complex manifold, say $X$, and interpret $Q_B$ as the Dolbeault operator $\bar{\partial}$ on $X$. This implies that the $Q_B$-cohomology contains the Dolbeault cohomology $H_{\bar{\partial}}^*(X) = H^*(\Omega^{0,*}(X), \bar{\partial})$ (of functions on $X$).

With additional arguments one can show that the linear combination $\overline{\psi}_+ - \overline{\psi}_-$ transforms as the holomorphic vector field $\frac{\partial}{\partial\phi}$, and that space-time derivatives of all fields are $Q_B$-exact. so that the full $Q_B$-cohomology equals the Dolbeault cohomology of polyvector fields,

$$H^*(\Omega^{0,*}(X) \otimes \wedge^* TX, \bar{\partial}), \tag{39}$$

---

[10]This is because we restrict ourselves to so-called massive $\mathcal{N} = (2,2)$ theories. As we will see in later sections, such theories may be described in terms of a superpotential $W$ with a finite set of non-degenerate critical points (that is, $W$ is a Morse function). If we would instead consider a superpotential such as $W = XYZ$ on $\mathbb{C}^3$ with coordinates $X$, $Y$ and $Z$, we would find that its $Q$-cohomology is strictly larger than its chiral ring.

where $\bar{\partial}$ acts trivially on the tangent vector $\frac{\partial}{\partial\phi}$. If we furthermore add a superpotential $W(\phi)$ to the theory, the Dolbeault operator $\bar{\partial}$ gets deformed into the equivariant differential $\bar{\partial} + \partial W \wedge$ acting on the polyvector fields $\Omega^{0,*}(X) \otimes \mathbb{C}[\frac{\partial}{\partial\phi}]$.

### 2.2.2 Topological twists

One way to think about the $Q^{\xi}_{A,B}$-cohomology is in terms of **topological twists** (see for instance [24] for an introduction). Topological twisting is a tool to preserve a part of the supersymmetry algebra on a curved Euclidean space-time. The goal is to define a new rotation group as a subgroup of the product of the old rotation group and the R-symmetry group, in such a way that (at least) a subset of the supersymmetry generators transform as scalars with respect to the new rotation group, and can thus be preserved on the curved space-time.

In the $\mathcal{N} = (2,2)$ theory we may choose the new two-dimensional rotation group $U(1)'_E$ to be the diagonal subgroup

$$U(1)'_E = \mathrm{diag}\,(U(1)_E \times U(1)_R) \tag{40}$$

of the old $U(1)_E$, which is generated by $M_E = iM$, and with $U(1)_R$ being either the vector R-symmetry group generated by $F_V$, or the axial R-symmetry group generated by $F_A$. That is, the new Lorentz generator is either

$$M_A = M_E + F_V, \quad \text{or} \tag{41}$$
$$M_B = M_E + F_A. \tag{42}$$

The resulting topological twists are known as the **A-twist** and the **B-twist**, respectively. In the A-twist we find that $Q_-$ and $\overline{Q}_+$ transform as scalars under the new rotation group, so that any $Q = Q^{\xi}_A$ can be picked as the new scalar supercharge. In the B-twist we find that $\overline{Q}_-$ and $\overline{Q}_+$ transform as scalars, so that any $Q = Q^{\xi}_B$ can be chosen as the scalar supercharge. To summarize,

$$\text{A-twist}: \quad U(1)_R = U(1)_V \implies Q = Q^{\xi}_A \text{ is scalar}, \tag{43}$$
$$\text{B-twist}: \quad U(1)_R = U(1)_A \implies Q = Q^{\xi}_B \text{ is scalar}. \tag{44}$$

The Lagrangian of the resulting twisted theory is $Q$-closed, and can be shown to be $Q$-exact up to terms that do not depend on the metric on the 2d space-time. The latter terms are usually referred to as "topological terms".[11] Since the $Q$-exact terms are essentially trivial in the twisted theory, this implies that the twisted theory does not depend on the metric and is therefore topological. Furthermore, if we consider just the $Q$-closed operators as physical observables, and keep in mind that $Q$-exact operators are essentially trivial, the physical observables in the twisted

---

[11]Topological terms for a non-linear sigma model describing maps from a 2d worldsheet into $X$ only depend on the Kähler structure of the target $X$ in the A-twist, whereas they only depend on the complex structure of the target $X$ in the B-twist.

theory are characterized by the $Q$-cohomology. We have already seen in §2.2.1 that the $Q_B^\xi$-cohomology may be identified with the classical Dolbeault cohomology. The $Q_A^\xi$-cohomology is instead a quantum-corrected version of the classical de Rham cohomology, which is known as **quantum cohomology**. For more details see for instance [33, 32]. This will come up again in §4.2.

Another helpful way to think of the topological twist is to turn on a specific **background gauge field** $A^R$ for the $U(1)_R$-symmetry that we use to twist. Such a background connection does not introduce any dynamical (Yang-Mills like) terms in the Lagrangian (hence the word background), but changes the covariant derivative

$$D_\mu = \partial_\mu + \omega_\mu \quad \mapsto \quad D'_\mu = \partial_\mu + \omega'_\mu = \partial_\mu + \omega_\mu + A^R_\mu, \tag{45}$$

where $\omega_\mu$ is the original and $\omega'_\mu$ the new spin connection. Indeed, if we choose

$$A^R_\mu = \frac{1}{2}\omega_\mu, \tag{46}$$

the supercharges whose charge under $M_E$ and $F$ add up to zero, transform as scalars under the new covariant derivative.

### 2.2.3 Supersymmetric ground states

Consider any $\mathcal{N} = (2,2)$ theory on a cylinder $S^1 \times \mathbb{R}$ with $S^1$ the spatial direction and $\mathbb{R}$ the time-like direction. Denote the corresponding Hilbert space by $\mathcal{H}$.

The ground states of the resulting theory can be characterized as those states in the Hilbert space $\mathcal{H}$ that are annihilated by half of the supercharges. Indeed, ground states $|\alpha\rangle$ in a supersymmetric theory necessarily have energy $H|\alpha\rangle = 0$, and this is equivalent to $Q|\alpha\rangle = \overline{Q}|\alpha\rangle = 0$ for either $Q = Q_A$ or $Q = Q_B$. We thus define the spaces $V_A$ and $V_B$ as

$$V_{A,B} = \{|\psi\rangle \in \mathcal{H} : Q_{A,B}|\psi\rangle = \overline{Q}_{A,B}|\psi\rangle = 0\}. \tag{47}$$

Henceforth, we shall simply write $V$ for $V_{A,B}$ and $Q$ for $Q_{A,B}$ to simplify the notation.

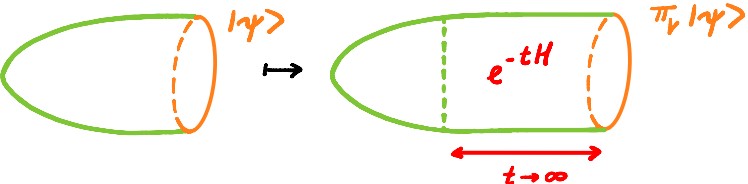

Figure 3: The projection map $\pi_V$ is defined by making the cigar-like geometry infinitely long.

Any state $|\psi\rangle$ in the Hilbert space $\mathcal{H}$ has a natural projection to the space $V$ of ground states. A nice argument for this is to consider the topologically twisted

theory on a cigar-like geometry, with a flat metric sufficiently far away from the tip.[12] Since the topological theory is identical to the physical untwisted theory in the flat region, the Hilbert space associated to the boundary of the cigar is $\mathcal{H}$. In the limit in which we make the flat region infinitely large, only the part of the state $|\psi\rangle$ that is annihilated by the Hamiltonian $H$ is preserved. This defines the projection

$$\pi_V : \mathcal{H} \to V$$
$$|\psi\rangle \mapsto \lim_{t\to\infty} e^{-tH}|\psi\rangle, \tag{48}$$

which is visualized in Figure 3.

The projection $\pi_V^\dagger$ of the path integral $\langle Z| \in \mathcal{H}^*$ – without any additional operator insertions – of the $\mathcal{N} = (2,2)$ theory on the cigar-like geometry, determines a distinguished ground state $\langle 0| \in V^*$. This state is sometimes referred to as the state generated by the smooth tip of the cigar.

In the following, we make the additional assumption that the **pairing**

$$\eta = \langle \,.\,|\,.\,\rangle : \ V^* \times V \to \mathbb{C} \tag{49}$$

is non-degenerate. This assumption automatically holds in these notes because we restrict ourselves to massive $\mathcal{N} = (2,2)$ theories with a finite number of vacua. With this assumption we continue to show that there is a 1-1 correspondence between the space $V$ of ground states and the $Q$-cohomology – or (twisted) chiral ring.

Note that since the pairing $\eta$ is assumed to be non-degenerate, we can construct a diagonal basis on $V$. From now on, we will label the elements of this diagonal basis as $|\alpha_l\rangle$ or simply $|\alpha\rangle$.

**Claim 1:** In each $Q$-cohomology class $[\mathcal{O}']$ there is an operator $\mathcal{O}$ such that

$$|\mathcal{O}\rangle := \mathcal{O}|0\rangle \tag{50}$$

is a ground state.

**Proof:** Suppose $\mathcal{O}'$ is a $Q$-closed operator. First note that the projection operator $\pi_V$ only depends on the cohomology class of $\mathcal{O}'$. That is, for any operator $\Lambda$, we have that

$$\pi_V\left(\mathcal{O}'|0\rangle\right) = \pi_V\left((\mathcal{O}' + [Q,\Lambda])|0\rangle\right). \tag{51}$$

This easily follows from the relation $[H, Q] = 0$.

Using Hodge theory we then choose the (unique) **harmonic** representative of the cohomology class of $\mathcal{O}'$. This is the operator $\mathcal{O}$ which is obtained from $\mathcal{O}'$ by adding $Q$-exact terms such that $\mathcal{O}$ is not only $Q$-closed, but also $\overline{Q}$-closed.

Then, we find that

$$Q|\mathcal{O}\rangle = \overline{Q}|\mathcal{O}\rangle = 0, \tag{52}$$

---

[12]More details on this cigar geometry can be found in §5.2.

which implies that $|\mathcal{O}\rangle$ is indeed a ground state. $\square$

**Claim 2:** The **state-operator correspondence** defines a mapping between ground states $\alpha$ and local operators $\mathcal{O}_\alpha$, such that

$$|\alpha\rangle = \mathcal{O}_\alpha|0\rangle. \tag{53}$$

This mapping is a bijection iff the pairing $\eta_{\alpha\beta}$ is non-degenerate.

**Proof:** Each local operator $\mathcal{O}_\alpha$ defines an element in the dual of $V$ through

$$\mathcal{O}_\alpha : |\beta\rangle \mapsto \langle\alpha|\beta\rangle. \tag{54}$$

This gives a bijection between the space of operators and $V^*$ because the pairing $\eta_{\alpha\beta}$ is non-degenerate. Furthermore, the mapping $\Phi : V \to V^*$ defined by

$$|\alpha\rangle \mapsto (\phi_\alpha : |\beta\rangle \mapsto \langle\alpha|\beta\rangle) \tag{55}$$

is a bijection iff $\eta_{\alpha\beta}$ is non-degenerate. $\square$

Note that the states $|\alpha\rangle$ form a basis of $V$ iff the operators $\mathcal{O}_\alpha$ form a basis of $V^*$.

**Claim 3:** The operator $\mathcal{O}_\alpha$ defines an automorphism on $V$ and is $Q$-closed.

**Proof:** To show that $\mathcal{O}_\alpha$ defines a bijection on $V$ we just need to show that it is injective, i.e.

$$\mathcal{O}_\alpha|\beta\rangle = \mathcal{O}_\alpha|\beta'\rangle \Rightarrow |\beta\rangle = |\beta'\rangle. \tag{56}$$

This follows by contracting with the vacuum state $\langle 0|$ and using that $\eta_{\alpha\beta}$ is non-degenerate. Moreover, since $|\alpha\rangle$ is a ground state, we have that

$$Q|\alpha\rangle = \overline{Q}|\alpha\rangle = 0. \tag{57}$$

In particular, this implies that $[Q, \mathcal{O}_\alpha] = 0. \square$

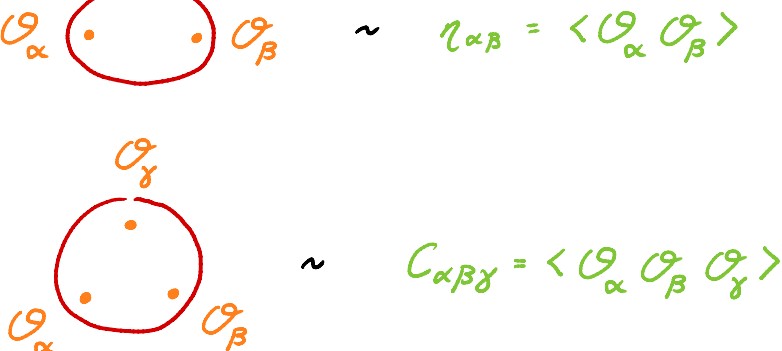

Figure 4: Two- and three-point functions $\eta_{\alpha\beta}$ and $C_{\alpha\beta\gamma}$ in a 2d TFT.

Using the state-operator correspondence, we can interpret the pairing on $V$ as a correlation function of two $Q$-closed operators $\mathcal{O}_\alpha$ and $\mathcal{O}_\beta$ on the two-sphere. This is illustrated on top in Figure 4. Note that

$$\langle\alpha|\beta\rangle = \langle 0|\mathcal{O}_\alpha^\dagger \mathcal{O}_\beta|0\rangle = \langle 0|\mathcal{O}_\alpha \mathcal{O}_\beta|0\rangle, \tag{58}$$

because moving the operator insertion $\mathcal{O}_\alpha$ corresponds to a $Q$-exact deformation. And note that with the same reasoning we find that

$$\langle\alpha|\beta\rangle = \langle 0|\mathcal{O}_\alpha\mathcal{O}_\beta|0\rangle = \langle 0|\mathcal{O}_\beta\mathcal{O}_\alpha|0\rangle = \langle\beta|\alpha\rangle, \tag{59}$$

showing that the chiral ring structure is **commutative**.

We thus find that the $Q$-cohomology – or (twisted) chiral ring – is a commutative ring with unit given by identity operator $\mathbf{1}$, corresponding to the vacuum $|0\rangle$. The multiplicative structure of the chiral ring is encoded in the **three-point functions**

$$C_{\alpha\beta\gamma} = \langle\mathcal{O}_\alpha\mathcal{O}_\beta\mathcal{O}_\gamma\rangle, \tag{60}$$

which are illustrated at the bottom of Figure 4.

Indeed, using the argument illustrated in Figure 5 we find that

$$\mathcal{O}_\alpha\mathcal{O}_\beta = \sum_\gamma C_{\alpha\beta}{}^\gamma\,\mathcal{O}_\gamma, \tag{61}$$

where $C_{\alpha\beta}{}^\gamma = \sum_\gamma C_{\alpha\beta\eta}\,\eta^{\eta\gamma}$. The idea is that, using topological invariance in the twisted theory, one stretches out a long tube where only asymptotic states survive. These can then be represented as chiral operator insertions through the state-operator correspondence. We sum over these insertions using a matrix, which one can show (by repeating the same procedure in the picture, starting with only one insertion) is the inverse of the two-point function.

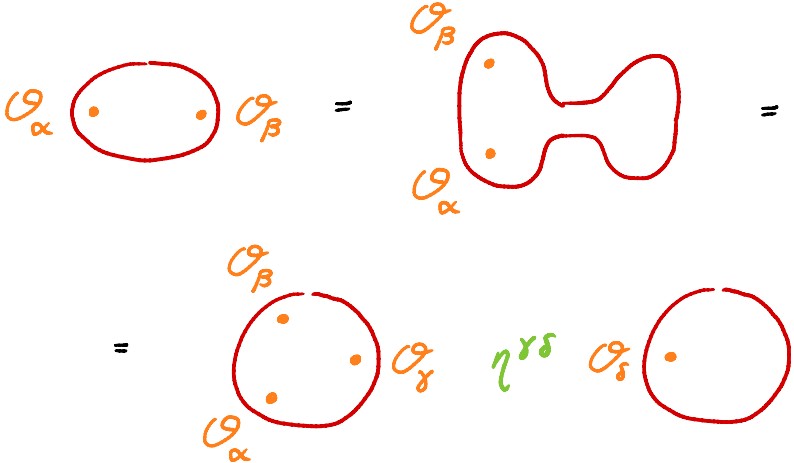

Figure 5: Pictorial derivation of the (twisted) chiral structure for a 2d TFT

Let us illustrate the chiral ring structure with a few examples.

### 2.2.4    Example: Landau-Ginzburg models

Landau-Ginzburg (LG) models are a rich class of 2d $\mathcal{N} = (2,2)$ theories built out of $n$ chiral superfields $\Phi^i$ and a superpotential $W(\Phi^i)$. The fields $\Phi^i$ take values in a

Kähler manifold $X$ [13]. The Kähler metric

$$g_{i\bar{j}} = \partial_i \partial_{\bar{j}} K(\Phi^i, \overline{\Phi^i}) \tag{62}$$

is defined in terms of a Kähler potential $K$. We will assume the superpotential $W : X \to \mathbb{C}$ to be a holomorphic function on $X$.[14] For us, the Kähler manifold $X$ will just be $\mathbb{C}^n$. Spelling this out in components of the chiral superfields $\Phi^i$, reveals that the bosonic part of the Lagrangian is given by

$$\mathcal{L}_{\text{bos}} = g_{i\bar{j}} \partial^\mu \phi^i \partial_\mu \overline{\phi^j} + \frac{1}{4} g^{\bar{i}j} \overline{\partial_i W} \partial_j W, \tag{63}$$

after integrating out the auxiliary fields.

The perturbative vacua of the theory are therefore given by the critical points of $W$, i.e. the solutions of $\partial_i W = 0$. In the following we assume that the Hessian of $W$ (the matrix of second derivatives) is non-degenerate at every critical point. This implies that the theory is massive, i.e. it has a discrete spectrum and has no massless modes in any vacuum. As a consequence, there is a natural length-scale in the theory.

Suppose that the superpotential $W(\phi^i)$ of a Landau-Ginzburg model is a holomorphic function of some parameters $\mathbf{z} \in C$. Since the chiral ring is dual to the space of vacua, it can be identified with the **Jacobian ring**

$$E_{\mathbf{z}} = \mathbb{C}[\phi^i]/\langle \partial_i W \rangle, \tag{64}$$

where $\langle \partial_i W \rangle$ is the ideal generated by the $\partial_i W$.

As a concrete example, consider the LG model with a single superfield and the cubic superpotential

$$W(\phi) = \frac{1}{3} \phi^3 - \mathbf{z} \, \phi, \tag{65}$$

so that $C = \mathbb{C}_{\mathbf{z}}$. This is a deformation of the LG model with the quasi-homogeneous superpotential $W(\phi) = \frac{1}{3} \phi^3$ that flows in the IR to the conformal $A_2$ model. Since $\partial_\phi W = \phi^2 - \mathbf{z}$, the chiral ring is generated by the fields $1$ and $\phi$ with the relation

$$\phi^2 = \mathbf{z}. \tag{66}$$

## 2.3 Descendants and deformations

Consider a $d$-dimensional field theory with nilpotent supercharge $Q^2 = 0$, such that the stress-energy tensor $T_{\alpha\beta}$ is $Q$-exact. This implies that the linear momentum operator is also $Q$-exact:

$$T_{\alpha\beta} = \{Q, G_{\alpha\beta}\} \implies P_\alpha = \{Q, G_\alpha\}. \tag{67}$$

---

[13]That is, the bosonic components $\phi^i$ of the chiral superfields $\Phi^i$ are maps $\phi^i : \mathbb{R}^2 \to X$, whereas the fermionic coordinates $\psi_\pm$ are spinors on $\mathbb{R}^2$ valued in the pull-back of the tangent bundle $\phi^* T_X$.

[14]In §4 we will introduce a generalised class of superpotentials corresponding to holomorphic 1-forms on $X$.

Take a local bosonic $Q$-invariant operator $\mathcal{O}^{(0)}$ and build the following supermultiplet:

$$e^{\theta^\alpha G_\alpha}\mathcal{O}^{(0)} = \mathcal{O}^{(0)} + \theta^\alpha \mathcal{O}^{(1)}_\alpha + \cdots + \theta^{\alpha_1}...\theta^{\alpha_d}\mathcal{O}^{(d)}_{\alpha_1...\alpha_d}, \tag{68}$$

where the $\theta_\alpha$ are anticommuting variables. Clearly, the operators $\mathcal{O}^{(n)}_{\alpha_1...\alpha_n}$ have anti-commuting indices and can be promoted to differential forms

$$\mathcal{O}^{(n)} = \mathcal{O}^{(n)}_{\alpha_1...\alpha_n}dx^{\alpha_1}...dx^{\alpha_n}. \tag{69}$$

It can be easily shown that the operators satisfy the so-called **descent equations**

$$\{Q, \mathcal{O}^{(n+1)}\} = d\mathcal{O}^{(n)}. \tag{70}$$

$\mathcal{O}^{(n)}$ is called the degree $n$ descendant of $\mathcal{O}^{(0)}$. One can then build $Q$-invariant non-local operators by integrating the descendants on non-trivial cycles of spacetime:

$$\int_{\Sigma_n} \mathcal{O}^{(n)}. \tag{71}$$

Now, consider a deformation of a $2d\ \mathcal{N} = (2,2)$ theory

$$\delta S = \delta\mathbf{z}\int_{\mathbb{R}^2}\mathcal{O}, \tag{72}$$

were $\mathcal{O}$ is a priori just a $2$-form operator. We ask that it preserve the nilpotent supercharge $Q$. This has the following consequences:

$$\begin{aligned}
&\{Q, \int_{\mathbb{R}^2}\mathcal{O}\} = 0 \\
&\implies \{Q, \mathcal{O}\} = d\mathcal{O}^{(1)} \implies \{Q, \{Q, \mathcal{O}\}\} = -d\{Q, \mathcal{O}^{(1)}\} = 0 \\
&\implies \{Q, \mathcal{O}^{(1)}\} = d\mathcal{O}^{(0)} \implies \{Q, \{Q, \mathcal{O}^{(1)}\}\} = -d\{Q, \mathcal{O}^{(0)}\} = 0 \\
&\implies \{Q, \mathcal{O}^{(0)}\} \in \mathbb{C}.
\end{aligned} \tag{73}$$

After requiring that $\mathcal{O}$ (and so $\mathcal{O}^{(0)}$) be bosonic, an inspection of the available fields in the theory and their supersymmetry transformations allows one to conclude that $\{Q, \mathcal{O}^{(0)}\}$ must be zero. This implies that $\mathcal{O} = \mathcal{O}^{(2)}$ is the cohomological descendant of a local bosonic $Q$-invariant operator $\mathcal{O}^{(0)}$.

We have thus shown that deformations of any $\mathcal{N} = (2,2)$ theory that are invariant under a nilpotent supercharge $Q$, are in one-to-one correspondence with local bosonic $Q$-invariant operators. Two important classes of such deformations are built from twisted chiral and chiral operators $\mathcal{O}^{(0)}$, corresponding to the nilpotent supercharges $Q_A$ and $Q_B$, respectively.

### 2.3.1 Example: perturbations of LG models

In the context of Landau-Ginzburg models the natural question to ask is how do the above described deformations modify the superpotential? The solution to the descent equations for $Q_B$ reads

$$\mathcal{O}^{(2)} = -dzd\bar{z}\,\frac{1}{2}\{Q_+, \{Q_-, \mathcal{O}^{(0)}\}\}. \tag{74}$$

Consider an LG model with a single chiral superfield $\Phi = \phi + \dots$ and chiral operator $\mathcal{O}(\phi)$. After a relatively mild computation one alights at

$$\delta\mathcal{L} = \delta z \left(-\partial_\phi^2 \mathcal{O}(\phi)\,\psi_-\psi_+ - \partial_\phi \mathcal{O}(\phi)F\right). \tag{75}$$

Then, comparing the above to the superpotential terms in the lagrangian

$$\mathcal{L}_W = -\partial_\phi^2 W(\phi)\,\psi_-\psi_+ - \partial_\phi W(\phi)F \tag{76}$$

one concludes that perturbing the action by the descendant of a chiral operator is equivalent to perturbing the superpotential by the same chiral operator:

$$W(\phi) \to W(\phi) + \delta\mathbf{z}\,\mathcal{O}(\phi), \tag{77}$$

where $\delta\mathbf{z}$ is the deformation parameter.

In particular, given the superpotential $W(\phi) = \frac{\phi^k}{k}$ with the associated chiral ring $\mathbb{C}[\phi]/\langle\phi^{k-1}\rangle$ spanned by $\{1, \phi, \dots, \phi^{k-2}\}$, it is natural to add all possible $Q_B$-invariant perturbations to $W(\phi)$ and to work instead with the superpotential

$$W(\phi, \mathbf{z}^{(1)}, \dots, \mathbf{z}^{(k-2)}) = \frac{\phi^k}{k} + \mathbf{z}^{(k-2)}\,\frac{\phi^{k-2}}{k-2} + \dots + \mathbf{z}^{(1)}\phi. \tag{78}$$

## 2.4 Spectral geometry

Consider a family $\mathcal{T}_\mathbf{z}$ of massive $\mathcal{N} = (2,2)$ theories that depend on a set of parameters $\mathbf{z}$ which parameterise (twisted) chiral deformations. In this case, the (twisted) chiral ring defines a holomorphic bundle $\mathcal{E}$ of commutative algebras $E_\mathbf{z}$ over the parameter space $C$. Moreover, because infinitesimal deformations can be constructed by perturbing the Lagrangian by $Q_{A,B}$−closed operators (as we have seen in §2.3), we find that there exists a holomorphic map of vector bundles

$$q: TC \to \mathcal{E}, \tag{79}$$

that sends the tangent vector $\partial_{\mathbf{z}^{(\alpha)}}$ to the $Q_{A,B}$−closed operator $\mathcal{O}_\alpha$.

Dually, we may consider the **spectrum** $\Sigma_\mathbf{z}$ of the commutative algebra $E_\mathbf{z}$. The point of $\Sigma_\mathbf{z}$ are in 1-1 correspondence with the ground states of the two-dimensional theory $\mathcal{T}_\mathbf{z}$. If our two-dimensional theory is a massive theory, i.e. it has a discrete

set of ground states and a mass-gap, then the ground states sweep out a branched covering

$$\Sigma \to C. \tag{80}$$

We will refer to this branched cover $\Sigma$ as the **spectral curve**.

For instance, the spectral curve $\Sigma$ for the cubic Landau-Ginzburg model is parametrized by the equation

$$\Sigma : \ \phi^2 = \mathbf{z}, \tag{81}$$

which is a double covering of $\mathbb{C}$ branched over $\mathbf{z} = 0$. This is illustrated in Figure 6. The value of the superpotential in the two vacua $\phi = \pm\sqrt{\mathbf{z}}$ of the two-dimensional theory $\mathcal{T}_{\mathbf{z}}$ is given by

$$W(\phi) = \mp\frac{2}{3}\mathbf{z}^{3/2}. \tag{82}$$

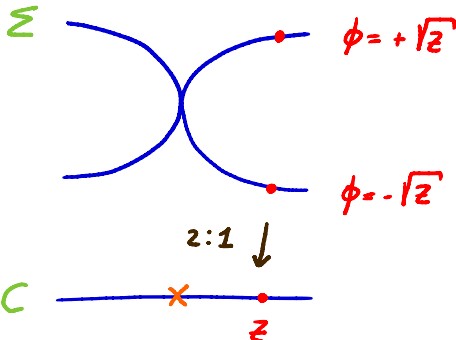

Figure 6: Spectral curve $\Sigma$ for the cubic Landau-Ginzburg theory. For every $\mathbf{z} \neq 0$ there are two vacua with $\phi = \pm\sqrt{\mathbf{z}}$.

The reason for naming the branched cover $\Sigma$ the spectral curve is that $\Sigma$ may be realised as the spectral curve associated to a (possibly higher-dimensional) **Higgs bundle** $(\mathcal{E}, \varphi)$ over $C$ [20]. To see this, define the Higgs field

$$\varphi : \ TC \to \operatorname{End}\mathcal{E} \tag{83}$$

through

$$(\varphi(v))(w) = q(v) \cdot w. \tag{84}$$

The spectral curve $\Sigma$ may then equivalently be defined through the characteristic equation for $\varphi$. If the holomorphic map $q$ defines a bijection between $TC$ and $\mathcal{E}$, we may embed the spectral curve $\Sigma$ in $T^*C$. In that case there is a natural meromorphic 1-form on $\Sigma$, defined by the restriction of the tautological 1-form on $T^*C$. We will work this out in detail in the next subsection §2.4.1 for the class of Landau-Ginzburg models.

### 2.4.1 Example: spectral geometry for LG models

Here we spell out the embedding of $\Sigma$ as a spectral curve in $T^*C$ for the Landau-Ginzburg model with a single chiral field $\phi$ valued in $\mathbb{C}$.

The starting data is the generic superpotential

$$W(\phi, \mathbf{z}^{(1)}, \ldots, \mathbf{z}^{(k-2)}) = \frac{\phi^k}{k} + \mathbf{z}^{(k-2)} \frac{\phi^{k-2}}{k-2} + \ldots + \mathbf{z}^{(1)}\phi \tag{85}$$

of degree $k$, so that the spectral curve $\Sigma$ is defined by the equation

$$\partial_\phi W = \phi^{k-1} + \mathbf{z}^{(k-2)} \phi^{k-3} + \ldots + \mathbf{z}^{(1)} = 0. \tag{86}$$

To find out how $\Sigma$ is embedded in $T^*C$, note that the $\partial_{\mathbf{z}^{(l)}}W = \frac{\phi^l}{l}$ are linearly independent in the chiral ring $\mathbb{C}[\phi]/\langle\partial_\phi W\rangle$ and generate the chiral ring together with the identity. Define $C = \oplus_l \mathbb{C}_{\mathbf{z}^{(l)}}$ and the bundle of algebras $\mathcal{E} \to C$ with fibre $\mathbb{C}[\phi]/\langle\partial_\phi W\rangle$. We have the map

$$\begin{aligned} q : TC &\to \mathcal{E} \\ \partial_{\mathbf{z}^{(l)}} &\mapsto \partial_{\mathbf{z}^{(l)}}W \\ v &\mapsto d_C W(v) \end{aligned} \tag{87}$$

Now consider $T^*C$ as a complex manifold with holomorphic Darboux coordinates $(\mathbf{z}^{(l)}, q_l)$, where

$$q_l = q(\partial_{\mathbf{z}^{(l)}}). \tag{88}$$

Then we can define embed $\Sigma \hookrightarrow T^*C$ in two (equivalent) ways.

Identifying $\partial_{\mathbf{z}^{(l)}}W$ with $q_l$, which we can do as the map $q$ is an isomorphism, there is a canonical lift of $W$ (the highest power is taken to be $\frac{q_1^k}{k}$) to $\mathbb{C}[\mathbf{z}^{(l)}, q_l]$ which we can consider as a holomorphic function $\overline{W} \in \mathcal{O}(T^*C)$. Consider the Liouville one-form

$$\lambda = \sum_l q_l \, d\mathbf{z}^{(l)} \in \Omega^1(T^*C), \tag{89}$$

where $d$ is the exterior derivative on $T^*C$. Then

$$d\overline{W} - \lambda = \sum_l \left( \partial_{\mathbf{z}^{(l)}}\overline{W} \, d\mathbf{z}^{(l)} + \partial_{q_l}\overline{W} \, dq_l - q_l d\mathbf{z}^{(l)} \right) = \sum_l \partial_{q_l}\overline{W} \, dq_l. \tag{90}$$

If we now restrict to the $((k-2)+1)$-dimensional submanifold

$$N_\phi \subset T^*C, \tag{91}$$

defined by reintroducing the $\phi$ dependence to the $q_l$, we find that

$$\sum_l \partial_{q_l}\overline{W} dq_l|_{N_\phi} = \sum_l \frac{\partial\phi}{\partial q_l}\frac{\partial q_l}{\partial\phi} \partial_\phi W \, d\phi = \partial_\phi W d\phi. \tag{92}$$

Note that $N_\phi$ is equivalent to a one-parameter family of sections of $T^*C$, and what we have really done here is embed a copy of $\mathbb{C}_\phi$ into each fibre of $T^*C$. This short procedure shows that we can write

$$\Sigma = \{p \in N_\phi \mid d\overline{W} - \lambda = 0\} \subset T^*C. \tag{93}$$

Alternatively, we can write the spectral curve as a Higgs bundle spectral curve, although this is a little bit more subtle. To see why, consider

$$\{\mathbf{z} \in C, \eta \in T_{\mathbf{z}}^*C \mid \det(\varphi(\mathbf{z}) - \eta \cdot \mathrm{id}) = 0\} \tag{94}$$

Take $\eta = \eta_l d\mathbf{z}^{(l)}$, identifying $\eta_l$ with $\partial_{\mathbf{z}^{(l)}} \overline{W}$, we get

$$\left\{\mathbf{z} \in C, \eta \in T_{\mathbf{z}}^*C \mid \forall l: \ \det(\partial_{\mathbf{z}^{(l)}} W \cdot_E -\partial_{\mathbf{z}^{(l)}} \overline{W} \, \mathrm{id}) = 0\right\}. \tag{95}$$

We see that the determinant can only be non-zero via what has been quotiented out of $E_z$ (i.e. the chiral ring relations), which are precisely $\partial_\phi W$. Hence

$$\Sigma = \{z \in C, \eta \in T_z^*C \mid \det(\varphi(z) - \eta \cdot id) = 0\} \subset T^*C. \tag{96}$$

As an example, consider the spectral curve $\Sigma$ for the quartic LG model with superpotential $W = \frac{\phi^4}{4} + \mathbf{z}^{(2)}\frac{\phi^2}{2} + \mathbf{z}^{(1)}\phi$, which is defined by the equation

$$\Sigma: \ \phi^3 + \mathbf{z}^{(2)}\phi + \mathbf{z}^{(1)} = 0, \tag{97}$$

and may be embedded in $T^*C$ with tautological form

$$\lambda = \frac{\phi^2}{2} \, d\mathbf{z}^{(2)} + \phi \, d\mathbf{z}^{(1)}. \tag{98}$$

The spectral curve $\Sigma$ can then also be rephrased as the characteristic equation for the Higgs field $\varphi$. Indeed we can check that

$$\det(\partial_{\mathbf{z}^{(1)}} W \cdot_{\mathcal{E}} +\partial_{\mathbf{z}^{(1)}} \tilde{W}) \ \sim \ \det(\phi \cdot_{\mathcal{E}} -\phi \, \mathrm{id}) \ \sim \ \partial_\phi W, \tag{99}$$
$$\det(\partial_{\mathbf{z}^{(2)}} W \cdot_{\mathcal{E}} -\partial_{\mathbf{z}^{(2)}} \tilde{W}) \ \sim \ \det(\phi^2 \cdot_{\mathcal{E}} -\phi^2 \, \mathrm{id}) \ \sim \ (\partial_\phi W)^2.$$

### 2.4.2 Remark: relation to Seiberg-Witten geometry

Spectral curves and Higgs fields may sound familiar from Andy Neitzke's lectures last week. Remember though that, in this purely two-dimensional setting, the space $C$ is the moduli space parametrizing deformations of the 2d theory $\mathcal{T}_{\mathbf{z}}$.

## 3 BPS solitons and spectral networks

In this section we study BPS states in the low energy description of 2d $\mathcal{N} = (2,2)$ theories. We introduce BPS solitons in 2d Landau-Ginzburg models and show how they are encoded in the structure of a spectral network. We start with a motivation to find out more about the low-energy description of LG models.

## 3.1 BPS solitons and Morse flow

You may be worried that there is an issue in the LG model. Indeed, remember the (brief) discussion of supersymmetric quantum mechanics (SQM) in Andy's lectures in the Les Houches school (see also [20]), whose Euclidean Lagrangian contains the bosonic terms

$$\mathcal{L}_{\text{bos}}^{\text{SQM}} = \frac{1}{2}\dot{q}^2 + \frac{1}{2}\frac{dh}{dq}^2 , \tag{100}$$

in terms of a particle $q(\tau)$ moving on a compact Riemannian manifold $M$ and a real Morse function $h : M \to \mathbb{R}$. The SQM supercharge $Q$ is conjugate to the exterior derivative on $M$, and as a consequence the vacuum structure of the SQM should be independent on the choice of $h$. However, the number of critical points of $h$ is clearly dependent on the choice of $h$.

The resolution is that not all critical points of $h$ are necessarily exact vacua: there may be non-perturbative contributions that lift the vacuum energy. These non-perturbative contributions are parameterized by field configurations that describe the tunnelling between the critical points of $h$. Such corrections are called BPS instantons. The vacuum structure is then governed by the so-called **Morse-Smale-Witten (MSW) complex**, whose basis is given by the critical points of $h$ (i.e. the perturbative vacua), and whose differential $Q^\xi$ counts (with signs) the number of instantons between the perturbative vacua. The true vacua of the SQM are determined by the cohomology of the MSW complex, and these are indeed independent on the choice of $h$ [34].

Since a Landau-Ginzburg model can be dimensionally reduced to a SQM, for instance by taking all fields to be constant in time and considering the usual spatial coordinate $\sigma$ as the new "time", a similar story holds.[15] Let us therefore find out how to describe the corresponding **BPS solitons** in the Landau-Ginzburg model.

Suppose that we consider the Landau-Ginzburg model on an interval $I_\sigma \times \mathbb{R}_\tau$ with space-like coordinate $\sigma$ and **Euclidean** time $\tau$. Then we need to specify a boundary condition at the ends of the interval $I_\sigma$. Suppose that the fields $\phi^i(\sigma)$ approach the vacuum value $\phi_\alpha^i$ on one end and $\phi_\beta^i$ on the other. The energy of such a tunnelling field configuration is given by

$$E_{\alpha\beta} = \int_{I_\sigma} d\sigma \left( g_{i\bar{j}}\frac{d\phi^i}{d\sigma}\frac{d\phi^{\bar{j}}}{d\sigma} + \frac{1}{4}\partial_i W \overline{\partial_j W} \right) , \tag{101}$$

and has a lower bound given by [36][16]

$$E_{\alpha\beta} \geq |W(\beta) - W(\alpha)|. \tag{102}$$

---

[15]To be precise, the resulting SQM has target $M = X$, superpotential $h = \text{Re}W$, and double the amount of supersymmetry, since $X$ is Kähler. A finer approach would be to rewrite the Landau-Ginzburg model as a SQM into the space of maps $I_\sigma \to X$. In this setup the space of critical points is the set of BPS solitons, whereas the MSW differential is determined by solutions to the so-called $\zeta$-instanton equation (see for instance [35]) We will say more about the latter equation in §5.9.

[16]This bound is exact quantum-mechanically, since the holomorphic superpotential $W$ does not receive any quantum corrections [30].

Only the field configurations that minimize the energy $E_{\alpha\beta}$ are stable against deformations. They are called BPS solitons and satisfy the PDE

$$\frac{d\phi^i}{d\sigma} = \frac{i\zeta}{2} g^{i\bar{j}} \overline{\partial_j W}, \tag{103}$$

with

$$i\zeta = \frac{W(\beta) - W(\alpha)}{|W(\beta) - W(\alpha)|}. \tag{104}$$

Equation (103) is called the $\zeta$**-soliton equation**. Since the superpotential $W$ is a holomorphic function, the expected dimension of the reduced[17] moduli space of its solutions is -1 and therefore generically empty. This implies that there is generically a discrete set of phases $\zeta_{\alpha\beta}$ for which a solution exists.

Any solution to the soliton equation (103) corresponds to a BPS soliton with central charge

$$Z_{\alpha\beta} = W(\beta) - W(\alpha). \tag{105}$$

Indeed, since

$$E_{\alpha\beta} = (i\zeta)^{-1} Z_{\alpha\beta} = \mathrm{Im}(\zeta^{-1} Z_{\alpha\beta}), \tag{106}$$

it follows from equation (12) that the soliton preserves the Euclidean $\mathcal{N} = 2$ subalgebra generated by $Q_A^\xi$ and $\overline{Q}_A^\xi$, with

$$\xi = -\zeta^{-1} \tag{107}$$

Note that the $\zeta$-soliton equation (103) implies that

$$\partial_\sigma W = \frac{i\zeta}{2} g_{i\bar{j}} \partial_i W \overline{\partial_j W} \in i\zeta \mathbb{R}_{\geq 0}, \tag{108}$$

so that the BPS soliton corresponds in the $W$-plane to a straight line between the vacua $W(\alpha)$ and $W(\beta)$ with angle $\arg(\zeta)$, and so that the quantity

$$\mathbf{H} = -\mathrm{Re}(\zeta^{-1} W) \tag{109}$$

is constant along the soliton trajectory. The soliton equation thus has the interpretation as a Hamiltonian flow equation with respect to the Hamiltonian $\mathbf{H}$.

The soliton equation may also be interpreted as an upward flow equation with respect to the **Morse function**

$$\mathbf{h} = \mathrm{Im}(\zeta^{-1} W). \tag{110}$$

Indeed, the Morse function $\mathbf{h}$ is increasing along the Morse flow in $X$ defined by

$$\frac{d\phi^a}{d\sigma} = g^{ab} \frac{d\mathbf{h}}{d\phi^b}. \tag{111}$$

The equivalence of the Hamiltonian and the Morse flow equations follows from the Cauchy-Riemann equations for the holomorphic function $\zeta^{-1} W$.

---

[17]This means that we consider two solutions equivalent if they are equivalent under a symmetry transformation, such as a translation.

### 3.1.1 Lefschetz thimbles

The Morse flow generated by $\mathbf{h} = \text{Im}(\zeta^{-1}W)$ encodes all potential solutions to the $\zeta$-soliton equation (103). The union in $X$ of all such solutions with left (or right) boundary condition given by

$$\lim_{\sigma \to -\infty} \phi^i(\sigma) = \phi^i_\alpha, \text{ or} \tag{112}$$

$$\lim_{\sigma \to \infty} \phi^i(\sigma) = \phi^i_\alpha, \tag{113}$$

form a real, middle-dimensional, Lagrangian submanifold of $X$, that is known as a left (or right) **Lefschetz thimble** $J^\zeta_{\alpha,L}$ (or $J^\zeta_{\alpha,R}$) [37]. Note that

$$J^\zeta_{\alpha,R} = J^{-\zeta}_{\alpha,L}. \tag{114}$$

The Lefschetz thimbles $J^\zeta_{\alpha,L}$ define cycles in the homology of $X$ with boundary in the region $B$ where $\text{Im}(\zeta^{-1}W)$ is sufficiently large. They are also called wave-front trajectories in [38].

The BPS solitons with central charge $Z_{\alpha\beta}$ must clearly be part of the overlap

$$J^\zeta_{\alpha,L} \cap J^\zeta_{\beta,R} \tag{115}$$

between a left and a right Lefschetz thimble with $\zeta = \arg Z_{\alpha\beta}$. On the W-plane, the Lefschetz thimbles $J^{\zeta_{\alpha\beta}}_{\alpha,L}$ and $J^{-\zeta_{\alpha\beta}}_{\beta,L}$ project to straight half-lines with angles

$$\pm i\zeta_{\alpha\beta} = \pm \frac{Z_{\alpha\beta}}{|Z_{\alpha\beta}|}, \tag{116}$$

respectively. The intersection (115) thus projects to an interval in the $W$-plane. We will see an example of this in §3.1.2.

It is often convenient to instead think of the Lefschetz thimbles $J^\zeta_{\alpha,L}$ and $J^\zeta_{\alpha,R}$ as intersecting transversally. This may be achieved by slightly rotating the phase $\zeta$ and considering the intersection

$$J^{\zeta e^{i\epsilon}}_{\alpha,L} \cap J^{\zeta e^{-i\epsilon}}_{\beta,R}. \tag{117}$$

### 3.1.2 Example: Morse flow in the cubic Landau-Ginzburg model

Consider the cubic LG model as an example, at $\mathbf{z} = 1$, so that $X = \mathbb{C}$ and $W(x) = \frac{1}{3}x^3 - x$. This means that there can be (and in fact are) two solitonic solutions between the vacua at $x_\pm = \pm 1$ with central charge $Z = \mp\frac{2}{3}$ and $\zeta = \mp i$.

The Lefschetz thimbles $J^\zeta_{\pm,L}$ are submanifolds in $X = \mathbb{C}$, containing the critical points $x = \pm 1$, such that the Hamiltonian

$$\mathbf{H}(x) = -\text{Re}(\zeta^{-1}W(x)) \tag{118}$$

is constant, while the Morse function

$$\mathbf{h}(x) = -\text{Im}(\zeta^{-1}W(x)) \tag{119}$$

is decreasing in the direction of the flow. It is a good exercise to plot the Lefschetz thimbles $J_{\pm}^{\zeta}$ numerically, and check that they are disjoint for generic $\zeta$, but overlap precisely when $\zeta = \pm i$. Indeed, since $\mathbf{H}(x)$ vanishes along the real axis for $\zeta = \pm i$, whereas $\mathbf{h}(x)$ decreases/increases along the interval $[-1, 1]$ for $\zeta = \pm i$, it is clear that the Lefschetz thimbles $J_{+,L}^{\zeta=\pm 1}$ and $J_{-,R}^{\zeta=\pm 1}$ overlap on this interval. The interval $[-1, 1] \subset \mathbb{C}$ (with the two possible orientations) thus represents the two BPS solitons with central charges $Z = \pm\frac{2}{3}$. The thimbles for $\zeta = i$ are illustrated in Figure 7.

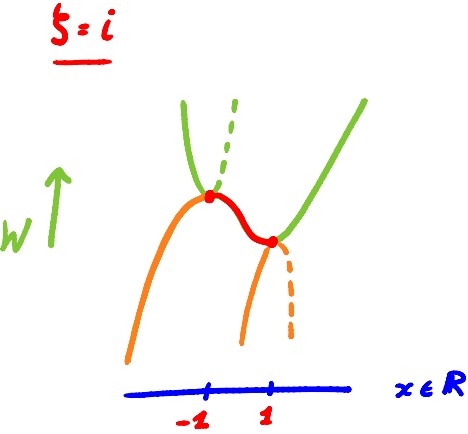

Figure 7: Lefschetz thimbles $J_{\pm}^{\zeta=i}$ in the cubic Landau-Ginzburg model. The thimbles are coloured in orange and red, with the red component corresponding to the BPS soliton of charge $Z = -2/3$.

## 3.2 BPS index and spectral networks

We will soon find that BPS solitons are encoded in a mathematical structure, called a spectral network, on the parameter space of the 2d theory $T_{\mathbf{z}}$. More precisely, the spectral network keeps track of a 2d BPS index counting (with signs) the BPS solitons.

### 3.2.1 BPS index and vanishing cycles

The number of BPS solitons that interpolate between the vacua $\alpha$ and $\beta$ is counted (with signs) by the **2d BPS index**

$$\mu_{\alpha\beta} = \text{Tr}(-1)^F e^{\beta\left(\mathbf{H}+\text{Re}(\zeta^{-1}Z_{\alpha\beta})\right)}. \tag{120}$$

Note that this index only receives contributions from the solutions to the $\zeta$-soliton equation (103), since for those solutions $\mathbf{H} + \text{Re}(\zeta^{-1}Z_{\alpha\beta}) = 0$. Geometrically, the 2d

BPS index (120) computes an intersection number between the so-called **vanishing cycles** $\Delta_\alpha$ in the pre-image of $W$ [36].

These vanishing cycles are constructed as follows. Choose a vacuum $\alpha$ and consider an arbitrary point $w$ as well as the point $W(\alpha)$ in the $W-$plane. Draw a straight line between these two points with angle

$$i\zeta = \frac{w - W(\alpha)}{|w - W(\alpha)|}. \tag{121}$$

The vanishing cycle $\Delta_\alpha$ is defined as the real, middle-dimensional, homology cycle of $W^{-1}(w)$ which consists of those points in $W^{-1}(w)$ that can be reached via solutions to the $\zeta$-soliton equation (103) that originate from the vacuum configuration $\phi_\alpha^i$. They are called vanishing cycles because they shrink to a point in the limit that $w \to W(\alpha)$. It turns out that the collection of vanishing cycles $\{\Delta_\alpha\}_\alpha$ forms a basis of the middle-dimensional homology of $W^{-1}(w)$ [37].

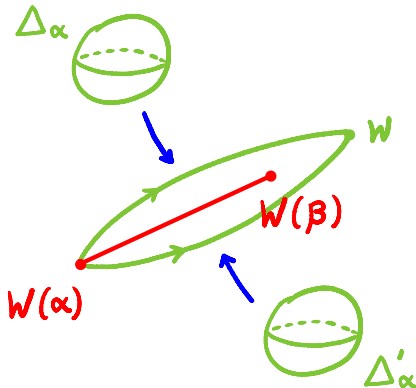

Figure 8: Illustration of the vanishing cycle $\Delta_\alpha$ and its change in homology when the path from $W(\alpha)$ to $w$ crosses another critical point $W(\beta)$.

The cycles $\Delta_\alpha^\zeta$ are defined with respect to a straight of path connecting $W(\alpha)$ to $w$, yet invariant under slight deformations of the path, as long as the new path is homotopic to the straight line in the $W$-plane minus the critical points. If, however, the path crosses a critical point $W(\beta)$, the vanishing cycle $\Delta_\alpha^\zeta$ picks up a contribution

$$\Delta_\alpha \mapsto \Delta_\alpha' = \Delta_\alpha \pm (\Delta_\alpha \circ \Delta_\beta) \Delta_\beta, \tag{122}$$

where $\Delta_\alpha \circ \Delta_\beta$ is the intersection number of the cycles $\Delta_\alpha$ and $\Delta_\beta$, and the $\pm$-sign depends on certain orientations. See Figure 8. This may be familiar to you from **Picard-Lefschetz theory**. Note that the path between $W(\alpha)$ and $w$ crossing the critical point $W(\beta)$ is equivalent to $\zeta$ crossing the value $\zeta_{\alpha\beta}$.

Since each intersection between the cycles $\Delta_\alpha$ and $\Delta_\beta$ corresponds to a BPS soliton that tunnels between the vacua $\alpha$ and $\beta$, we are led to the identification

$$\mu_{\alpha\beta} = \Delta_\alpha \circ \Delta_\beta, \tag{123}$$

of the 2d BPS index $\mu_{\alpha\beta}$ with the intersection number $\Delta_\alpha \circ \Delta_\beta$.

Instead of studying the BPS solitons through the vanishing cycles, we may also consider the collection of real, middle-dimensional, cycles in $X$ that are swept out by moving the vanishing cycles $\Delta_\alpha$ along the half-line with phase $i\zeta$, starting from the critical point $W(\alpha)$. These are precisely the left **Lefschetz thimbles** $J_\alpha^\zeta$, defined around equation (112), that can also be obtained through the upward flow with respect to the Morse function

$$\mathbf{h} = \mathrm{Im}(\zeta^{-1}W). \tag{124}$$

Their homology classes $[J_\alpha^\zeta]$ are known to form a complete basis for the middle-dimensional homology of $X$ with boundary in the region $B \subset X$ where $\mathrm{Im}(\zeta^{-1}W)$ is sufficiently large [37].

If we rotate $\zeta$ it may happen that we encounter critical values $\zeta_{\alpha\beta}$ such that there exist BPS solitons connecting the perturbative vacua labeled by $\alpha$ and $\beta$. In that situation, a topology change occurs amongst the Lefschetz thimbles, in which the homology class $[J_\beta^\zeta]$ stays invariant, but the homology class $[J_\alpha^\zeta]$ picks up a contribution

$$[J_\alpha^{\zeta'}] = [J_\alpha^\zeta] \pm \mu_{\alpha\beta}[J_\beta^\zeta]. \tag{125}$$

This is the equivalent of the Picard-Lefschetz transformation (122) for the vanishing cycles and illustrated in Figure 9. (Note that in the case that $X$ is complex one-dimensional, the vanishing cycles $\Delta_\alpha^\zeta$ consist of two points for $w \neq W(\alpha)$, so that the Lefschetz thimble $J_\alpha^\zeta$ looks like an infinitely long bell.)

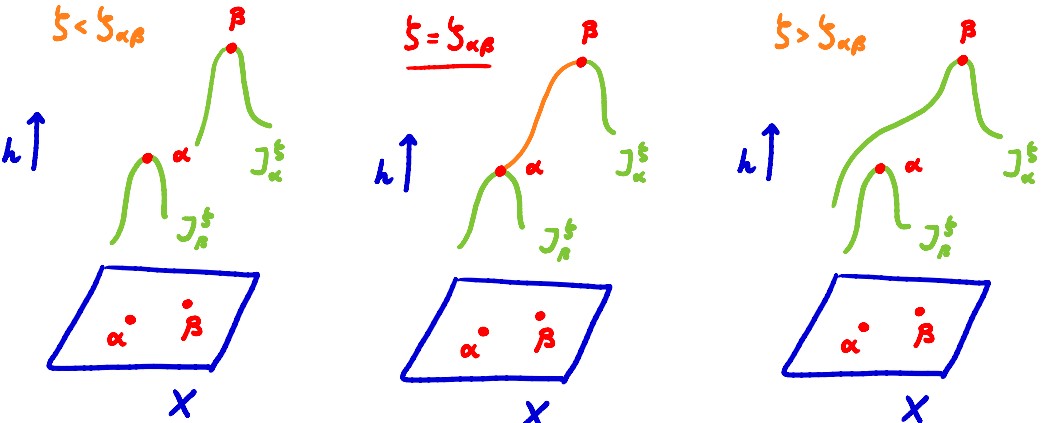

Figure 9: Topology change in Lefschetz thimbles when crossing the critical value $\zeta_{\alpha\beta}$. At this critical phase the Lefschetz thimble $J_\alpha$ contains a component (highlighted in red) that connects the critical values $\alpha$ and $\beta$. Across the critical phase the Lefschetz thimble $J_\alpha$ picks up a contribution proportional to $J_\beta$.

Equation (125) might remind you of the **Stokes phenomenon**, which we return to in §5. As we will see in §3.3, it is also the key element in the geometric formulation of the Cecotti-Vafa wall-crossing formula.

### 3.2.2 BPS solitons and spectral networks

Let us return to a generic massive 2d $\mathcal{N} = (2,2)$ theory $\mathcal{T}_{\mathbf{z}}$ with non-zero central charge $Z$.[18] Then we can define **BPS solitons** tunnelling between the vacua $\alpha$ and $\beta$ as solutions to the BPS bound

$$Z_{\alpha\beta} = \zeta_{\alpha\beta}\, E_{\alpha\beta} \tag{126}$$

for some phase $\zeta_{\alpha\beta}$. Just as before, these BPS states are invariant under the A-type $\mathcal{N} = 2$ subalgebra.

Now consider the spectrum of BPS solitons of the 2d theory $\mathcal{T}_{\mathbf{z}}$ as we move along its deformation space $C_{\mathbf{z}}$. This spectrum contains BPS solitons tunnelling between the vacua $\alpha$ and $\beta$ if and only if $\mathbf{z} \in C$ lies on a path $\mathbf{w}_{\alpha\beta}(t) \subset C$ such that

$$\mathrm{Im}\left(\zeta_{\alpha\beta}^{-1}\, Z(\mathbf{w}_{\alpha\beta}(t))\right) = 0. \tag{127}$$

That is, the parameter $\mathbf{z}$ should be part of a path $\mathbf{w}_{\alpha\beta}(t)$ along which the central charge function $Z$ has a constant phase $\zeta_{\alpha\beta}$. Equivalently, the path $\mathbf{w}_{\alpha\beta}(t)$ should solve the first-order ODE

$$\frac{dZ(\mathbf{w}_{\alpha\beta}(t))}{dt} \in \zeta_{\alpha\beta}\, \mathbb{R}_{\geq 0}. \tag{128}$$

The trajectories $\mathbf{w}_{\alpha\beta}(t)$ may be oriented in the direction in which $|Z_{\alpha\beta}|$ increases. They may either start at a point in $C$ where $Z_{\alpha\beta} = 0$, or be "born" at the intersection of some other trajectories $\mathbf{w}_{\alpha'\beta'}(t)$ at the same phase $\zeta_{\alpha'\beta'} = \zeta_{\alpha\beta}$. We will see examples of either type of trajectories in Figures 10 and 11.

Solving the first-order ODE (128) proves an efficient way of plotting the trajectories. This was first done in [1], resulting in a beautiful paper with lots of cool pictures. The collection of all BPS trajectories $\mathbf{w}_{\alpha\beta}(t)$ for a given phase $\zeta = e^{i\vartheta}$, but for any two vacua $\alpha$ and $\beta$, is called the **spectral network** $\mathcal{W}_{\vartheta}$.

As a simple example of a spectral network, consider the **cubic Landau-Ginzburg model** with spectral curve

$$\Sigma: \ x^2 = \mathbf{z}. \tag{129}$$

Fix the phase $\vartheta$. Then the 2d theory $\mathcal{T}_{\mathbf{z}}$ admits a 2d BPS state with central charge $Z_{\alpha\beta}$ such that

$$\arg(Z_{\alpha\beta}) = \vartheta \tag{130}$$

for every $\mathbf{z} \in \mathbb{C}$ such that

$$W(\beta) - W(\alpha) = \pm\frac{4}{3}\mathbf{z}^{3/2} \tag{131}$$

---

[18]If the $\mathcal{N} = (2,2)$ theory admits a non-zero twisted central charge $\widetilde{Z}$ instead – and remember that these two options are mutually exclusive, we can define BPS solitons similarly in terms of $\widetilde{Z}$. They will be invariant instead under the B-type $\mathcal{N} = 2$ subalgebra. We will come back to examples of this kind in §4.

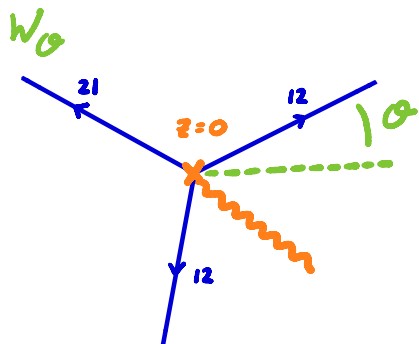

Figure 10: Spectral network $\mathcal{W}_\vartheta$ on $C = \mathbb{C}_{\mathbf{z}}$ that encodes BPS solitons in the cubic Landau-Ginzburg model. Each trajectory of the spectral network (in blue) is oriented away from the branch-point (at $\mathbf{z} = 0$) and labeled by a pair $\alpha\beta$ of vacua. The trajectory corresponds to the collection of theories $\mathcal{T}_{\mathbf{z}}$ that admit a BPS soliton tunneling from vacuum $\alpha$ to $\beta$ with central charge $Z_{\alpha\beta}$ such that $\arg Z_{\alpha\beta} = \vartheta$.

has phase $\zeta = e^{i\vartheta}$. This constraint determines the three blue trajectories in Figure 10.

If we orient the trajectories in the direction in which $|Z_{\alpha\beta}|$ increases, the three trajectories originate from the branch-point $\mathbf{z} = 0$, indicated by the orange cross in Figure 10. If we furthermore choose a trivialization of $\Sigma$, i.e. a choice of vacuum 1 and 2 across the parameter space $C$, by choosing a branch-cut on $C$, we may label the trajectories by 12 or 21 depending on whether the associated BPS soliton tunnels from vacuum 1 to 2 or from vacuum 2 to 1.

So suppose we fix a point $\mathbf{z} \in C$ corresponding to a 2d theory $\mathcal{T}_{\mathbf{z}}$. Then we can vary $\vartheta$ and check whether for which values of $\vartheta$ there may be BPS trajectories that run across the point $\mathbf{z}$. If there is such a trajectory with label $\alpha\beta$ for a certain phase $\vartheta_{\alpha\beta}$, then we know that the theory $\mathcal{T}_{\mathbf{z}}$ admits a BPS soliton tunneling from vacuum $\alpha$ to $\beta$ with $\arg Z_{\alpha\beta} = \vartheta_{\alpha\beta}$. By varying $\vartheta$ from 0 to $2\pi$ we thus find all the BPS solitons in the theory.

Note that the spectral network can be defined in purely geometric terms. Say that we are given the spectral geometry $\Sigma \subset T_x^* C_{\mathbf{z}}$ with tautological 1-form $\lambda = x\,d\mathbf{z}$. Given any choice of trivialization of the covering $\Sigma$, the $(\alpha\beta)$-trajectories of the spectral network $W_\vartheta$ are parametrized by all paths $p(t)$ on $C$ for which

$$(\lambda_\alpha - \lambda_\beta)(v) \in e^{i\vartheta}\mathbb{R}, \tag{132}$$

for any tangent vector $v$ to $p(t)$.

The tautological 1-form $\lambda$, when restricted to $\Sigma$, can be expressed in terms of the invariants of the Higgs field $\varphi$. In the case that the covering $\Sigma \to C$ is of degree 2, such as for the cubic Landau-Ginzburg model, the trace of $\varphi^2$ determines a quadratic differential $\phi_2$ on $C$. We then have

$$\lambda = \sqrt{\phi_2}. \tag{133}$$

The fact that trajectories of the corresponding spectral network do not intersect each

other, is geometrically because the spectral network $\mathcal{W}_\vartheta$ is the collection of singular leaves of a foliation of the quadratic differential $\phi_2$ with phase $\vartheta$.

Spectral networks of degree $> 2$ can get pretty complicated. For instance, Figure 11 illustrates a spectral network of degree 3. Such networks may have trajectories that intersect each other, and trajectories that start at the points of intersection. Any spectral network of degree 2, on the other hand, looks locally like a cubic Landau-Ginzburg network, for generic phase $\vartheta$. It may happen at special phases that two trajectories with opposite orientations, as well as opposite labels, come together. The resulting trajectory is sometimes called a **saddle trajectory** or a double trajectory. We will see examples of such trajectories in §4.

Here we just note that saddle trajectories cannot appear in Landau-Ginzburg models. Indeed, a saddle would indicate the presence of two BPS solitons, one mapping to a straight line from $W(\alpha)$ to $W(\beta)$ in the $W$-plane, and the other to a straight line from $W(\beta)$ to $W(\alpha)$, but both with the same angle $\vartheta$. This clearly implies that $W(\alpha) = W(\beta)$.

## 3.3 BPS wall-crossing

Whereas the 2d BPS spectrum stays invariant under small deformations, as Andy already discussed in his lecture about BPS states, there are real codimension-1 loci on the parameter space $C$, where

$$Z_{\alpha\beta} + Z_{\beta\gamma} = Z_{\alpha\gamma}. \tag{134}$$

At such a locus the 2d BPS states with central charge $Z_{\alpha\beta}$ and $Z_{\beta\gamma}$ may form a 2d BPS bound state with central charge $Z_{\alpha\gamma}$. This locus is called a two-dimensional wall of marginal stability. Instances of 2d **wall-crossing** can be conveniently read off from the spectral network $\mathcal{W}_\vartheta$.

Before we explain this, note that to see 2d wall-crossing we need to have at least three vacua in the 2d $\mathcal{N} = (2, 2)$ theory. Equivalently, the degree of the covering $\Sigma \to C$ should be at least three. The 2d wall-crossing then appears when two BPS trajectories labeled by $\alpha\beta$ and $\beta\gamma$ intersect each other. At such an intersection a new BPS trajectory with label $\alpha\gamma$ may emerge, which corresponds to the new 2d BPS state with label $\alpha\gamma$. In that sense, 2d-wall-crossing is literally the crossing of trajectories in the spectral network $\mathcal{W}_\vartheta$.

An example is given by the Landau-Ginzburg model with quartic superpotential

$$W(\phi) = \frac{1}{4}\phi^4 - \frac{1}{2}\mathbf{z}^{(2)}\phi^2 - \mathbf{z}^{(1)}\phi, \tag{135}$$

whose chiral ring is the Jacobian ring

$$E_z = \mathbb{C}[\phi]/\langle\phi^3 - \mathbf{z}^{(2)}\phi - \mathbf{z}^{(1)} = 0\rangle, \tag{136}$$

and whose spectral network $\mathcal{W}_\vartheta$ at $\vartheta = \pi/2$ is illustrated in Figure 11, when $\mathbf{z}^{(2)} = -1$ is held fixed.

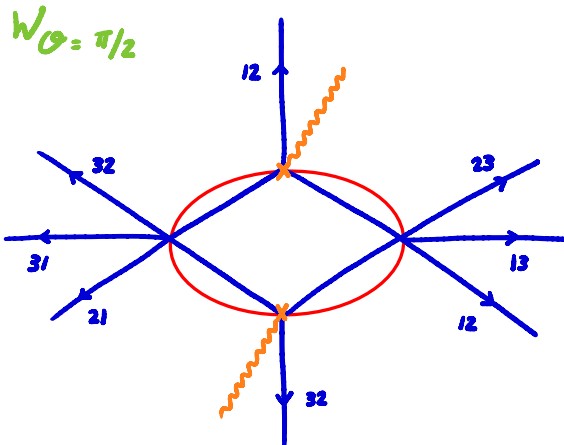

Figure 11: Spectral network encoding the 2d BPS solitons in the quartic Landau-Ginzburg model.

The **Cecotti-Vafa wall-crossing formula** can be obtained by going around any intersection point of BPS trajectories in a small loop. Suppose we decorate this loop with two marked points on its intersection with the 2d wall of marginal stability, and orient both intervals between the two marked points towards one of the marked points. See Figure 12.

To each intersection of a BPS trajectory with the small loop, we associate the corresponding transformation (125) of the vector of Lefschetz thimbles $(J_\alpha)_\alpha$. Composing the transformations when going around the half-loop either way, and imposing that the resulting transformations are equal, gives the Cecotti-Vafa wall-crossing formula

$$\mu'(\alpha, \beta) = \mu(\alpha, \beta)$$
$$\mu'(\beta, \gamma) = \mu(\beta, \gamma) \tag{137}$$
$$\mu'(\alpha, \gamma) = \mu(\alpha, \gamma) \pm \mu(\alpha, \beta)\mu(\beta, \gamma).$$

The Cecotti-Vafa wall-crossing formula was rederived in this way in [39] (see also [20]).

## 3.4 Open special Lagrangian discs

So far, we have found that the spectral network $\mathcal{W}_\vartheta$ encodes 2d BPS states with

$$\arg(Z_{\alpha\beta}) = \vartheta \tag{138}$$

as $\alpha\beta$-trajectories. Let us consider a simple $\alpha\beta$-trajectory that starts at a branch-point of the covering $\Sigma \to C$. This $\alpha\beta$-trajectory may be lifted to an open path $\gamma_{\alpha\beta}(\mathbf{z}) \subset \Sigma$ connecting the pre-images $x_\alpha(\mathbf{z})$ and $x_\beta(\mathbf{z})$. We refer to the open path $\gamma_{\alpha\beta}(\mathbf{z})$ as the **detour path**. It is found by starting at the pre-image $x_\alpha(\mathbf{z})$, following the lift of the $\alpha\beta$-trajectory to the $\alpha$th sheet backwards to the branch-point, going around the

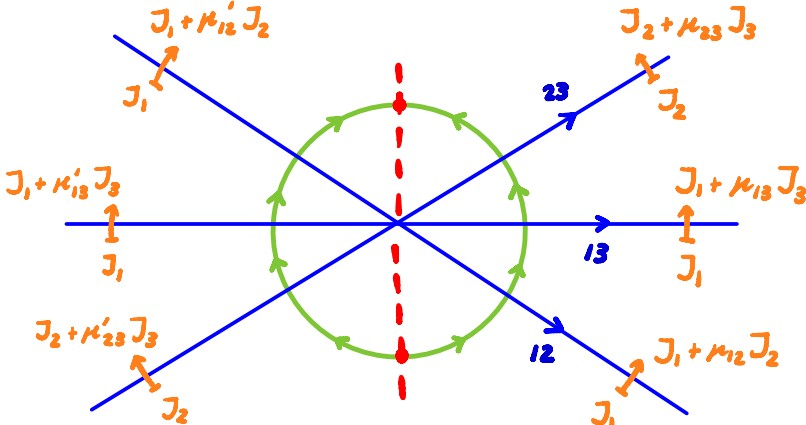

Figure 12: Obtaining the Cecotti-Vafa wall-crossing formula from a spectral network. Across each trajectory we have only written down the non-trivial Lefschetz transformation.

branch-point, and returning to the pre-image $x_\beta(\mathbf{z})$ along the lift of the $\alpha\beta$-trajectory to the $\beta$th sheet.

If the $\alpha\beta$-trajectory starts at an intersection of other trajectories instead, we will need to continue tracing these trajectories backwards until we reach the branch-points they originate from. Examples are shown in Figure 13. The collection of trajectories corresponding to a single detour path $\gamma_{\alpha\beta}(\mathbf{z})$ is called a **BPS web**.

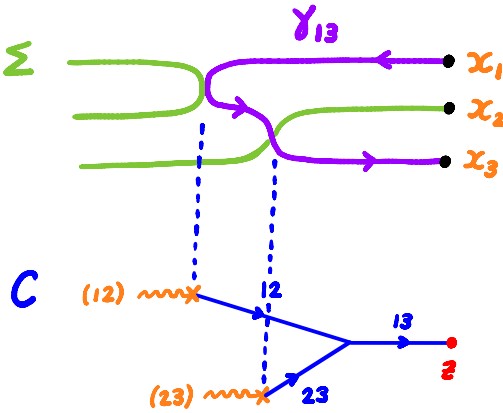

Figure 13: Example of a BPS web on $C$ (in blue) together with the corresponding detour path $\gamma_{13}$ on $\Sigma$ (in fuchsia).

If we also connect the pre-images $x_\alpha(\mathbf{z})$ and $x_\beta(\mathbf{z})$ by a path $\ell_{\alpha\beta} \subset F_\mathbf{z}$ in the fiber of $T^*C$, we can form a 2-cycle

$$D_{\alpha\beta} \subset T^*C \tag{139}$$

with boundary $\partial D_{\alpha\beta} = p_{\alpha\beta} \cup \ell_{\alpha\beta}$. Figure 14 illustrates such a 2-cycle $D_{\alpha\beta}$ in a two-dimensional cartoon. Even though the 2-cycle $D_{\alpha\beta}$ depends on the choice of the path $\ell_{\alpha\beta} \subset F_{\mathbf{z}}$, the central charge

$$Z_{\alpha\beta} = \int_{D_{\alpha\beta}} d\lambda \tag{140}$$

does not depend on this choice. Indeed, suppose we choose a different $\ell'_{\alpha\beta} \subset F_{\mathbf{z}}$, then the integral of $d\lambda$ over the 2-cycle in the fiber $F_{\mathbf{z}}$ bounded by $\ell_{\alpha\beta}$ and $\ell'_{\alpha\beta}$ is zero.

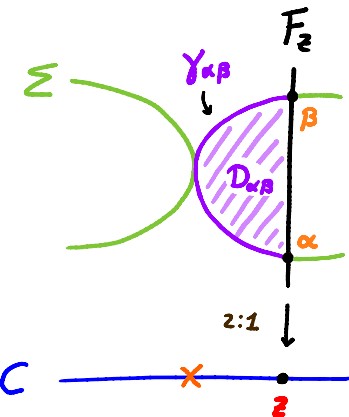

Figure 14: Two-dimensional picture of an open disc $D_{\alpha\beta} \subset T^*C$ (arcaded in fuchsia) with one boundary component $\gamma_{\alpha\beta} \subset \Sigma$ (in fuchsia) and another in the fiber $F_{\mathbf{z}} \subset T^*C$ (in light-green). A more detailed three-dimensional illustration is found in Figure 15.

Furthermore, the BPS condition on the BPS state implies that the 1-form $\lambda$ has a constant phase $\vartheta$ along $\gamma_{\alpha\beta}$. In particular,

$$\mathrm{Im}\left(e^{-i\vartheta}\int_{\gamma_{\alpha\beta}}\lambda\right) = \mathrm{Im}\left(e^{-i\vartheta}Z_{\alpha\beta}\right) = 0 \tag{141}$$

This shows that the 2-cycle $D_{\alpha\beta}$, considered as an open disc, is **special Lagrangian** with respect to the holomorphic symplectic form

$$\Omega_\zeta = e^{-i\vartheta}d\lambda. \tag{142}$$

The BPS condition in the 2d susy field theory therefore corresponds to a so-called calibration condition in the associated geometry. Such a correspondence occurs frequently when studying supersymmetric theories.

We conclude that 2d BPS states in the 2d $\mathcal{N} = (2,2)$ theory $\mathcal{T}_{\mathbf{z}}$ can be encoded as **open special Lagrangian discs** in the spectral geometry $T^*C$, that have one boundary component on $\Sigma$ and another on the fiber $F_{\mathbf{z}}$ of $T^*C$.[19]

---

[19]In §4 we will encounter $\alpha\beta$-trajectories that are part of saddle trajectories. In this case the two associated open cycles together form a closed cycle, that is again special Lagrangian with respect to the symplectic form $\zeta^{-1}d\lambda$.

### 3.4.1 Example: open discs in Landau-Ginzburg models

Let us construct the open discs $D_{\alpha\beta}$ explicitly for Landau-Ginzburg models with $n$ chiral fields and a polynomial superpotential $W$ of degree $k$. Remember from §2.4.1 that the spectral curve $\Sigma$ for a Landau-Ginzburg model can be embedded in $T^*C$, with horizontal coordinates $\mathbf{z}^{(l)}$ and vertical coordinates $q_l = \partial_{\mathbf{z}^{(l)}} W$. Also remember that we denoted the lift of $W$ to a holomorphic function on $T^*C$ as $\overline{W}$. And that imposition of the ring relations amongst the $q_l$, i.e. reintroducing $\phi^i$ dependence, gives an embedding $\iota_{\mathbf{z}} : \mathbb{C}^n_\phi \hookrightarrow T^*_{\mathbf{z}} C$ for all $\mathbf{z} \in C$.

Choose any $\mathbf{z} \in C$ on a $\alpha\beta$-trajectory in the spectral network $W_\vartheta$. Follow the $(\alpha\beta)$-trajectory backwards to the branch-point $\mathbf{z}_0$ that it is originating from. Consider all $\mathbf{z}'$ on the $\alpha\beta$-trajectory in between, and including, the branch-point $\mathbf{z}_0$ and the point $\mathbf{z}$, as well as their pre-images $\iota_{\mathbf{z}'}(\alpha)$ and $\iota_{\mathbf{z}'}(\beta)$ on $\Sigma \subset T^*C$.

At the phase $\vartheta$, and for each such $\mathbf{z}'$, the vacua $\alpha$ and $\beta$ are connected by the gradient flow with respect to the Morse function

$$\mathbf{h} = \mathrm{Im}(\zeta^{-1} W(\mathbf{z}')). \tag{143}$$

The corresponding gradient flow lines $\ell_{\alpha\beta}(\mathbf{z}')$ can be embedded in $T^*C$ using the embedding $\iota_{\mathbf{z}'}$. The open disc $D_{\alpha\beta} \subset T^*C$ is then defined as the union

$$D_{\alpha\beta} = \bigcup_{\mathbf{z}'} \iota_{\mathbf{z}'}(\ell_{\alpha\beta}(\mathbf{z}')) \tag{144}$$

for all $\mathbf{z}'$ between, and including, the branch-point $\mathbf{z}_0$ and $\mathbf{z}$. This defines a submanifold in $T^*C$ that truly represents the BPS state. The resulting open disc is illustrated in Figure 15 for the single field cubic model.

Note that in this setup

$$Z_{\alpha\beta} = W(\beta) - W(\alpha) = \int_{i_{\mathbf{z}}(\ell_{\alpha\beta})} d\overline{W}, \tag{145}$$

whereas

$$\int_{i_{\mathbf{z}}(\ell_{\alpha\beta})} d\overline{W} - \int_{\gamma_{\alpha\beta}} d\overline{W} = \int_{\partial D_{\alpha\beta}} d\overline{W} = 0. \tag{146}$$

because of Stokes theorem. Moreover, since we may identify the Liouville 1-form $\lambda$ with $d\overline{W}$ on $\Sigma$, we indeed conclude that

$$Z_{\alpha\beta} = \int_{\gamma_{\alpha\beta}} \lambda. \tag{147}$$

Note that we can rewrite this as the symplectic area of the disc:

$$Z_{\alpha\beta} = \int_{D_{\alpha\beta}} d\lambda. \tag{148}$$

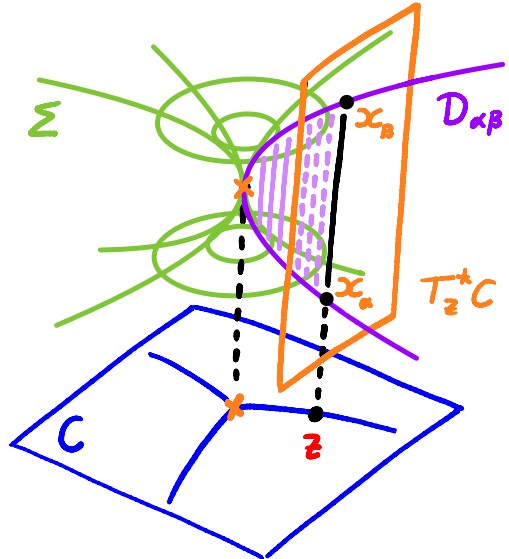

Figure 15: Three-dimensional picture of an open disc $D_{\alpha\beta} \subset T^*C$ (arcaded in fuchsia) in the Landau-Ginzburg model with cubic superpotential.

# 4 Sigma models and vortices

So far we have illustrated the properties of 2d $\mathcal{N} = (2,2)$ theories with Landau-Ginzburg examples. In particular, we have not studied 2d theories with gauge interactions yet. Let us remedy this here.

## 4.1 Vortices in 2d gauge theory

Consider a 2d $\mathcal{N} = (2,2)$ theory with a $U(k)$ gauge group coupled to $N$ massless chiral superfields that transform in the fundamental representation, and a twisted superpotential

$$\widetilde{W}_{\mathrm{FI}} = \frac{i\tau}{4}\mathrm{Tr}\,\Sigma, \tag{149}$$

where $\tau = ir + \frac{\theta}{2\pi}$ with **Fayet-Iliopoulos (FI) parameter** $r$ and theta angle $\theta$. Note that the FI term does not break $U(1)_A$-symmetry since $\Sigma$ carries axial $R$-charge 2.

The corresponding bosonic part of the Lagrangian reads

$$L_{\mathrm{bos}} = \frac{1}{e^2}\left(\frac{1}{2}\mathrm{Tr}\,F_A \wedge *F_A + (D_\mu\sigma)^2\right) + \sum_{i=1}^{N}|D_\mu\phi_i|^2 \tag{150}$$

$$- \sum_{i=1}^{N}\phi_i^\dagger\{\sigma,\sigma^\dagger\}\phi_i - \frac{e^2}{4}\mathrm{Tr}\left(\sum_{i=1}^{N}\phi_i\phi_i^\dagger - r1_k\right)^2$$

Note that the FI parameter enters in an essential way in this Lagrangian: if $r > 0$ it will allow us to turn on non-trivial vevs for the scalar fields $\phi_i$. This will be crucial to find **vortex configurations**.

We find these vortex configurations in a similar way to how we found BPS solitons in a LG model. We consider the $\mathcal{N} = (2, 2)$ theory in a Euclidean background and allow the scalar fields $\phi_i$ (the so-called **Higgs fields**) to have non-trivial winding when going around the circle at infinity of space-time. This winding is the most important characteristic of vortex configurations. It will localise the magnetic flux to a configuration of points in the two-dimensional space-time. These are the (zero-dimensional) vortices. See Figure 16 for an illustration. The scalar fields $\sigma$ will not play an important role in such configurations, and we will simply set them to zero for the time being.

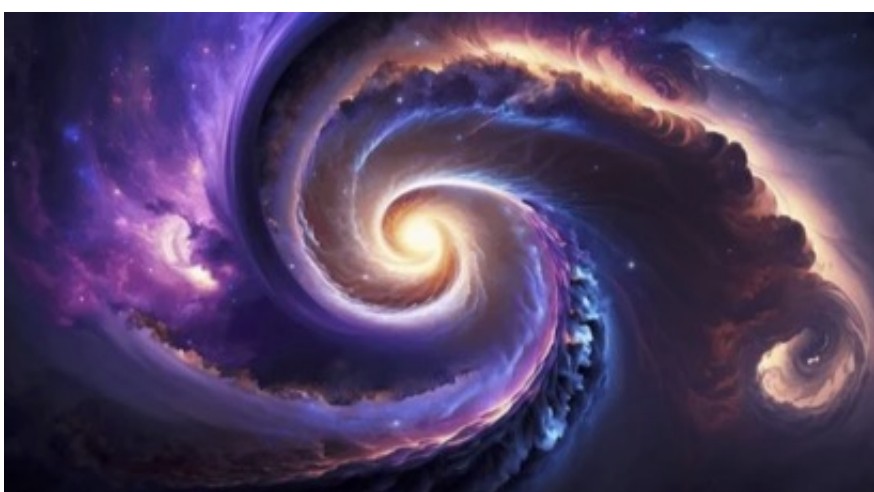

Figure 16: Illustration of a vortex configuration in space.

The energy of the resulting field configuration can be derived from the Lagrangian (150) and written in the form

$$E = \int_{\mathbb{R}^2} d^2z \, \frac{1}{e^2} \, \mathrm{Tr} \left( *F_A - \frac{e^2}{2} \left( \sum_{i=1}^{N} \phi_i \phi_i^\dagger - r \mathbb{1}_k \right) \right)^2 \tag{151}$$

$$+ \int_{\mathbb{R}^2} d^2z \sum_{i=1}^{N} |\overline{D}_A \phi_i|^2 + r \int_{\mathbb{R}^2} \mathrm{Tr} \, F_A,$$

where we have parametrized the Euclidean space-time with complex coordinates $z$ and $\bar{z}$, and the corresponding covariant derivatives $D_A$ and $\overline{D}_A$. This shows that the energy

$$E \geq r \int_{\mathbb{R}^2} \mathrm{Tr} \, F_A \tag{152}$$

is greater or equal than $r$ times the flux through the surface. The flux may be computed as the first Chern character of the gauge bundle and is known as the **vortex**

**number** $\mathfrak{m}$. Roughly, we may think of a field configuration with vortex number $\mathfrak{m}$ as a configuration of $\mathfrak{m}$ elementary point-like vortices in Euclidean space-time.

The energy is minimized for field configurations that obey the system of first order equations

$$* F_A = \frac{e^2}{2} \sum_{i=1}^{N} \phi_i \phi_i^\dagger - r 1_k,$$
$$\overline{D}_A \phi_i = 0. \tag{153}$$

These equations are called the **vortex equations**. Their solutions describe vortex configurations labeled by the vortex number $\mathfrak{m}$. Solutions to the vortex equations are invariant under the supersymmetry transformations generated by $Q_-$ and $\overline{Q}_+$, and are therefore half-BPS configurations.[20] Their twisted central charge $\tilde{Z}$ is computed by the right-hand side of equation (152).[21] Much more about vortices can be learnt, for instance, from David Tong's lecture notes on solitons [41].

These point-like vortices are analogues of point-like instantons in four dimensions. As in 4d $\mathcal{N} = 2$ theories, we can twist the 2d $\mathcal{N} = (2, 2)$ gauge theory in such a way that its partition function localizes to the moduli space of solutions to the vortex equations

$$\mathcal{M}_{\text{vortex}} = \bigcup_{\mathfrak{m}} \mathcal{M}_{\text{vortex},\mathfrak{m}}, \tag{154}$$

whose components are labeled by the vortex number $\mathfrak{m}$. Since the vortex solutions are annihilated by $Q_-$ and $\overline{Q}_+$, the relevant twist is the A-twist with scalar supercharge $Q_A^\xi$ (for any phase $\xi$).

The Lagrangian of the 2d gauge theory is $Q_A^\xi$-exact, up to the topological term

$$S_{\text{top}} = 2\pi i \tau \int_{\mathbb{R}^2} \text{Tr}\, F_A. \tag{155}$$

The resulting partition function therefore has the form

$$\mathcal{Z}^{\text{vortex}}(\mathbf{z}) = \sum_{\mathfrak{m}} \mathbf{z}^{\mathfrak{m}} \oint_{\mathcal{M}_{\text{vortex},\mathfrak{m}}} 1, \tag{156}$$

with exponentiated complexified FI parameter

$$\mathbf{z} = e^{2\pi i \tau} \in \mathbb{C}^*. \tag{157}$$

This partition function is known as the **vortex partition function**. Note that $\mathbf{z}$ is the symbol we use to parametrize UV deformations of 2d $\mathcal{N} = (2, 2)$ theories and,

---

[20] In the presence of a superpotential $W(\phi)$ the second vortex equation instead turns into the **BPS instanton equation** $\overline{D}_A \phi_i = \frac{i\zeta}{2} g^{i\bar{j}} \overline{\partial_j W}$. This is a slightly modified version of the BPS soliton equation (103) that preserves only a single supercharge $Q_A^\zeta$ (see for example equation (5.13) of [40]).

[21] Yet, remember that $\tilde{Z}$ is quantum-corrected (in contrast to $Z$).

following the philosophy of §2.4.1, the complex FI parameter $\tau$ indeed parametrizes deformations of GLSM by adding the FI term (149) to the microscopic Lagrangian.

We will get back to computing vortex partition functions in §4.4. We finish this section by noting that for more general theories, including for instance 2d quiver gauge theories, the vortex configurations will be labeled by multiple vortex numbers, and the vortex partition function will correspondingly be given by a sum over all vortex numbers.

## 4.2 GLSM's and their vacuum structure

Our aim in this section is to find the vacuum structure of the 2d $\mathcal{N} = (2,2)$ gauge theories that we introduced in §4.1. Such gauge theories, with the $U(k)$ gauge theory coupled to $N$ chiral fields as an important example, are known as **gauged linear sigma models (GLSM)**. They have a Lagrangian description in terms of 2d gauge fields and 2d chiral matter fields, and are called linear because the corresponding Kähler potential is quadratic in the fields (and in particular does not include any higher order interaction terms).[22]

In a classical approximation, the supersymmetric vacua of a GLSM can be found by solving for the field configurations for which the potential energy $U$ is zero, as a function of the value of the FI parameter $r$. We say that value of the FI parameter $r$ labels the different **phases** of the GLSM.

In our $U(k)$ example the potential energy $U$ is given by

$$U = \frac{e^2}{2}\text{Tr}\left(\sum_{i=1}^{N}\phi_i\phi_i^{\dagger} - r1_k\right)^2 + \frac{1}{2e^2}\text{Tr}[\sigma,\sigma^{\dagger}]^2 + \sum_{i=1}^{N}\phi_i^{\dagger}\{\sigma,\sigma^{\dagger}\}\phi_i. \tag{158}$$

If we choose $r > 0$ the potential energy is minimised by field configurations such that

$$\sum_{i=1}^{N}|\phi_i|^2 = r \quad \text{and} \quad \sigma = 0, \tag{159}$$

modulo $U(k)$ gauge transformations. If, on the other hand, we choose $r = 0$ then all $\phi_i$ need to be zero, whereas (the diagonal part of) $\sigma$ is free. And if $r < 0$ there are no supersymmetric vacua at all, so that the supersymmetry appears to be spontaneously broken. (We will argue soon that this is not the case at the quantum level.)

The moduli space of (classical) supersymmetric vacua for $r > 0$ is known as the (classical) **Higgs branch**, since, through a supersymmetric extension of the Higgs mechanism, the gauge group is spontaneously broken after turning on vevs for the Higgs fields $\phi_i$. The moduli space of (classical) supersymmetric vacua for $r = 0$ is known as the (classical) **Coulomb branch**, since the gauge group is broken to a product of $U(1)$'s.

---

[22] An introduction to GLSM's, as well as an explanation of many of its intricacies, can be found in §15 of [42].

The classical Higgs branch in our example is equal to the Grassmannian of $k$-planes inside of $\mathbb{C}^N$. For example, when $k = 1$ we find that

$$\mathcal{M}_{\text{Higgs}} = \{(\phi_1, \ldots, \phi_N) \,|\, \sum_{i=1}^{N} |\phi_i|^2 = r\} /\!\!/ U(1) = \mathbb{P}^{N-1}. \tag{160}$$

We will therefore refer to this GLSM as the $\mathbb{P}^{N-1}$-**model**. More generally, if we add new chiral fields in the anti-fundamental representation of $U(k)$ to our GLSM, we find that its classical Higgs branch is described by a flag manifold. And even more generally, the Higgs branch for an abelian GLSM can often be described as a toric manifold, whereas the Higgs branch for a non-abelian GLSM could be any Kähler quotient.

Quantum-mechanically, we have to take into account one-loop corrections. In particular, there is a divergent loop which renormalises the FI parameter $r$. For instance, in the $\mathbb{P}^{N-1}$-model we use the renormalised quantity

$$r(\mu) = r_0 - \frac{N}{2\pi} \ln\left(\frac{\Lambda_0}{\mu}\right) \tag{161}$$

where $\Lambda_0$ is the UV cutoff and $\mu$ is the energy scale. By choosing $\mu > \Lambda_0$ we can thus make sure that FI parameter $r$ is positive. The FI parameter $r'$ at a lower energy scale $\mu'$ is obtained from the FI parameter at the energy scale $\mu$ by

$$r(\mu) = r'(\mu') + \frac{N}{2\pi} \log\left(\frac{\mu}{\mu'}\right). \tag{162}$$

For a general GLSM, the running of the FI parameters is determined by the charges of the chiral fields under the gauge groups. The theory is asymptotically free (just like the $\mathbb{P}^{N-1}$ model) when the vacuum manifold is Fano (i.e. its first Chern class is positive on any holomorphic curve), whereas it is conformal (the FI parameters do not run) when the vacuum manifold is Calabi-Yau.

### 4.2.1 Non-linear sigma model on the Higgs branch

Let us consider the case $r > 0$ in the $\mathbb{P}^{N-1}$-model in more detail. Note that the modes of $\phi_i$ that are tangent to the classical vacuum manifold are massless, whereas the field $\sigma$ and the modes of $\phi_i$ that are transverse to the vacuum manifold have obtained a mass $e\sqrt{2r}$. The gauge field acquires the same mass by the Higgs mechanism. Furthermore, the massless modes of the fermion fields may be interpreted as the (shifted) tangent vectors to the vacuum manifold, whereas all other fermionic modes have the same mass $e\sqrt{2r}$.

If we consider the theory in the regime $e\sqrt{r} \gg \mu$, the massive modes decouple and can be integrated out. The massless modes can instead be reorganised into a $\mathcal{N} = (2,2)$ theory of maps from the 2d space-time into the vacuum manifold $\mathbb{P}^{N-1}$. The kinetic terms are of the form

$$g_{i\bar{j}}(\phi)\, \partial^\mu \phi^i \partial_\mu \overline{\phi^j} + i g_{i\bar{j}}\, \overline{\psi^j} D_\mu \psi^i \tag{163}$$

where the metric is proportional to the Fubini-Study metric on $\mathbb{P}^{N-1}$ (and in particular has a non-trivial dependence on the fields $\phi^i$), and there is an additional four-fermion interaction of the form

$$R_{i\bar{j}k\bar{l}}\,\psi^i\overline{\psi^j}\psi^k\overline{\psi^l}, \tag{164}$$

that results from plugging in the background value for the $\sigma$-field. Altogether, one can argue that, at energies much smaller than $e\sqrt{r}$, the linear sigma model is effectively described by a **non-linear sigma model**, or NLSM, into the vacuum manifold $\mathbb{P}^{N-1}$.

A similar discussion holds for any GLSM that is asymptotically free – in this case the FI parameter $r$ takes values in the Kähler cone of the vacuum manifold. If the theory is conformal, the FI parameter $r$ does not run, and there might be additional phases (with $r \leq 0$) in which the vacuum manifold may develop (orbifold) singularities.

The vortices introduced in §4.1 can be interpreted in the NLSM in terms of quasi-maps from $\mathbb{P}^1$ into the Grassmannian, with suitable boundary conditions at infinity of $\mathbb{P}^1$. This relates 2d $\mathcal{N} = (2,2)$ gauge theories to topics as Gromov-Witten theory, geometric representation theory and Givental's J-functions [43] (see for instance [44] for an overview of such relations).

So far we have studied the low energy description of the GLSM at energies smaller than $e\sqrt{r}$ and seen that in this regime they have an effective description in terms of NLSM's. It is needed to go to much lower energies though to find the discrete vacuum structure that we are looking for. One way to do so is to study the supersymmetric ground states of the NLSM. In the A-twist these ground states are in 1-1 correspondence with de Rham cohomology classes of the vacuum manifold. In particular, the Witten index, computing the number of ground states, is equal to the Euler characteristic of the vacuum manifold. This tells us for instance that the $\mathbb{P}^{N-1}$-model admits $N$ supersymmetric vacua. The full chiral ring of the NLSM can be obtained as the so-called quantum cohomology ring in Gromov-Witten theory.

### 4.2.2 Effective twisted superpotential on the Coulomb branch

Here we take an alternative approach.[23] In general, the vacuum structure of a GLSM is a combination of Higgs, Coulomb and mixed branches, and we could study our GLSM in any of the associated phases. As long as these phases are connected smoothly, their vacuum structure should be equivalent. After all, the vacuum structure is determined by a topological supercharge, and thus invariant under small deformations. In particular, the Witten index tells us that the number of vacua stays invariant.

So instead of focusing on the Higgs branch, we could also study the low energy structure on the Coulomb branch, where the gauge group is broken to a product of $U(1)$ factors. We do this by assuming that the complex scalar field $\sigma$ is large and

---

[23]Many more details may be found in §15.5 of [42].

slowly varying. This assumption implies that the chiral fields $\phi_i$ are heavy, since their masses are proportional to the eigenvalues of $\sigma$, as can read off from the potential energy $U$ in equation (158). To find the low energy description on the Coulomb branch, we thus need to integrate out all matter fields. Because of supersymmetry, the low energy description can be specified in terms of an effective Kähler potential $K_{\text{eff}}(\Sigma, \overline{\Sigma})$ and an **effective twisted superpotential** $\widetilde{W}_{\text{eff}}(\Sigma)$.

The scalar potential for the effective theory is

$$U_{\text{eff}} = g^{\Sigma\overline{\Sigma}} \left| \frac{\partial \widetilde{W}_{\text{eff}}}{\partial \Sigma} \right|^2, \tag{165}$$

where $g^{\Sigma\overline{\Sigma}}$ is the inverse of

$$g_{\Sigma\overline{\Sigma}} = \frac{1}{4} \frac{\partial^2 K_{\text{eff}}(\Sigma, \overline{\Sigma})}{\partial \Sigma \, \partial \overline{\Sigma}}. \tag{166}$$

The (quantum) Coulomb vacua are thus encoded as the critical points of the effective twisted superpotential.

This superpotential $\widetilde{W}_{\text{eff}}$ consists of the original FI term plus an additional sum of 1-loop contributions for all matter fields that are integrated out.[24] If $\Phi$ is a chiral superfield of charge 1 under a $U(1)$ gauge group, its contribution to the effective twisted superpotential at energy scale $\mu$ is

$$\widetilde{W}_{\text{eff}}(\Sigma) = -\frac{1}{8\pi} \Sigma \left( \log \frac{\Sigma}{\mu} - 1 \right). \tag{167}$$

For the $\mathbb{P}^{N-1}$-model this implies that

$$\widetilde{W}_{\text{eff}}(\Sigma) = \frac{i}{4} \tau \Sigma - \frac{N}{8\pi} \Sigma \left( \log \frac{\Sigma}{\mu} - 1 \right). \tag{168}$$

Let us make two important remarks about this expression:

- The GLSM is thus described on the (quantum) Coulomb branch by a Landau-Ginzburg model with twisted chiral field $\Sigma$ and twisted superpotential $\widetilde{W}_{\text{eff}}(\Sigma)$. In contrast to the holomorphic superpotentials we saw before, $\widetilde{W}_{\text{eff}}(\Sigma)$ has a logarithmic singularity at the origin of the Coulomb branch. As we alluded to in §2.1.3, we will thus generalise the notion of LG superpotential from hereon.

---

[24]The $\mathcal{N} = (2,2)$ decoupling theorem says that there cannot be any mixing between parameters in the superpotential and the twisted superpotential in the renormalization flow. Furthermore, parameters from the (twisted) superpotential can enter the Kähler potential, but not vice versa. Moreover, the $\mathcal{N} = (2,2)$ non-renormalisation theorem says that the terms in the (twisted) superpotential do not change at all in the flow, unless some massive fields get integrated out. The expressions for the integrated out matter fields resemble quantum corrections to the remaining fields though. See for instance §14.3 of [42] for proofs.

- The derivative

$$\tau_{\text{eff}}(\sigma) := -4i\,\partial_\sigma \widetilde{W}_{\text{eff}}(\sigma) = \tau + \frac{iN}{2\pi} \log \frac{\sigma}{\mu} \tag{169}$$

defines the **effective complexified FI parameter**. This parameter takes the role of the complexified coupling constant in the LG model defines by $\widetilde{W}_{\text{eff}}(\Sigma)$. Note that this indeed agrees with the running of the effective FI parameter $r_{\text{eff}} = \text{Im}\,\tau_{\text{eff}}$ as discussed around equation (162). Note that $r_{\text{eff}}$ is large and positive when $\sigma \gg \mu$ and large and negative when $\sigma \ll \mu$. That is, the GLSM is in a **strong (weak) coupling** regime when $\sigma \gg \mu$ ($\sigma \ll \mu$).

The twisted chiral ring for any GLSM is then obtained from the effective twisted superpotential as the Jacobian ring in $\sigma$ with the relation

$$\frac{\partial \widetilde{W}_{\text{eff}}(\sigma)}{\partial \sigma} = 0. \tag{170}$$

In particular, this implies that the spectral curve for the $\mathbb{P}^{N-1}$-model is cut out by the equation

$$\sigma^N = \mu^N\, \mathbf{z}. \tag{171}$$

with exponentiated complexified FI parameter $\mathbf{z} = e^{2\pi i \tau} \in \mathbb{C}^*$. Note that from the Coulomb branch perspective there is no restriction of the value of the FI parameter $r$. In particular, we find that there are $N$ vacua for each fixed choice of $\tau \in \mathbb{C}$.

The spectral curve equation (171) may be familiar to you from the Gromov-Witten perspective, where the quantum cohomology ring for $\mathbb{P}^{N-1}$ is generated by the hyperplane class $H$ with relation

$$H^N = e^{2\pi i \tau}, \tag{172}$$

where $\tau$ has the interpretation of the complexified Kähler class of $\mathbb{P}^{N-1}$. This relation indicates that the classical cohomology ring, with relation $H^N = 0$, gets quantum-deformed by holomorphic maps from $\mathbb{P}^1$ into $\mathbb{P}^{N-1}$ weighted by $\tau$.

### 4.2.3 Turning on twisted masses

The previous discussion changes slightly if we turn on **twisted masses**. Combining the relevant terms from §2.1.3, the microscopic Lagrangian for the $\mathbb{P}^{N-1}$-model with twisted masses reads

$$\frac{1}{4} \int d^4\theta \left\{ -\frac{1}{2e^2} \bar{\Sigma}\Sigma + \Phi_j\, e^{2V + 2\langle \widehat{V}_j \rangle}\, \overline{\Phi}_j \right\} + \int d^2\theta\, \widetilde{W}(\Sigma) + h.c. \tag{173}$$

with FI term $\widetilde{W}(\Sigma) = i\tau\Sigma/4$.

The expectation values $\langle \widehat{V}_j \rangle$ introduce twisted masses associated to the $U(1)$ factors of the maximal torus

$$\prod_{j=1}^{N} U(1)_j \tag{174}$$

of the $U(N)$ flavour symmetry. Each $U(1)_j$-factor acts (only) on the chiral field $\Phi_j$ with charge $+1$, and therefore induces a twisted mass $\widetilde{m}_j$ for this chiral field. After turning on these twisted masses, the flavour symmetry is broken to its maximal torus. Since the global symmetry is really $SU(N) = U(N)/U(1)_G$, we fix

$$\sum_{j=1}^{N} \widetilde{m}_j = 0 \tag{175}$$

using a gauge transformation.

The scalar potential takes the adjusted form

$$U = \sum_{j=1}^{N} \frac{1}{2} |\sigma - \widetilde{m}_j|^2 |\phi_j|^2 + \frac{e^2}{2} \left( \sum_{j=1}^{N} |\phi_j|^2 - r \right)^2 . \tag{176}$$

As before, there are two cases:

1. When $r \geq 0$ we can solve for $U = 0$ with $|\phi_j|^2 = \delta_{ja} r$ and $\sigma = \widetilde{m}_a$, for any given $1 \leq a \leq N$. In the massive model we thus find a discrete set of $N$ classical vacua, parametrized by the chiral fields $\phi_j$. As we will see below, this will remain the case when we add quantum corrections. Also note that $\sigma$ is no longer free at $r = 0$.

2. When $r < 0$ we cannot solve for $U = 0$ and therefore supersymmetry is broken on the classical level. However, the Witten index argument tells us that we expect the $N$ vacua to re-appear at the quantum level.

The analysis at the quantum level is similar to before. In the first case we assume that $e\sqrt{r} \gg 1$ and integrate out the gauge field to obtain a non-linear sigma model into $\mathbb{P}^{N-1}$. The homogeneous coordinates of $\mathbb{P}^{N-1}$ are given by the chiral scalars $\phi_i$ and, in the coordinate patch where $\phi_k = 1$, the Lagrangian reads

$$-\frac{1}{4g^2} \int d^4\theta \, \log \left( 1 + \sum_{j \neq k} \overline{W}_j \, e^{2\langle \hat{V}_j \rangle - 2\langle \hat{V}_k \rangle} \, W_j \right), \tag{177}$$

where $W_j := \Phi_j/\Phi_k$. The metric $g_{i\bar{j}}$ on $\mathbb{P}^{N-1}$ is now a version of the Fubini-Study metric deformed by the twisted masses with $g^2 = -\frac{1}{r}$. The mass deformation does not change the Euler characteristic of $\mathbb{P}^{N-1}$, so that we still have $N$ vacua at the quantum level.

In the second case, the correct approach is to consider the limit $\sigma \gg e$ in which the chiral fields are very massive and should be integrated out in the path integral. The resulting effective Lagrangian is

$$\frac{1}{4} \int d^4\theta \, K_{\text{eff}}(\Sigma, \bar{\Sigma}) + \left( \int d^2\theta \, \widetilde{W}_{\text{eff}}(\Sigma) + h.c. \right) \tag{178}$$

with

$$\widetilde{W}_{\text{eff}}(\Sigma) = \frac{1}{4} \left[ i\tau \, \Sigma - \frac{1}{2\pi} \sum_{j=1}^{N} (\Sigma - \widetilde{m}_j) \left( \log\left( \frac{\Sigma - \widetilde{m}_j}{\mu} \right) - 1 \right) \right]. \tag{179}$$

Note that this superpotential has a logarithmic singularity at each $\sigma = \widetilde{m}_j$. The resulting effective FI coupling is given by

$$\tau_{\text{eff}}(\sigma) = -4i \, \partial_\sigma \widetilde{W}_{\text{eff}}(\sigma) = \tau + \frac{i}{2\pi} \sum_{j=1}^{N} \log\left( \frac{\sigma - \widetilde{m}_j}{\mu} \right). \tag{180}$$

Setting the scalar potential (165) to zero implies that the spectral curve for the massive $\mathbb{P}^{N-1}$−model is cut out by the equation

$$\Sigma: \quad \prod_{j=1}^{N} (\sigma - \widetilde{m}_j) = \mu^N e^{2\pi i \tau} = \mu^N \mathbf{z}. \tag{181}$$

We see that this equation has $N$ solutions for every choice of $\mathbf{z} \in \mathbb{C}^*$, which confirms that this $\mathbb{P}^{N-1}$-model indeed has $N$ quantum vacua.

Let us emphasize that this section has shown that Landau-Ginzburg models are universal: they describe the low-energy physics of any GLSM. Yet, compared to the LG models discussed in §3 we need to allow for one generalisation: the exterior derivative of the superpotential $W$ should be allowed to be a closed (but not necessarily exact) holomorphic 1-form $dW$, as in equation (30).

### 4.2.4 Spectral geometry

Following the philosophy of §2.4.1, the expression (179) for the superpotential $\widetilde{W}_{\text{eff}}$ tells us that the space of deformations of our GLSM is $C = \mathbb{C}_\tau$. Furthermore, the relation

$$\partial_\tau \widetilde{W}_{\text{eff}} = \frac{i}{4} \sigma \tag{182}$$

implies that we are allowed to parameterise the fibers of $T^*C$ by $-i\sigma$. The spectral curve (181) can then be embedded into $T^*C$.[25]

---

[25]To embed the spectral curve into $T^*C$ as prescribed in equation (93), one would need to parametrise the fibers by $\partial_\tau W_{\text{eff}} = \frac{i}{4}\sigma$ instead. We take a slightly different approach here to avoid inconvenient prefactors in later expressions.

We will later find it useful to introduce the (strong coupling) variable

$$s = e^{-2\pi i \tau} = \mathbf{z}^{-1}, \tag{183}$$

in terms of which the Liouville one-form $\lambda = -i\sigma d\tau$ reads

$$\lambda = \frac{\sigma}{2\pi} \frac{ds}{s}. \tag{184}$$

When restricted to the **spectral curve** (181), $\lambda$ can be expressed as

$$\lambda|_\Sigma = \frac{1}{2\pi} \left( N d\sigma - \sum_{j=1}^{N} \frac{\widetilde{m}_j d\sigma}{\sigma - \widetilde{m}_j} \right). \tag{185}$$

Note that the spectral curve for $N = 2$ can be written in the form

$$\sigma^2 = \frac{1}{s} + m^2, \tag{186}$$

if we choose $\widetilde{m}_1 = m = -\widetilde{m}_2$ as well as $\mu = 1$, with

$$\lambda|_\Sigma^2 = \frac{1}{(2\pi)^2} \left( \frac{1}{s^3} + \frac{m^2}{s^2} \right) ds^2. \tag{187}$$

These formulae may look familiar to you: they define "half" of the Seiberg-Witten geometry of the four-dimensional pure $SU(2)$ theory. This is because the $\mathbb{P}^1$-model appears as the world-volume description of a "canonical" surface defect in the four-dimensional pure $SU(2)$ theory. In particular, the Seiberg-Witten curve reduces to $\Sigma$ in the limit where we decouple the 4d gauge dynamics.

The Higgs and Coulomb phase correspond to disjoint regions in the parameter space $C = \mathbb{C}_\tau$. To see this, assume that the gauge coupling $e$, the mass parameters $m$ and the energy scale $\mu$ are all fixed, while introducing the new scale

$$\Lambda^N = \mu^N e^{2\pi i \tau} = \mu^N s^{-1}. \tag{188}$$

In the Higgs phase the FI parameter $e\sqrt{r}$ is assumed much larger than the energy scale $\mu$, so that $\Lambda \ll e$. In the Coulomb phase the expectation value of $\sigma$ is assumed much larger than the energy scale $\mu$. Since $\sigma^2 \sim 1/s$, this implies that $\Lambda \gg e$. The Higgs vacua are thus located far away from the origin of the $s$-plane, while the Coulomb vacua are situated close to the origin of the $s$-plane.

The Higgs vacua are moreover weakly coupled (since $r_{\text{eff}}$ is large and negative), while the Coulomb vacua are strongly coupled (since $r_{\text{eff}}$ is large and positive). The weak and strong coupling regions are separated by a wall of marginal stability, which we describe in detail for the $\mathbb{P}^1$-model in §4.3. This is illustrated in Figure 17.

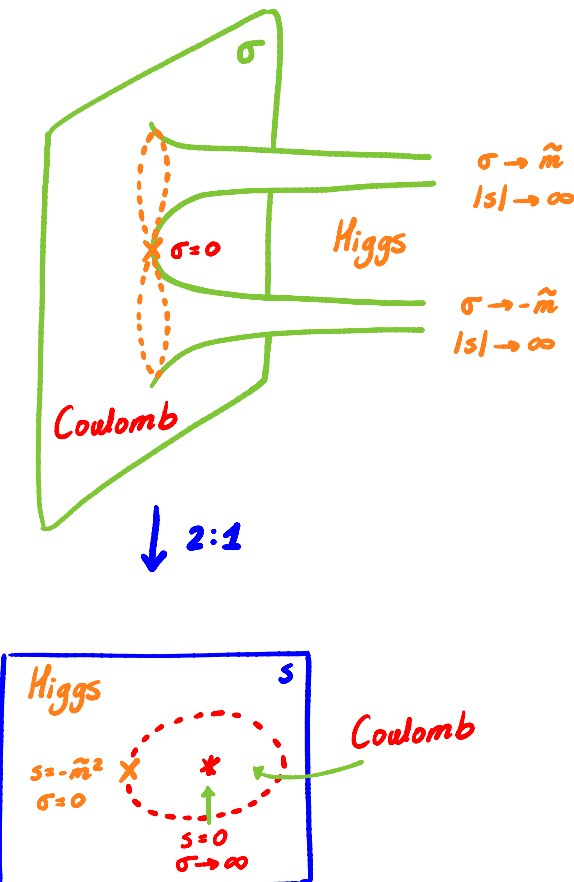

Figure 17: Illustration of the spectral curve $\Sigma$ (in light-green) for the $\mathbb{P}^1$-model as a double covering over the $s$-splane $\mathbb{C}^*_s$ (in blue), together with an indication of the Higgs branch (weak coupling region) and the Coulomb branch (strong coupling region), which are separated by a wall of marginal stability (in dashed red).

Figure 18: Mirror symmetry for the $\mathbb{P}^{N-1}$-model.

### 4.2.5 Remark: mirror symmetry

Remember that at the level of the supersymmetry algebra, **mirror symmetry** corresponds to the automorphism

$$Q_- \leftrightarrow \bar{Q}_-, \quad F_V \leftrightarrow F_A, \quad Z \leftrightarrow \tilde{Z}. \tag{189}$$

This automorphism maps chiral superfields to twisted chiral superfields and vice versa. It thus suggests that there is a duality between pairs of 2d $\mathcal{N} = (2,2)$ theories in which the (quantum) Higgs branch of one theory is exchanged with the (quantum) Coulomb branch of the other. This duality indeed exists and can be traced back to T-duality in string theory [45]. We say that the two theories in each such pair are **UV mirrors** of each other.

The UV mirror for any abelian GLSM can be found through the Hori-Vafa prescription by T-dualising the phase of the chiral fields $\Phi_j$ [42]. The field content of the UV mirror for the massive $\mathbb{P}^{N-1}$-model consists of a twisted chiral field $\Sigma$ (the field strength constructed from the vector multiplet) coupled to $N$ (neutral) twisted chiral fields $\widetilde{Y}_j$ and their complex conjugates $\overline{\widetilde{Y}}_j$. These mirror fields are related to the original vector and chiral fields, $V$ and $\Phi_j$ respectively, as

$$\widetilde{Y}_j + \overline{\widetilde{Y}}_j = 2\,\overline{\Phi}_j\, \mathrm{e}^{2V+2\langle V_j\rangle}\, \Phi_j. \tag{190}$$

The Lagrangian of the mirror theory takes the form

$$\widetilde{L} = -\int \mathrm{d}^4\theta\, \frac{1}{2e^2}\, \bar{\Sigma}\Sigma - \sum_{j=1}^{N} \int \mathrm{d}^4\theta\, \frac{1}{2}\, (\widetilde{Y}_j + \overline{\widetilde{Y}}_j) \log (\widetilde{Y}_j + \overline{\widetilde{Y}}_j)$$
$$+ \frac{1}{2}\int \mathrm{d}^2\widetilde{\theta}\, \widetilde{W}_{\mathrm{exact}}(\widetilde{Y}_j, \Sigma) + c.c., \tag{191}$$

with the twisted superpotential

$$\widetilde{W}_{\text{exact}}(\widetilde{Y}_j, \Sigma) = \Sigma\left(\sum_{j=1}^{N} \widetilde{Y}_j - 2\pi i\tau\right) + \mu\sum_{j=1}^{N} e^{-\widetilde{Y}_j} - \sum_{j=1}^{N} \widetilde{m}_j\widetilde{Y}_j. \qquad (192)$$

Note that this superpotential consists of a linear term in $\Sigma$, a Toda-like interaction term, and a mass term. The linear term in $\Sigma$ suggests that the fields $\widetilde{Y}_j$ should be interpreted as dynamical FI parameters, whereas the interaction term gives this model the name $A_{N-1}$ affine Toda field theory or affine Toda LG model.

The Coulomb branch of the affine Toda theory is obtained by integrating out the twisted chiral $\Sigma$ in the Lagrangian (191), in the limit where this field is very massive. This yields the condition

$$\sum_{j=1}^{N} \widetilde{Y}_j = 2\pi i\tau, \qquad (193)$$

which is solved by the choices $\widetilde{Y}_{k<N} = \frac{2\pi i\tau}{N} - \log\widetilde{\Phi}_k$ and $\widetilde{Y}_N = \frac{2\pi i\tau}{N} + \sum_{k=1}^{N-1}\log\widetilde{\Phi}_k$. The resulting effective twisted superpotential is given by

$$\widetilde{W}_{\text{eff}}(\widetilde{\Phi}_k) = \Lambda\left(\widetilde{\Phi}_1 + \cdots + \widetilde{\Phi}_{N-1} + \prod_{k=1}^{N-1}\frac{1}{\widetilde{\Phi}_k}\right) + \sum_{k=1}^{N-1}(\widetilde{m}_k - \widetilde{m}_N)\log\widetilde{\Phi}_k \qquad (194)$$

where the twisted fields $\widetilde{\Phi}_k$ are valued in $\mathbb{C}^*$ and $\Lambda = \mu\, e^{2\pi i\tau/N}$.

To determine the chiral ring, we first compute $-\frac{1}{2\pi i}\frac{d\widetilde{W}_{\text{eff}}}{d\tau}$ (which we will soon recognize as the generator). Keeping in mind that the $\widetilde{\Phi}_i$ have a $\tau$ dependence, one finds

$$-\frac{1}{2\pi i}\frac{d\widetilde{W}_{\text{eff}}}{d\tau} = \Lambda\prod_{i=1}^{N-1}\frac{1}{\widetilde{\Phi}_i} + \widetilde{m}_N. \qquad (195)$$

The chiral ring relations can then be written as

$$\frac{\partial\widetilde{W}_{\text{eff}}}{d\widetilde{\Phi}_k} = 0 \implies -\frac{1}{2\pi i}\frac{d\widetilde{W}_{\text{eff}}}{d\tau} - \widetilde{m}_k = \Lambda\widetilde{\Phi}_k, \qquad (196)$$

and subsequently subsumed into the familiar equation

$$\prod_{j=1}^{N}\left(-\frac{1}{2\pi i}\frac{d\widetilde{W}_{\text{eff}}}{d\tau} - \widetilde{m}_j\right) = \mu^N e^{2\pi i\tau}, \qquad (197)$$

using relations (196) for the first $N-1$ factors and simply (195) for the $N$th factor.

The Coulomb branch of the affine Toda theory may thus be identified with the Higgs branch of the $\mathbb{P}^{N-1}$-model GLSM. We therefore say that the LG model with

superpotential $\widetilde{W}_{\text{eff}}$ is the **IR mirror** of the $\mathbb{P}^1$-model on the Higgs branch. In particular, we find that the IR mirror of the $\mathbb{P}^1$-model on the Higgs branch is given by the LG model with superpotential

$$d\widetilde{W}_{\text{eff}} = \left( \Lambda + \frac{2\widetilde{m}}{\widetilde{\Phi}} - \frac{\Lambda}{\widetilde{\Phi}^2} \right) d\widetilde{\Phi}. \tag{198}$$

On the other hand, if one moves to the Higgs branch of the mirror by integrating out the twisted chiral fields $Y_i$, one simply recovers the LG model of §4.2.3 with twisted chiral $\Sigma$. All these mirror symmetry statements are summarized in Figure 18. Note that mathematically, mirror symmetry is usually studied on the IR level as a correspondence between Fano varieties and LG models (see for instance [46, 47]).

## 4.3 Solitons and spectral networks for GLSM's

In this section we study the BPS soliton spectrum for 2d $\mathcal{N} = (2,2)$ gauge theories through spectral networks, with the $\mathbb{P}^1$-model as our main example. In §4.3.4 we make a little detour for those of you intrigued by string and M-theory. Following [48] we realize any GSLM as well as its BPS soliton spectrum using M2-branes in M-theory. This picture is also important for understanding surface defects in 4d $\mathcal{N} = 2$ theories.

### 4.3.1 BPS solitons in GLSM's

Given any GSLM with a finite set of vacua, we expect that there exist BPS solitons interpolating between these vacua. As before, these solitons may be found as solutions to the $\zeta$-soliton equation

$$\frac{d\sigma^i}{dx} = \frac{i\zeta}{2} g^{i\bar{j}} \overline{\partial_j \widetilde{W}_{\text{eff}}}, \tag{199}$$

and have central charge[26]

$$\widetilde{Z}_{\alpha\beta} = 4 \left( \widetilde{W}_{\text{eff}}(\beta) - \widetilde{W}_{\text{eff}}(\alpha) \right). \tag{200}$$

For the massive $\mathbb{P}^{N-1}$-model the superpotential $\widetilde{W}_{\text{eff}}$ is given in equation (179). We then find that

$$\widetilde{Z}_{\alpha\beta} = \frac{1}{2\pi} \left( N(\sigma_\beta - \sigma_\alpha) + \sum_{j=1}^{N} \widetilde{m}_j \log \left( \frac{\sigma_\beta - \widetilde{m}_j}{\sigma_\alpha - \widetilde{m}_j} \right) \right). \tag{201}$$

Note that this seems to lead to a complication when some of the twisted masses $\widetilde{m}_j$ are non-zero: the logarithm in $\widetilde{Z}_{\alpha\beta}$ gives rise to an ambiguity of the form

$$\Delta\widetilde{Z}_{\alpha\beta} = i \sum_{j=1}^{N} \widetilde{m}_j n_j, \tag{202}$$

---

[26]In this section we will adopt normalization conventions from [48].

with $n_j \in \mathbb{Z}$. This ambiguity reflects that the BPS solitons may be charged under the flavour symmetry $U(1)_j$ with charge $n_j$. Indeed, these charges precisely contribute the sum (202) to the central charge $\widetilde{Z}_{\alpha\beta}$.

It turns out that the $n_j$ can be determined by writing the central charge of the BPS soliton as an integral

$$\widetilde{Z}_{\alpha\beta} = \int_{\gamma_{\alpha\beta}} \lambda = \frac{1}{2\pi} \int_{\gamma_{\alpha\beta}} \sigma \frac{ds}{s} \tag{203}$$

of the 1-form $\lambda$ over its associated detour path $\gamma_{\alpha\beta}$ in the spectral curve $\Sigma$. We will see this explicitly in §4.3.2, where we derive the spectrum of the BPS solitons in the $\mathbb{P}^1$-model, across the parameter space $C = \mathbb{C}_\tau$, using the technology of spectral networks. In §4.3.4 we realize the BPS solitons in the $\mathbb{P}^{N-1}$-model as open M2-branes in M-theory.

### 4.3.2  Spectral networks for GLSM's

As we have learned in §3.2.2, the spectrum of BPS solitons can be conveniently read off from the family of spectral networks $\mathcal{W}^\vartheta$ embedded in $C_{\mathbf{z}}$. The 2d theory $T_{\mathbf{z}}$ admits a BPS soliton in its spectrum with central charge $\arg(\widetilde{Z}_{\alpha\beta}) = \vartheta$ if and only if $\mathbf{z} \in C$ is part of an $\alpha\beta$-trajectory in the network $\mathcal{W}^\vartheta$. Remember from §3.4 that each BPS soliton is thus associated with a BPS web in $C_{\mathbf{z}}$, which may be lifted to a detour path $\gamma_{\alpha\beta}$ in the spectral cover $\Sigma$.

The central charge $\widetilde{Z}_{\alpha\beta}$ of the BPS soliton is then obtained by integrating the 1-form $\lambda$ along the open path $\gamma_{\alpha\beta}$, or equivalently, by integrating $d\lambda$ over an associated open special Lagrangian 2-cycle $D_{\alpha\beta}$. Remember that the 2-cycle $D_{\alpha\beta}$ has two boundary components: the open path $\gamma_{\alpha\beta} \subset \Sigma$ and a path $\ell_{\alpha\beta} \subset T_{\mathbf{z}}^* C$.

Allowing logarithmic singularities in the superpotential $\widetilde{W}_{\text{eff}}$ implies that the spectral networks $\mathcal{W}^\vartheta$ may degenerate at special phases, where **saddle trajectories** (starting *and* ending at a branch point) appear. This implies that the soliton spectrum may contain non-trivial BPS solitons that tunnel from a vacuum $\alpha$ to itself. Such **self-solitons** correspond to closed special Lagrangian 2-cycles $D_\alpha \subset T^* C_{\mathbf{z}}$. Moreover, across such a saddle trajectory we may see a change in the soliton spectrum of the 2d theory $T_{\mathbf{z}}$. Indeed, at this locus in $C_{\mathbf{z}}$ there will be two distinct BPS solitons with the same phase.

As an illustration, let us plot the relevant spectral networks $\mathcal{W}^\vartheta$ explicitly in the example of the $\mathbb{P}^1$-model.[27] For simplicity in notation we set $\mu = 1$. We do not need to look at $\vartheta > \pi$ as these networks are simply those with $\vartheta \leq \pi$ with the trajectories running in the opposite direction. Remember that the trajectories of the spectral network $\mathcal{W}^\vartheta$ are found by solving the first order PDE

$$\lambda(\partial_t) \in e^{i\vartheta} \mathbb{R}_{\geq 0} \tag{204}$$

---

[27]More examples of spectral networks for 2d GLSM's can be found for instance in [39] and [49].

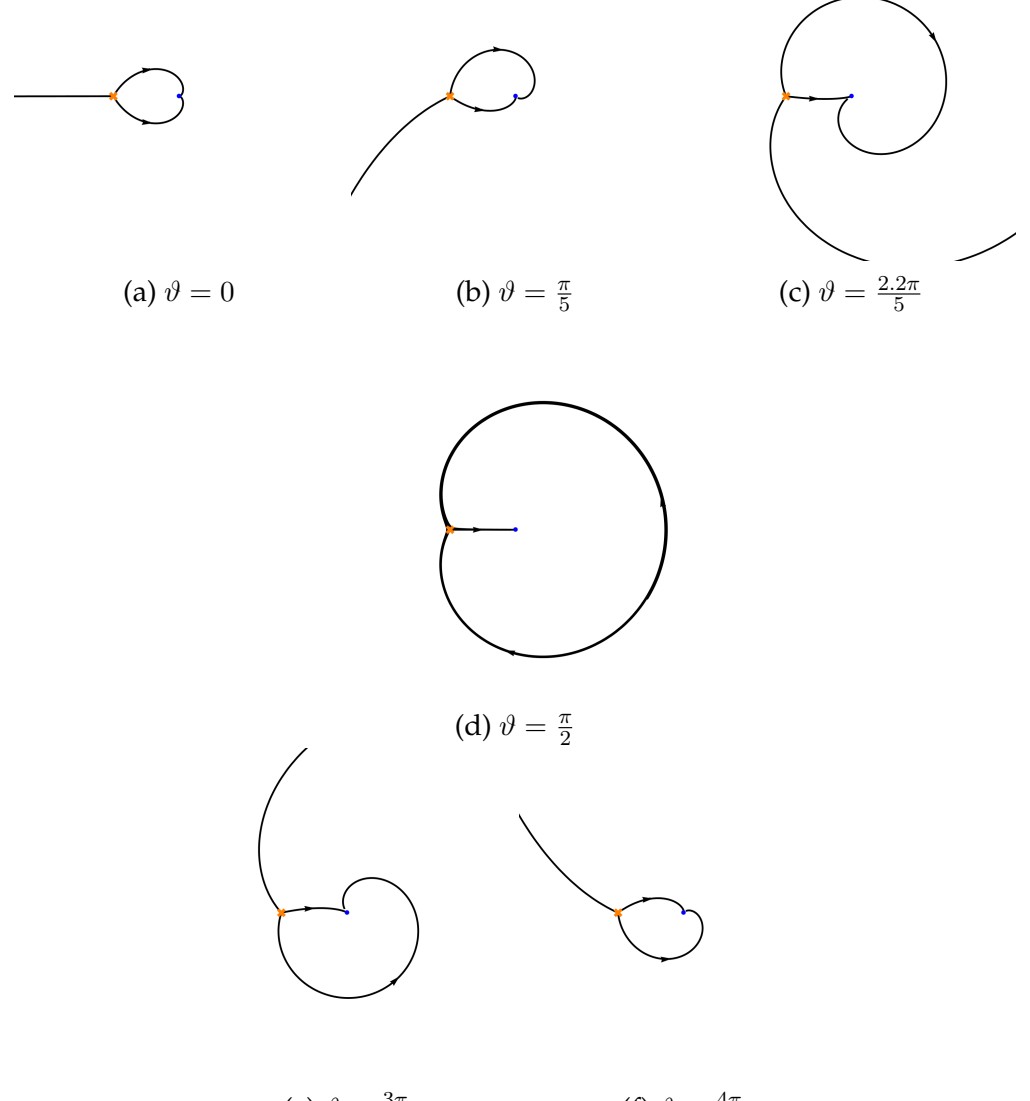

(a) $\vartheta = 0$

(b) $\vartheta = \frac{\pi}{5}$

(c) $\vartheta = \frac{2.2\pi}{5}$

(d) $\vartheta = \frac{\pi}{2}$

(e) $\vartheta = \frac{3\pi}{5}$

(f) $\vartheta = \frac{4\pi}{5}$

Figure 19: Spectral networks for the $\mathbb{P}^1$-model in the $s$-plane (the strong coupling region) with $\mu = \widetilde{m} = 1$. The orange point is the branch point $s = -1$ and the blue point is the singularity $s = 0$.

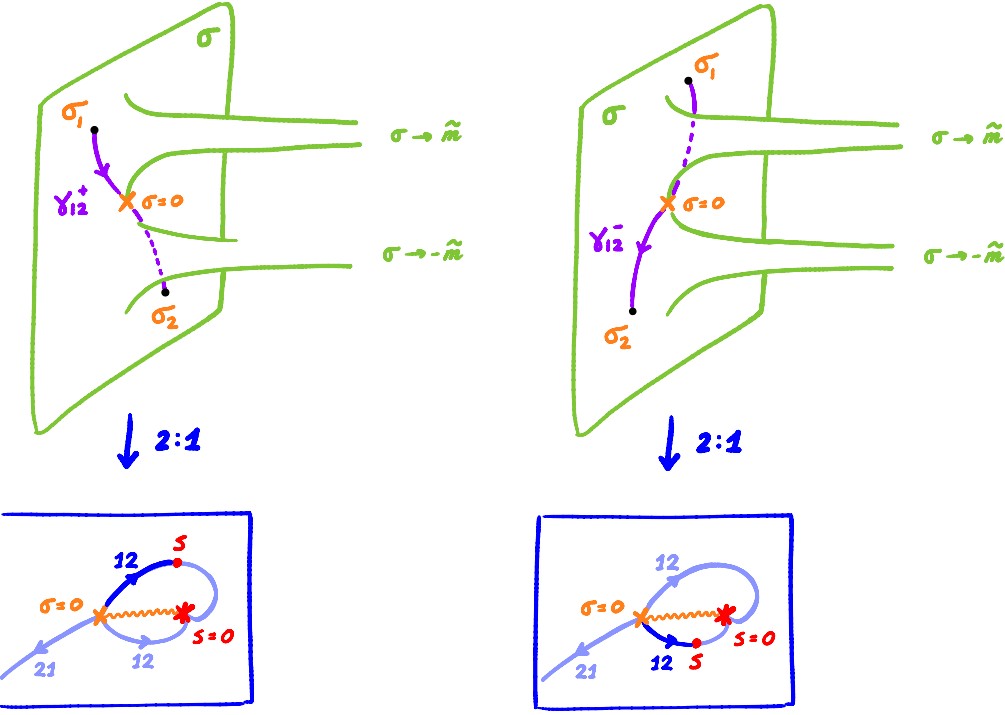

Figure 20: Detour path $\gamma_{12}^{\pm} \subset \Sigma$ for the BPS soliton with central charge $Z_{12}^{\pm}$ on the left/right, respectively.

with the 1-form $\lambda$ for the $\mathbb{P}^1$-model given in equation (187) and $t \in \mathbb{R}$. Note that $\lambda$ has singularities at both $s = \mathbf{z}^{-1} = 0$ and $s = \infty$, and a branch point at $s = -\widetilde{m}^2$.

We start with plotting the networks $\mathcal{W}_{\vartheta}$ in the **strong coupling** region, i.e. the $s$-plane. Figure 19 illustrates how the spectral network changes when $\vartheta$ is varied from 0 to $\pi$. We see that points close to the origin are crossed twice by the network, and that these crossings correspond to two distinct BPS states interpolating between the vacua at $\sigma_1 = \sqrt{s^{-1} + \widetilde{m}^2}$ and $\sigma_2 = -\sqrt{s^{-1} + \widetilde{m}^2}$. Their central charges $\widetilde{Z}_{12}^{\pm}$ can be found by integrating $\lambda$ along the corresponding open paths $\gamma_{12}^{\pm}$, which are sketched in Figures 20 at a generic strong coupling point. This determines

$$\widetilde{Z}_{12}^{\pm} = \frac{1}{2\pi}\left[ -4\sqrt{s^{-1} + \widetilde{m}^2} + 2\widetilde{m} \log\left( \frac{\sqrt{s^{-1} + \widetilde{m}^2} + \widetilde{m}}{\sqrt{s^{-1} + \widetilde{m}^2} - \widetilde{m}} \right) \right] \pm i\widetilde{m}, \qquad (205)$$

with the logarithm in its principal branch.

Next, we plot the networks $\mathcal{W}_{\vartheta}$ in the **weak coupling** region, i.e. the z-plane. Figure 21 illustrates how the spectral network change when $\vartheta$ is varied. We see that points close to the origin of the z-plane are crossed an infinite number of times by the network as it coils and uncoils around the origin. This corresponds to an infinite

number of BPS solitons interpolating between $\sigma_1$ and $\sigma_2$. Their central charges are

$$\widetilde{Z}_{12}^k = \frac{1}{2\pi} \left[ -4\sqrt{\mathbf{z} + \widetilde{m}^2} + 2\widetilde{m} \log \left( \frac{\sqrt{\mathbf{z} + \widetilde{m}^2} + \widetilde{m}}{\sqrt{\mathbf{z} + \widetilde{m}^2} - \widetilde{m}} \right) \right] + i\widetilde{m}(2k+1) \tag{206}$$

with $k \in \mathbb{Z}$ and the logarithm in its principal branch. See the left picture in Figure 22 for a sketch of the relevant contours for the $k = 2$ soliton at a generic weak coupling point.

At $\vartheta = \frac{\pi}{2}$ we see yet a new feature. At this phase there is an additional family of closed trajectories that go through every point in the **ring domain** enclosed by the saddle connection in Figure 19 (d) and Figure 21 (d). This family indicates the presence of two self-solitons interpolating between the vacuum $\sigma_\alpha$ and itself!

Indeed, Figure 22 sketches the detour paths $\gamma_1^+$ and $\gamma_2^-$ associated to the saddle trajectory. The detour path $\gamma_1^+$ is obtained as the concatenation

$$\gamma_1^+ = \gamma_{12}^+ \circ \gamma_{21}^- \tag{207}$$

of the lift $\gamma_{12}^+$ of the 12-trajectory, starting at the branch-point and ending at the point $\mathbf{z}$, and the lift $\gamma_{21}^-$ of the 21-trajectory, also starting at the branch-point and ending at the point $\mathbf{z}$, but going in the other way around the puncture. The detour path $\gamma_2^-$ is similarly obtained as $\gamma_2^- = \gamma_{21}^- \circ \gamma_{12}^+$. The central charge of these self-solitons is thus given by

$$\widetilde{Z}_\alpha^\pm = \pm 2i \, (-1)^{\alpha-1} \, \widetilde{m}. \tag{208}$$

The BPS spectrum is clearly different at strong and weak coupling, and we conclude that there must be a wall of marginal stability in $C$ where the spectrum jumps. By inspecting the spectral network $\mathcal{W}_\vartheta$ at strong and weak coupling one concludes that this is precisely the ring shaped saddle connection depicted in Figures 19(d) and 21(d). This can be explained as follows. The wall of marginal stability is the maximal locus where

$$\arg(\widetilde{Z}_{12}^+) = \arg(\widetilde{Z}_{12}^-) + \pi = \arg(\widetilde{Z}_{21}^-) = \arg(\widetilde{Z}_{21}^+) + \pi = \frac{\pi}{2}. \tag{209}$$

This implies that the central charges $\widetilde{Z}_1^+ = \widetilde{Z}_{12}^+ + \widetilde{Z}_{21}^- = \widetilde{Z}_2^-$ and $\widetilde{Z}_{12}^k = \widetilde{Z}_{12}^+ + k(\widetilde{Z}_{12}^+ + \widetilde{Z}_{21}^-)$ all have argument $\frac{\pi}{2}$. Hence, as explained in §3.3, the corresponding bound states of BPS solitons may form at this phase and at this locus of $C$.

### 4.3.3 Remark: Exponential networks

In the previous subsection we viewed the spectral curve $\Sigma$ as a branched covering over the Higgs branch (parametrised by $s = \mathbf{z}^{-1}$) with tautological 1-form

$$\lambda_s = \frac{\sigma}{2\pi} \frac{ds}{s}. \tag{210}$$

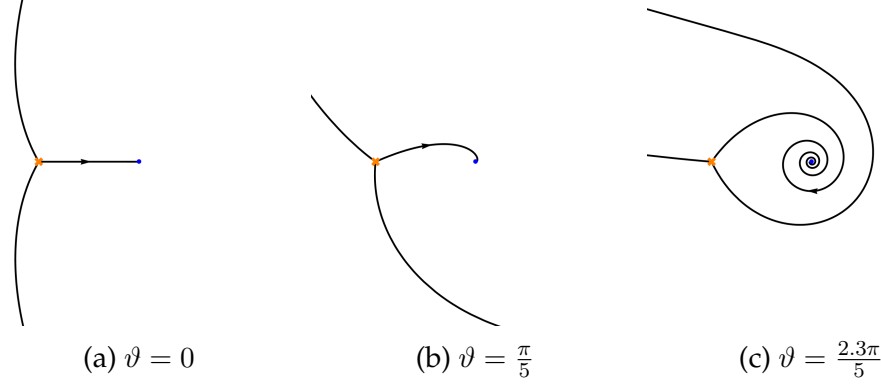

(a) $\vartheta = 0$  (b) $\vartheta = \frac{\pi}{5}$  (c) $\vartheta = \frac{2.3\pi}{5}$

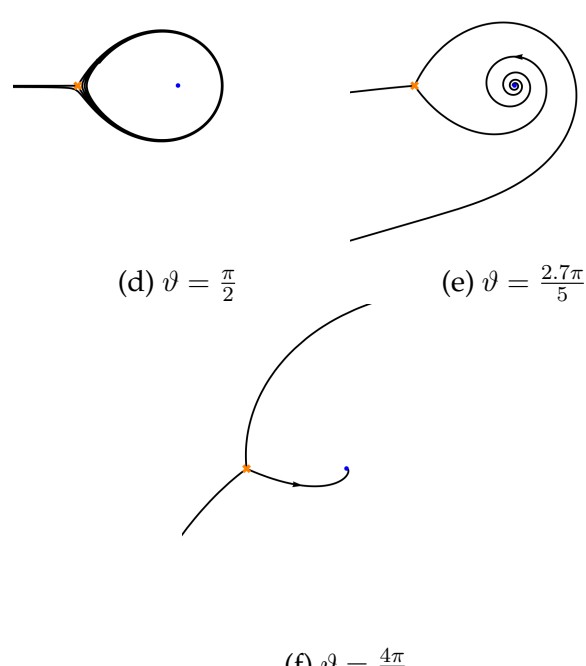

(d) $\vartheta = \frac{\pi}{2}$  (e) $\vartheta = \frac{2.7\pi}{5}$

(f) $\vartheta = \frac{4\pi}{5}$

Figure 21: Spectral networks for the $\mathbb{P}^1$-model in the z-plane (the weak coupling region) with $\mu = \widetilde{m} = 1$. The orange point is the branch point $z = -1$ and the blue point is the singularity $z = 0$.

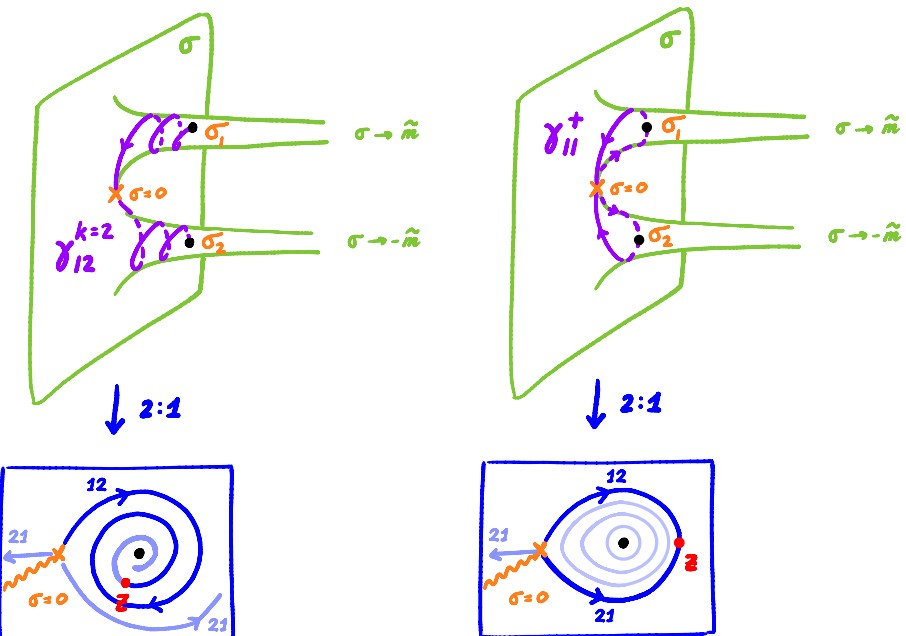

Figure 22: Left: detour path $\gamma_{12}^{k=2} \subset \Sigma$ for BPS soliton with central charge $\widetilde{Z}_{12}^{k=2}$. Right: detour path $\gamma_1^+$ (starting and ending at $\sigma_1$) for the BPS soliton with central charge $\widetilde{Z}_1^+$. (The detour path $\gamma_2^-$ is the same path in homology, but now starting and ending at $\sigma_2$.)

Instead, one might also view $\Sigma$ as a branched cover over the Coulomb branch (parametrised by $\sigma$) with tautological 1-form

$$\lambda_\sigma = \frac{\log s}{2\pi}\, d\sigma. \tag{211}$$

The logarithm in $\lambda_\sigma$ suggests that it is helpful to consider the universal covering $\widetilde{\Sigma}$ of $\Sigma$. If we choose a trivialization for this universal covering (i.e. a choice of logarithmic branch cuts), BPS solitons can be encoded in trajectories defined by

$$(\log s_\alpha - \log s_\beta + 2\pi i n)\frac{d\sigma}{dt} \in e^{i\vartheta}\, \mathbb{R}_+, \tag{212}$$

where the extra integer $n$ originates in the multi-valued-ness of the logarithms. These trajectories are therefore labelled by the tuple $(\alpha, \beta; n)$. The corresponding structure is called an **exponential network** (see for more details [50, 51] and follow-ups).

### 4.3.4 Embedding GLSM's in M-theory

GLSM's can be embedded in M-theory using a collection of M2 and M5-branes [48]. In M-theory physical properties of the 2d theory get translated into geometric properties of the extended branes. Furthermore, string theoretic dualities can be employed

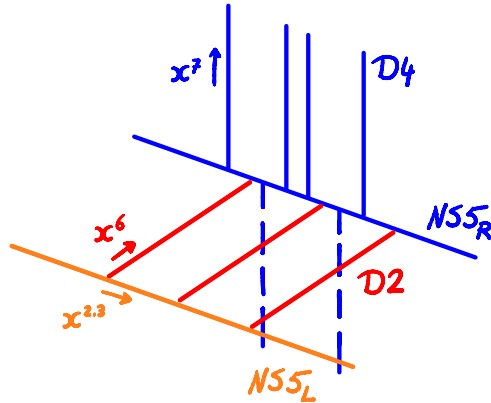

Figure 23: Brane configuration that embeds a 2d GLSM in string theory.

to then relate the 2d theory to other theories and set-ups. In this section we give an introduction to the embedding of the $\mathbb{P}^{N-1}$-model in M-theory and show that its BPS solitons can be realized as open M2-branes.

Before explaining the M-theory set-up though, let us start in string theory. Consider the type IIA background $\mathbb{R}^{10}$ with $\kappa$ dynamical D2-branes stretched between two NS5-branes $\text{NS5}_L$ and $\text{NS5}_R$. The $\text{NS5}_L$-brane is placed at

$$x^{012345} = \text{free}, \quad x^{6789} = 0, \tag{213}$$

while the $\text{NS5}_R$-brane is placed at

$$x^{012389} = \text{free}, \quad x^{45} = 0, \quad x^6 = \frac{1}{e^2}\frac{g_{\text{st}}}{l_{\text{st}}}, \quad x^7 = -r\,g_{\text{st}}l_{\text{st}}, \tag{214}$$

where we identify $x^2 + ix^3 = l_{\text{st}}^2\sigma$. The string length $l_{\text{st}}$ and string coupling $g_{\text{st}}$ factors are just inserted in the above formulae on dimensional grounds, most important is how the field theory paramaters $\sigma$, $e$ and $r$ are embedded in the geometry.

The D2-branes end on the NS5-branes with $x^{01}$ free. See Figure 23. The total brane system then preserves 4 supersymmetries. The (low energy) worldvolume theory on the D2-branes in the $x^{01}$-directions is a 2d $\mathcal{N} = (2,2)$ gauge theory with gauge group $U(\kappa)$. The rotational symmetry $U(1)_{23}$ may be identified with its $U(1)_A$ R-symmetry, and the $x^{23}$-directions parametrize its Coulomb branch.

The chiral fields $\Phi_j$, that transform in the fundamental representation of the gauge group, can be introduced by inserting additional $\text{D4}_j$-branes at

$$x^{01789} = \text{free}, \quad x^2 + ix^3 = l_{\text{st}}^2\,\widetilde{m}_j, \quad x^{45} = 0, \quad x^6 = \frac{1}{e^2}\frac{g_{\text{st}}}{l_{\text{st}}} = L, \tag{215}$$

while ending from above on the $\text{NS5}_R$-brane in the $x^7$-direction. See again Figure 23. Each chiral field $\Phi_j$ originates from an open string stretching between the D2 and the $\text{D4}_j$-brane.

The rotational symmetry $U(1)_{89}$, which the $\text{NS5}_R$ brane and the D4 branes have in common, may be identified with the $U(1)_V$ R-symmetry, while the movement

of the D2-branes along with the D4-branes in the $x^7$-direction parametrizes the Higgs branch.[28] Additional chiral fields $\widetilde{\Phi}_j$, transforming in the anti-fundamental representation of the gauge group, may be introduced as D4-branes ending from below on the NS5$_R$-brane.

The quantum features of the GLSM become apparent when lifting the brane setup to M-theory. We thus consider the M-theory background $\mathbb{R}^{10} \times S_R^1$ with metric

$$ds^2 = -(dx^0)^2 + \sum_{i=1}^{9}(dx^i)^2 + R^2\,(dx^{10})^2, \tag{216}$$

where $x^{10}$ is the periodic coordinate on the M-theory circle $S_R^1$ of radius R.

In M-theory, the NS5$_L$-brane lifts to a flat M5$_L$-brane placed at

$$x^{012345} = \text{free}, \quad x^{678910} = 0, \tag{217}$$

while the D4-branes and the NS5$_R$-brane combine into an M5-brane M5$_R$, that wraps a Riemann surface $\Sigma_M$ embedded in the directions $x^{23710}$ . If we introduce the complex coordinate

$$-\hat{t} = R^{-1}x^7 + ix^{10}, \tag{218}$$

and define $\hat{s} = e^{-\hat{t}}$, the M-theory curve $\Sigma_M$ is embedded in $\mathbb{C}_\sigma \times \mathbb{C}_{\hat{s}}^*$ through the equation

$$\Sigma_M: \quad \prod_{j=1}^{N}(\sigma - \widetilde{m}_j) = q\hat{s}^{-1}, \tag{219}$$

where $q$ is a new M-theory parameter. This equation may be derived by analysing how the NS5-brane bends when ending on a D4-brane [52].

The dynamical D2-branes lift to dynamical M2-branes stretched between the two fixed M5 branes M5$_L$ and M5$_R$. Note that the $x^7$-position of the M5$_R$-brane is not fixed anymore (as was the case for the $x^7$-position of the NS5$_R$-brane), but varies a function of $\sigma$. Because the $x^7$-coordinate is proportional to $r$, this implies that M-theory setup naturally encodes the running of the FI parameter! The same is true for the $x^{10}$-coordinate, which may be interpreted as the effective theta-angle of the field theory. More precisely, the relation to the 2d GLSM will come through the identifications

$$q = \mu^N e^{2\pi i\tau}, \quad \text{and} \quad \hat{t} = 2\pi i\,\tau_{\text{eff}}(\sigma), \tag{220}$$

where $\tau_{\text{eff}}(\sigma)$ was defined in equation (180).

Vacua are given by M2 configurations extending only in $x^{016}$ as $\mathbb{R}_{01}^2 \times I$, where the interval $I$ stretches between $x^6 = 0$ and $x^6 = L$, and ending on common points of the two M5-branes in the transverse coordinates. These common points are given by:

---

[28] Each D2-brane will need to end on a different D4-brane to avoid so-called s-configurations.

- $x^{4578910} = 0 \implies \hat{t} = 0 \implies \hat{s} = 1$,

- $x^{23}$ must be solutions of the equation

$$\Sigma : \quad \prod_{j=1}^{N} (\sigma - \widetilde{m}_j) = \mu^N e^{2\pi i \tau}. \tag{221}$$

This implies that there are $N$ vacua, whose description precisely matches the field theory description.

Now is a good time to emphasise certain distinctions. Since $s$ is different from $\hat{s}$, the M-theory curve $\Sigma_M$ is not the spectral curve $\Sigma$ of the GLSM. When thinking about quantities in M-theory, we treat $q = \mu^N s$ as a parameter and $\hat{s}$ as a coordinate, and when thinking about quantities in the field theory we think of $s$ as a coordinate in the spectral geometry and $\hat{s}$ as a parameter controlled by the vev of $\sigma$ (through equation (220)). Because $s$ and $\hat{s}$ enter $\Sigma_M$ as the product $s\hat{s}$, it is easy to relate the M-theory geometry (where $s$ is fixed) to the spectral geometry (where $\hat{s}$ is fixed) by exchanging $s$ and $\hat{s}$; we will do this in §4.3.5.

Note that $\Sigma_M$ is naturally embedded in $T_\sigma^* \times \mathbb{C}_{\hat{s}}^* \cong \mathbb{C}_\sigma \times \mathbb{C}_{\hat{s}}^*$ with holomorphic symplectic form

$$\widehat{\Omega} = \frac{1}{2\pi} \, d\sigma \wedge \frac{d\hat{s}}{\hat{s}} = d\hat{\lambda}. \tag{222}$$

If we consider $\Sigma_M$ from the field theory perspective instead (with $\hat{s}$ fixed and $s$ as a coordinate), we may identify $\hat{\lambda}$ with $\lambda$.

### 4.3.5 Solitons as open M2-branes

GLSM-solitons can be embedded in M-theory as open M2-brane configurations which are constant in time $x^0$ and which interpolate between vacua $\alpha$ and $\beta$ at $\hat{s} = 1$. This implies that the M2-branes have world-volume $\mathbb{R}_{x^0} \times S_{\alpha\beta}$, where

$$S_{\alpha\beta} \subset I \times \mathbb{C}_\sigma \times \mathbb{C}_{\hat{s}}^* \cong I \times T^* \mathbb{C}_{\hat{s}}^* \quad \text{with } x^{4589} = 0 \tag{223}$$

is a two-dimensional surface. We label the segment $0 \le x^6 \le L$ located at $\sigma = \sigma_a$ and $\hat{s} = 1$ (as well as $x^{4589} = 0$) by $I_\alpha$.

The surface $S_{\alpha\beta}$, or just $S$, has four boundary components, with the properties

$$\begin{aligned} \partial S_{x^1 \to -\infty} &\simeq I_\alpha, & J_L &= \partial S_{x^6=0} \subset T_{\hat{s}=1}^* \mathbb{C}_{\hat{s}}^*, \\ \partial S_{x^1 \to +\infty} &\simeq I_\beta, & J_R &= \partial S_{x^6=L} \subset \Sigma_M, \end{aligned} \tag{224}$$

where $J_L$ and $J_R$ are embedded in the M5$_L$-brane and M5$_R$-brane, respectively. The boundary component $J_L$ is embedded in the fiber $T_{\hat{s}=1}^* \mathbb{C}_{\hat{s}}^*$, since the M5$_L$-brane is placed at $\hat{s} = 1$. The end-points of the interval $S|_{x^6=\text{fixed}}$, and thus in particular of the boundary component $J_R$, remain at the vacua $\sigma_\alpha$ and $\sigma_\beta$ (in the same fiber $T_{\hat{s}=1}^* \mathbb{C}_{\hat{s}}^*$) when varying $0 \le x^6 \le L$.

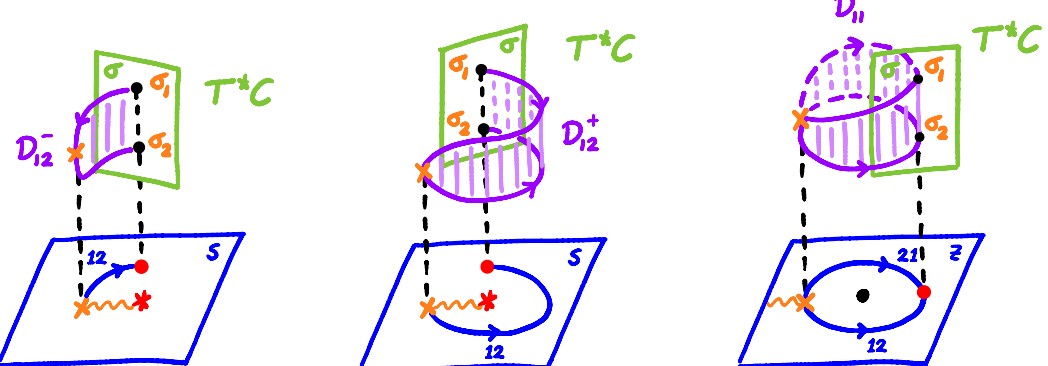

Figure 24: Open and closed discs corresponding to the BPS solitons with central charge $\widetilde{Z}_{12}^{\pm}$ and $\widetilde{Z}_1^+$ in the $\mathbb{P}^1$-model. The fuchsia boundary component of each disc corresponds to the 1-cycle $\gamma_{\alpha\beta} \subset \Sigma$, whereas the dashed black boundary component is embedded in the fiber $T_s^*\mathbb{C}^*$.

The surface $S_{\alpha\beta}$ thus projects to an open 2-cycle $\widehat{D}_{\alpha\beta}$ in $T^*\mathbb{C}_{\hat{s}}^*$, which has vertices at $\sigma_\alpha$, $\sigma_\beta \in T_{\hat{s}=1}^*\mathbb{C}^*$ and boundary components given by $J_L \subset T_{\hat{s}=1}^*\mathbb{C}^*$ and $J_R \subset \Sigma_M$, respectively. This should remind the reader of the open 2-cycles $D_{\alpha\beta}$ discussed in §3.4. Indeed, for the open M2-brane configuration to be BPS, the boundary $J_R$ will need to end on the M5$_R$-brane along a path $\hat{\gamma}_{\alpha\beta} \subset \Sigma_M$ such that

$$\mathrm{Im}\left(e^{-i\arg(Z_{\alpha\beta})}\,\hat{\lambda}\left(v\right)\right) = 0, \tag{225}$$

for any non-zero tangent vector $v$ to $\hat{\gamma}_{\alpha\beta}$ [53]. This implies that the 2-cycle $\widehat{D}_{\alpha\beta}$ should be special Lagrangian, with symplectic volume

$$\widetilde{Z}_{\alpha\beta} = \int_{\widehat{D}_{\alpha\beta}} d\hat{\lambda}. \tag{226}$$

In fact, we may view $\widehat{D}_{\alpha\beta}$ as an open 2-cycle $D_{\alpha\beta}$ embedded in $T^*\mathbb{C}_s^*$, by considering $s$ to be dynamical instead of $\hat{s}$. In other words, the special Lagrangian 2-cycles $D_{\alpha\beta}$ from §3.4 may be embedded in M-theory as open M2-branes!

As an example, we have drawn a few M2-branes wrapping special Lagrangian discs $\widehat{D}_{\alpha\beta} \cong D_{\alpha\beta}$ in the $\mathbb{P}^1$-model in Figure 24. Note that the solitons with central charge $\widetilde{Z}_{12}^{\pm}$ correspond to open discs $D_{12}^{\pm}$, respectively, whereas self-solitons with central charge $\widetilde{Z}_{\alpha\alpha}^{\pm}$ correspond to closed discs $D_{\alpha\alpha}^{\pm}$.[29] It is also entertaining to imagine how these 2-cycles are realised in the spectral geometry illustrated (for instance) in Figures 20 and 22. This is sketched in Figure 25.

---

[29]We note that the analysis here, using spectral network techniques, improves the analysis of the open M2-branes in §6 of [48]. In particular, the spectral network analysis leads to the full spectrum in both the weak and the strong coupling region, and in particular to the correct boundaries $\gamma_{\alpha\beta}$ of the open M2-branes in the strong coupling region.

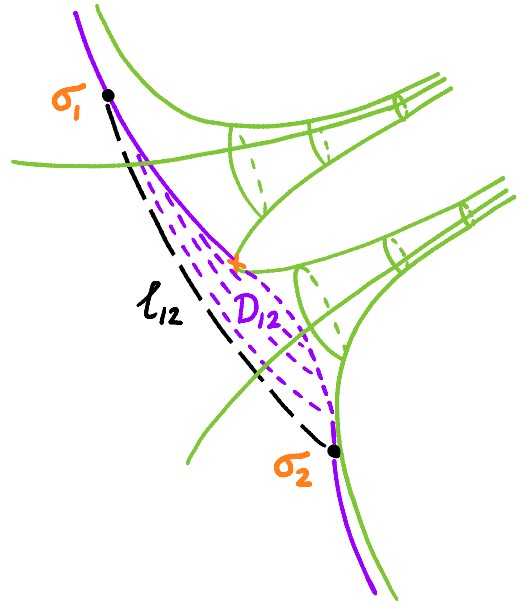

Figure 25: Sketch of an open disc embedded in the spectral geometry. The fuchsia boundary component $\gamma_{12}$ is embedded in the spectral curve $\Sigma$, whereas the dashed black boundary component $\ell_{12}$ is embedded in the fiber $T_s^*\mathbb{C}^*$. Note that pairs of points on $\gamma_{12}$ are connected by lines in the fibers of $T^*\mathbb{C}^*$.

The central charges of these BPS solitons were computed in equations (205) to (209) by integrating $\lambda$ over the boundary components $\gamma_{\alpha\beta}$. That is

$$\widetilde{Z}_{\alpha\beta} = \int_{\gamma_{\alpha\beta}} \lambda \tag{227}$$

In particular, by equating the result to the field theory expression (201), we saw that this resolves the log-ambiguities in the field theory expression and determines uniquely the $U(1)$-charges $n_j$. Here we want to return to this argument, in the generality of the $\mathbb{P}^{N-1}$-model, to show that the $U(1)$-charges can be interpreted in terms of open strings.

Remember that the central charge $\widetilde{Z}_{\alpha\beta}$ in field theory is given by the expression

$$\widetilde{Z}_{\alpha\beta} = 4\left(\widetilde{W}(\sigma_\beta) - \widetilde{W}(\sigma_\alpha)\right) + i\sum_{j=1}^{N} \widetilde{m}_j n_j \tag{228}$$

$$= \frac{1}{2\pi}\left[N(\sigma_\beta - \sigma_\alpha) + \sum_{j=1}^{N} \widetilde{m}_j \log\left(\frac{\sigma_\beta - \widetilde{m}_j}{\sigma_\alpha - \widetilde{m}_j}\right)\right] + i\sum_{j=1}^{N} \widetilde{m}_j n_j, \tag{229}$$

where we assume that the logarithm is in its principal branch. Note that we can bring the spectral geometry expression (227) in the above form by splitting the integral (227) over $\gamma_{\alpha\beta}$ as an integral over a fixed open 1-cycle $\Gamma_{\alpha\beta} \subset \Sigma$ (with boundaries at $\sigma_\alpha$ and

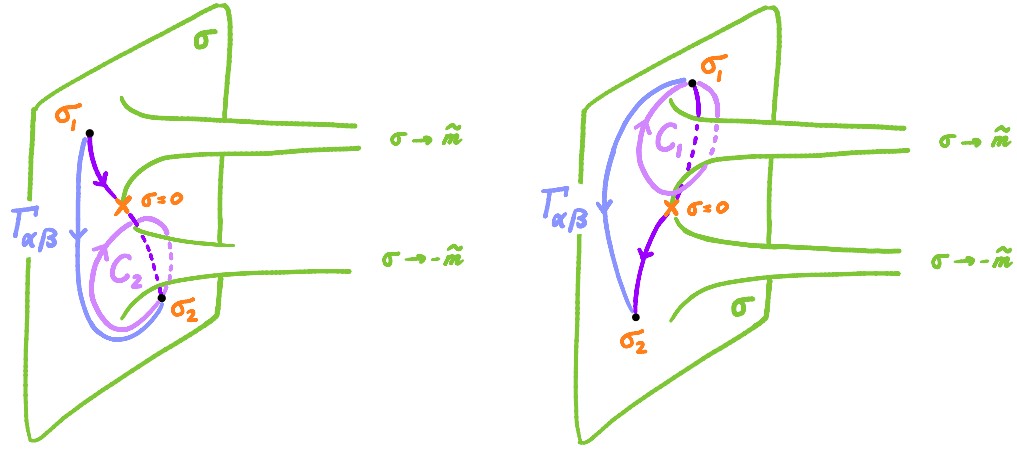

Figure 26: All detour paths $\gamma_{\alpha\beta}$ can be decomposed (in homology) as a sum of a fixed open 1-cycle $\Gamma_{\alpha\beta}$ and a linear combination of closed 1-cycles $C_j$. This is illustrated on the left/right for the detour paths $\gamma_{12}^{\pm}$.

$\sigma_\beta$) plus an integral over a linear combination of integrals over closed 1-cycles $C_j \subset \Sigma$, such that

$$\int_{\Gamma_{\alpha\beta}} \lambda = \frac{1}{2\pi}\left[N(\sigma_\beta - \sigma_\alpha) + \sum_{j=1}^N \widetilde{m}_j \log\left(\frac{\sigma_\beta - \widetilde{m}_j}{\sigma_\alpha - \widetilde{m}_j}\right)\right],$$
$$\int_{C_j} \lambda = i\widetilde{m}_j,$$

(230)

with the logarithm again in its principal branch. That is, we write

$$\widetilde{Z}_{\alpha\beta} = \int_{\Gamma_{\alpha\beta}} \lambda + \sum_{j=1}^N n_j \int_{C_j} \lambda.$$

(231)

The cycles $\Gamma_{\alpha\beta}$ and $C_j$ are illustrated in Figure 26 in the $N = 2$ example.

More generally, in the strong coupling region of the $\mathbb{P}^{N-1}$-model, BPS solitons that interpolate between adjacent vacua can have $U(1)$-charge $n_\alpha = 1$, for any single $1 \leq \alpha \leq N$, and all other $U(1)$-charges $n_{\beta \neq \alpha} = 0$. They may therefore be labelled by a single generator $C_\alpha$ of the first homology of the spectral curve $\Sigma$. Solitons that tunnel between arbitrary vacua $\sigma_\alpha$ and $\sigma_\beta$ are instead labelled by

$$K = |\beta - \alpha| \bmod N$$

(232)

distinct boundary circles $C_k$. Note that this implies that there are $\binom{N}{K}$ such solitons.

The decomposition (231) has a neat interpretation in terms of string theory. It suggests that the open M2-branes wrapping the 2-cycles $D_{\alpha\beta}$ should be interpreted as lifts to M-theory of the combined system of D2-brane and open D2-D4 strings,

just as the D4 and NS5-branes merge into a single M5-brane. Whereas the D2-brane boundary lifts to the open 1-cycle $\Gamma_{\alpha\beta}$, the boundary of the any open string connecting the D2-brane to the $\alpha$th D4-brane lifts to the closed 1-cycle $C_\alpha$.

In particular, this interpretation shows that the each soliton interpolating between adjacent vacua may be interpreted as an open fundamental string, connecting the D2-brane to the $\alpha$th D4-brane, which in turn generates the chiral multiplet $\Phi_\alpha$ in the fundamental representation of $SU(N)$. This is indeed consistent with the field theory analysis in [54]. Arbitrary $\alpha\beta$-solitons with the same number $K$ instead assemble into antisymmetric tensor representations $\bigwedge^K \mathbb{C}^N$ of $SU(N)$.

## 4.4 Higgs and Coulomb branch partition functions

In this section we introduce the two-dimensional $\Omega$-background and calculate the vortex partition function introduced in §4.1 through equivariant localisation. More precisely, we will refer to the complete partition function in the two-dimensional $\Omega$-background as the Higgs branch partition function, and to its non-perturbative contribution in $\tau$ as the vortex partition function. We will spell out the computation of the Higgs branch partition function in two examples, and describe in which sense the $\Omega$-background quantizes the spectral geometry. As a supplement, we will introduce a Coulomb branch partition function and show how it is related to the Higgs branch partition function through a Fourier transform.

### 4.4.1 $\Omega$-background and the vortex partition function

Computing the vortex partition function

$$\mathcal{Z}^{\text{vortex}}(\mathbf{z}) = \sum_{\mathfrak{m}} \mathbf{z}^{\mathfrak{m}} \oint_{\mathcal{M}_{\text{vortex},\mathfrak{m}}} 1, \tag{233}$$

is not easy. Just like in four dimensions, we need to introduce an additional ingredient: the $\Omega$-**background**. Parallel to Nikita's explanation of the four-dimensional $\Omega$-background, we define the two-dimensional $\Omega$-background starting with a four-dimensional background $M_4$, which is a fibration

$$\mathbb{C}_z \hookrightarrow M_4 \twoheadrightarrow T^2_w \tag{234}$$

of the two-dimensional space-time $\mathbb{C}_z$ (with complex coordinate $z$) over an auxiliary torus $T^2_w$ (with complex coordinate $w$ and complex structure $\tau = i$).

More precisely, if we go around the circle $S^1_{\text{Re}(w)}$ of the torus, i.e. $w \mapsto w + 1$, we want our space-time to rotate as

$$z \mapsto \exp\left(i\text{Re}(\epsilon)\right) z, \tag{235}$$

whereas if we go around the $S^1_{\text{Im}(w)}$ circle, i.e. $w \mapsto w + i$, it should rotate as

$$z \mapsto \exp\left(-i\text{Im}(\epsilon)\right) z. \tag{236}$$

The corresponding metric of the four-dimensional background is given by

$$ds^2 = |dz - iz(\epsilon dw + \bar{\epsilon}d\overline{w})|^2 + |dw|^2. \tag{237}$$

The two-dimensional $\Omega$-background is then obtained by dimensional reduction of this four-dimensional background along the periodic directions. The advantage of such a reduction is twofold: one, it preserves two out of the four supercharges of the original theory, and two, it effectively **localises** the vortex dynamics to the origin of the two-dimensional space-time $\mathbb{C}_z$, thus resolving IR divergences that arise because the vortices can run off to infinity. Let us see how this works.

To preserve half of the supersymmetry in the four-dimensional background (237) we will need to turn on a specific background gauge field for the vector R-symmetry. This turns out to be equivalent to considering the theory in the A-twist (defined in §2.2.2). The supercharges $Q_-$ and $\overline{Q}_+$ thus transform as scalars and are conserved, while the supercharges $\overline{Q}_-$ and $Q_+$ can be promoted to a holomorphic and an anti-holomorphic one-form, respectively, which can be combined into the one-form

$$G = \overline{Q}_- dz + Q_+ d\bar{z}. \tag{238}$$

Turning on the $\epsilon$-parameter deforms the conserved supercharge $Q_A$ into[30]

$$Q_A^\epsilon = Q_A + \epsilon \, \iota_V G, \tag{239}$$

where $V = i(z\partial_z - \bar{z}\partial_{\bar{z}})$ is the 2d rotation generator. The deformed supercharge $Q_A^\epsilon$ is no longer nilpotent but squares to the Lie derivative along the vector field $\epsilon V$. In formulae,

$$(Q_A^\epsilon)^2 = i\epsilon(zP_z - \bar{z}P_{\bar{z}}). \tag{240}$$

Observables of the resulting 2d theory are those operators that are part of the $Q_A^\epsilon$-cohomology. Equation (240) implies that observables must be invariant under the rotation generated by $\epsilon V$. This effectively constrains the theory to the origin of $\mathbb{C}_z$ for non-zero $\epsilon$. At the origin we have that $z = \bar{z} = 0$, so that the $\epsilon$-dependent terms vanish and $Q_A^\epsilon|_{z=0} = Q_A$. As a result, the partition function in the 2d $\Omega$-background localises to vortex configurations constrained to the origin of $\mathbb{C}_z$.

Mathematically, this implies that the vortex partition function in the $\Omega$-background can be computed using **equivariant localisation** techniques. Calculating the vortex partition function turns equivalent to computing the equivariant volume of the vortex moduli space, with respect to the $\mathbb{C}^*$-action

$$z \mapsto \exp(i\epsilon)z. \tag{241}$$

Explicit expressions can be found using the available descriptions of the vortex moduli space as a symplectic quotient.[31] This was the strategy of the original study [55], and is concisely summarized in for instance the Appendix G of [56].

---

[30]We could have equivalently deformed the conserved supercharge $Q_A^\xi$ into $Q_A^\xi + \epsilon \, \iota_V G^\xi$, for any other phase $\xi$.

[31]This description is analogous to the ADHM description of the instanton moduli space in four dimensions.

To illustrate an equivariant localisation computation, let us find the equivariant volume of $\mathbb{C}$ with respect to the $\mathbb{C}^\times$−action (241). Remember that given a group action $G$ on a symplectic manifold $(M, \omega)$, the equivariant volume with respect to this action is defined as

$$\text{vol}_G(M) = \int e^{\omega - H}, \tag{242}$$

where $H$ is the Hamiltonian function for the group action. Consider the standard symplectic form $\omega = \frac{i}{2} dz \wedge d\bar{z}$ on $\mathbb{C}$. We can compute the Hamiltonian function of the $\mathbb{C}^\times$−action using

$$\iota_{\epsilon V} \omega = -dH_\epsilon$$

where $V$ is the 2d rotation generator. This turns out to be

$$H_\epsilon = \frac{1}{2} \epsilon |z|^2. \tag{243}$$

It is then straight-forward to evaluate the integral (242) on $\mathbb{C}$, resulting in the equivariant volume

$$\text{vol}_\epsilon(\mathbb{C}) = \int e^{\omega - H_\epsilon} = \frac{i}{2} \int e^{-\frac{1}{2} \epsilon |z|^2} dz d\bar{z} = \frac{2\pi}{\epsilon}. \tag{244}$$

Computations like these play an important role when calculating vortex partition functions.

Note that the equivariant volume (244) of $\mathbb{C}$ is finite when $\epsilon \neq 0$. This means that the $\Omega$-deformation parameter $\epsilon$ acts as a regulator. In the limit $\epsilon \to 0$ we may recover non-trivial information about the original gauge theory.

In addition to its role as a regulator, the parameter $\epsilon$ has an interpretation as a quantization parameter. This will become clear in the following examples, where one of the main observations will be that the vortex partition function $\mathcal{Z}^{\text{vortex}}$ is annihilated by a differential operator $d_\epsilon$, which **quantizes** the twisted ring equation. This differential operator $d_\epsilon$ is sometimes called a quantum curve.

**Remark:** The $\Omega$-background can also be studied in the B-twist. Given any scalar supercharge $Q_B^\xi$ we can construct the BRST operator

$$Q_\epsilon^\xi = Q_B^\xi + \epsilon \, \iota_V G^\xi, \tag{245}$$

in the $\Omega$-deformed theory, where $G^\xi = G_z dz + \xi^{-1} G_{\bar{z}} d\bar{z}$ and $V$ is a Killing vector field. This construction is different though from the four-dimensional reduction described in this section. See for instance [57].

### 4.4.2 Example: vortex partition function for the abelian Higgs model

An elementary example is the **abelian Higgs model**. This is the two-dimensional $U(1)$-theory coupled to a single massless chiral multiplet, i.e. the massless $\mathbb{P}^0$-model.[32] We read off from equation (171) that its spectral curve $\Sigma$ is cut out by

---

[32]Here we follow the discussion in [58].

the equation

$$\Sigma: \quad \sigma = \mathbf{z}, \tag{246}$$

where $\mathbf{z} = e^{2\pi i \tau}$. That is, the spectral curve $\Sigma$ is a single-sheeted covering over the parameter space $C = \mathbb{C}_{\mathbf{z}}$.

The moduli space of $\mathfrak{m}$ vortices on $\mathbb{C}$ in the abelian Higgs model is simply parametrized by their positions,

$$\mathcal{M}_{\text{vortex},\mathfrak{m}} = \mathbb{C}^{\mathfrak{m}}/S_{\mathfrak{m}}, \tag{247}$$

where the quotient by $S_{\mathfrak{m}}$ reflects the fact that the vortices are indistinguishable. The $\Omega$-background acts as a rotation on each $\mathbb{C}$-factor in the product.

The vortex partition function of the abelian Higgs model can then be computed to be

$$\mathcal{Z}^{\text{vortex}}(\mathbf{z}, \epsilon) = \sum_{\mathfrak{m}} \mathbf{z}^{\mathfrak{m}} \frac{1}{\epsilon^{\mathfrak{m}} \, \mathfrak{m}!} = \exp\left(\frac{\mathbf{z}}{\epsilon}\right), \tag{248}$$

because $2\pi r > 0$.

Nikita explained to us how the four-dimensional $\Omega$-background quantizes the Seiberg-Witten geometry. In two dimensions something similar happens: the two-dimensional $\Omega$-background quantizes the twisted chiral ring equation. Let us explain this in more detail.

Remember that the twisted chiral ring

$$\sigma - e^{2\pi i \tau} = 0 \tag{249}$$

defines a spectral curve inside $T_{\sigma}^* \mathbb{C}_{\tau}$, with respect to the holomorphic symplectic form

$$d\lambda = \frac{1}{2\pi i} \, d\sigma \wedge d\tau. \tag{250}$$

Canonical quantization means replacing the coordinates $\tau$ and $\sigma$ with the operators $\hat{\tau}$ and $\hat{\sigma}$, respectively, where $\hat{\tau}$ acts as multiplication by $\tau$ and $\hat{\sigma}$ acts as the differential operator $\frac{\epsilon}{2\pi i} \partial_\tau$. These replacements turn the spectral curve into the differential operator

$$d_\epsilon = \frac{\epsilon}{2\pi i} \partial_\tau - e^{2\pi i \tau}. \tag{251}$$

This differential operator plays a very special role: it annihilates the vortex partition function:

$$\left(\frac{\epsilon}{2\pi i} \partial_\tau - e^{2\pi i \tau}\right) \mathcal{Z}^{\text{vortex}}(\mathbf{z}, \epsilon) = 0. \tag{252}$$

### 4.4.3 Example: Higgs branch partition function for the $\mathbb{P}^{N-1}$-model

The derivation of the vortex partition function for the $\mathbb{P}^{N-1}$-model is more involved. Instead of following the original line of thought [55, 59], we will summarize the first-principle topological localisation computation as presented in [60].

That is, after lifting the 2d GLSM to a 4d $\mathcal{N} = 1$ theory in the background $M_4$, we can formulate it as a **cohomological field theory** (CohTFT) with respect to the scalar supercharge $\mathcal{Q} = Q_A^\epsilon$ [33]. Its partition function can thus be written in the form

$$\mathcal{Z} = \int \mathcal{DF} \, e^{-\mathcal{QV}+\mathcal{I}}, \tag{253}$$

where $\mathcal{F}$ represents all fields in the theory and where

$$\mathcal{Q}^2 \mathcal{V} = \mathcal{Q}I = 0. \tag{254}$$

Using standard CohTFT arguments we may replace $\mathcal{V}$ in the Lagrangian by $t\mathcal{V}$, for a new parameter $t$, and take the limit $t \to \infty$. The path integral then localises to the field configurations such that $\mathcal{QV} = 0$. In our case,

$$(\mathcal{QV})_{\text{bos}} = \int d^4 x \left[ \frac{1}{2e^2} \left\{ \left| 2iF_{z\bar{z}} + e^2 \left( |\phi_j|^2 - r \right) \right|^2 + \left| F_{\xi\bar{\xi}} \right|^2 + 8 \left| F_{\xi z} \right|^2 + 8 \left| F_{\xi\bar{z}} \right|^2 \right\} \right.$$
$$\left. + 4 \left| \mathcal{D}_{\bar{z}} \phi_j \right|^2 + 2 \left| \mathcal{D}_\xi \phi_j \right|^2 + 2 \left| \mathcal{D}_\xi \bar{\phi}_j \right|^2 \right], \tag{255}$$

where the $\xi$-index refers to the Killing vector field $\partial_\xi = \partial_w + i\epsilon \left( z\partial_z - \bar{z}\partial_{\bar{z}} \right)$ of the 4d geometry. Setting this to zero yields the BPS conditions

$$\mathcal{D}_{\bar{z}} \phi_j = 0, \quad 2iF_{z\bar{z}} + e^2 \left( |\phi_j|^2 - r \right) = 0,$$
$$F_{\xi\bar{\xi}} = F_{\xi z} = F_{\xi\bar{z}} = \mathcal{D}_\xi \phi_j = \mathcal{D}_\xi \bar{\phi}_j = 0. \tag{256}$$

We see that the first two equations are precisely the vortex equations, whereas the equations on the second row fix $A_\xi$ to a constant. This shows that we are computing a representative of the vortex partition function.

The exact partition function $\mathcal{Z}$ of the CohTFT can then simply be computed by a first-order saddle point analysis [61]. It is of the form

$$\mathcal{Z} = \sum_{\text{saddles}} \left[ e^{\mathcal{I}} \frac{\det \Delta_F}{\det \Delta_B} \right]_{\text{saddle}}, \tag{257}$$

where the saddles are the solutions to the BPS equations (256), and the determinants $\Delta_B$ and $\Delta_F$ can defined by expanding $\mathcal{QV}$ around each saddle in terms of the bosonic fluctuations $\Phi$ and the fermionic fluctuations $\Psi$, as in

$$\mathcal{QV} = \int d^4 x \left( \Phi^\dagger \Delta_B \Phi + \Psi^\dagger \Delta_F \Psi \right) + \text{higher order terms}, \tag{258}$$

and where we have used that

$$\int \mathcal{D}\Phi \mathcal{D}\Psi \, e^{-\int d^4x \left(\Phi^\dagger \Delta_B \Phi + \Psi^\dagger \Delta_F \Psi\right)} = \frac{\det \Delta_F}{\det \Delta_B}. \tag{259}$$

The saddles are labelled by a choice of vacua $\alpha$ and vortex number $\mathfrak{m}$. The bosonic fields are parametrised as

$$A_\xi = -m_\alpha - \mathfrak{m}\epsilon, \quad A_{\bar{z}} = -\frac{i}{2}\partial_{\bar{z}}\omega, \quad \phi_\beta = \begin{cases} \sqrt{r}e^{-\frac{1}{2}\omega}z^{\mathfrak{m}} & \text{for } \beta = \alpha, \\ 0 & \text{for } \beta \neq \alpha, \end{cases} \tag{260}$$

where $\omega$ is the so-called profile function solving the equation

$$\partial_z \partial_{\bar{z}}\omega = \frac{e^2 r}{2}\left(1 - z^{\mathfrak{m}}\bar{z}^{\mathfrak{m}}e^{-\omega}\right), \quad \lim_{|z|\to\infty} \omega = 2\mathfrak{m}\log\left(|z|\right). \tag{261}$$

The fermionic fields can simply be set to zero. This implies that each saddle contributes to the topological term $\mathcal{I}$ as

$$\mathcal{I}_{\alpha,\mathfrak{m}} = 2\pi i \left(\frac{\widetilde{m}_\alpha}{\epsilon} + \mathfrak{m}\right)\tau_0, \tag{262}$$

where the $0$-subscript in $\tau$ refers to its UV value.

The bosonic and fermionic fluctuations are instead parametrised by

$$\Psi = \begin{pmatrix} \bar{\lambda}_0 \\ \bar{\lambda}_{\bar{z}} \\ \psi_0 \\ \psi_{\bar{z}} \end{pmatrix}, \quad \Phi = \begin{pmatrix} -\delta A_\xi \\ \delta A_{\bar{z}} \\ \frac{-i}{\sqrt{2}}\delta\phi \\ 0 \end{pmatrix}, \tag{263}$$

whereas the bosonic and fermionic determinants are given by

$$\Delta_F = -i\nabla, \quad \Delta_B = \nabla\left(\nabla - \mathcal{D}_\xi - \mathcal{D}_{\bar{\xi}}\right), \tag{264}$$

with

$$\nabla = \begin{pmatrix} \mathcal{D}_\xi & \partial_z & \frac{\bar{\phi}}{\sqrt{2}} & 0 \\ -\partial_{\bar{z}} & \mathcal{D}_{\bar{\xi}} & 0 & -\frac{\bar{\phi}}{\sqrt{2}} \\ \frac{\phi}{\sqrt{2}} & 0 & \mathcal{D}_{\bar{\xi}} & \mathcal{D}_z \\ 0 & -\frac{\bar{\phi}}{\sqrt{2}} & -\mathcal{D}_{\bar{z}} & \mathcal{D}_\xi \end{pmatrix}. \tag{265}$$

An analysis of the eigenmodes of $\Delta_B, \Delta_F$ shows that only eigenmodes $\Phi$ satisfying the linearised vortex equations and their singular superpartners $\Psi$ contribute. For the $(\alpha, \mathfrak{m})$-saddle this yields

$$\left[\frac{\det \Delta_F}{\det \Delta_B}\right]_{\alpha,\mathfrak{m}} = \frac{(-1)^{\mathfrak{m}}}{\mathfrak{m}!}\left(\frac{\Lambda_0}{\epsilon}\right)^{N\left(\frac{\widetilde{m}_\alpha}{\epsilon} + \mathfrak{m}\right)} \prod_{\beta \neq \alpha} \sqrt{\frac{\Lambda_0}{2\pi\epsilon}}\,\Gamma\left(\frac{\widetilde{m}_\beta - \widetilde{m}_\alpha - \mathfrak{m}\epsilon}{\epsilon}\right), \tag{266}$$

after taking the zero-volume limit for the torus $T_w^2$ and regularising.

If we fix the vacuum $\alpha$ and sum over all corresponding saddles with vortex number $\mathfrak{m}$, we obtain the partition function

$$\mathcal{Z}_\alpha(\mathbf{z}, \epsilon) = e^{\frac{2\pi i \widetilde{m}_\alpha \tau}{\epsilon}} \sum_{\mathfrak{m}=0}^\infty \frac{(-1)^\mathfrak{m}}{\mathfrak{m}!} \left( \frac{\epsilon^N \mathbf{z}}{\mu^N} \right)^\mathfrak{m} \prod_{\beta \neq \alpha} \Gamma \left( \frac{\widetilde{m}_\beta - \widetilde{m}_\alpha - \mathfrak{m}\epsilon}{\epsilon} \right) \tag{267}$$

in the vacuum $a$, where

$$2\pi i \tau = 2\pi i \tau_0 + \log \frac{\Lambda_0^N}{\mu^N} \tag{268}$$

is the renormalized parameter at energy scale $\mu$.

There are two important alternative reformulations of the expression (267). The second reformation is related to mirror symmetry and will be discussed in §4.4.4. In the first reformulation we write $\mathcal{Z}_\alpha$ in the form

$$\mathcal{Z}_\alpha(\mathbf{z}, \epsilon) = \mathcal{Z}_\alpha^{\text{pert}}(\tau, \epsilon) \, \mathcal{Z}_\alpha^{\text{vortex}}(\mathbf{z}, \epsilon, \mu). \tag{269}$$

The perturbative contribution (in $\tau$) to the partition function is given by

$$\mathcal{Z}_\alpha^{\text{pert}}(\tau, \epsilon) = \exp \left( \frac{2\pi i \widetilde{m}_\alpha \tau}{\epsilon} \right) \prod_{\beta \neq \alpha} \Gamma \left( \frac{\widetilde{m}_\beta - \widetilde{m}_\alpha}{\epsilon} \right), \tag{270}$$

and contains the classical and one-loop contributions to the partition function. The remaining part can be expressed in terms of the generalised hypergeometric function $_0F_{N-1}$ as

$$\mathcal{Z}_\alpha^{\text{vortex}}(\mathbf{z}, \epsilon) = \sum_{\mathfrak{m}=0}^\infty \frac{(-1)^\mathfrak{m}}{\mathfrak{m}!} \left( \frac{\epsilon^N \mathbf{z}}{\mu^N} \right)^\mathfrak{m} \prod_{\beta \neq \alpha} \prod_{k=1}^\mathfrak{m} \frac{\epsilon}{\widetilde{m}_\beta - \widetilde{m}_\alpha - k\epsilon}$$
$$= \ _0F_{N-1} \left( \left\{ 1 - \frac{\widetilde{m}_\beta - \widetilde{m}_\alpha}{\epsilon} \right\}_{\beta \neq \alpha}, \left( -\frac{\mu}{\epsilon} \right)^N \mathbf{z} \right). \tag{271}$$

It may be identified with the vortex partition function (233) in the vacuum $\alpha$ and encodes all non-perturbative corrections (in $\tau$) due to vortices.[33]

We will in the following refer to the total partition function $\mathcal{Z}_\alpha$ as the **Higgs branch partition function** in the vacuum $\alpha$. Note that the decomposition (267) of $\mathcal{Z}_\alpha$ is analogous to the decomposition of the 4d $\mathcal{N} = 2$ partition function into a perturbative part, containing the classical and 1-loop contributions, and a non-perturbative part encoding the instanton corrections [62].

The differential operator $d_\epsilon$ annihilating the partition function $\mathcal{Z}_\alpha$ is given by

$$d_\epsilon = \mu^N e^{2\pi i \tau} - \prod_{j=1}^N (\hat{\sigma} - \widetilde{m}_j) \tag{272}$$

---

[33]We emphasize that these contributions are non-perturbative in the effective coupling $\tau$, as opposed to the non-perturbative corrections in $\epsilon$ that we will discuss in §5.

with

$$\hat{\sigma} = -\frac{\epsilon}{2\pi i}\,\partial_\tau. \tag{273}$$

Note that the associated differential equation indeed reduces to the spectral curve (181) in the semi-classical limit. The partition functions $\mathscr{Z}_\alpha$ in the vacua $\alpha$ are the linearly independent solutions of this quantum curve.

For $N = 2$ and $\widetilde{m}_1 = \widetilde{m} = -\widetilde{m}_2$, and with the change of variables

$$x = \frac{2i\,\mu\,\sqrt{\mathbf{z}}}{\epsilon}, \tag{274}$$

the differential operator (272) defines the Bessel equation

$$\left[x^2\partial_x^2 + x\partial_x + x^2 - \left(\frac{2\widetilde{m}}{\epsilon}\right)^2\right]\psi(x) = 0 \tag{275}$$

Hence, the Higgs branch partition functions $\mathscr{Z}_1$ and $\mathscr{Z}_2$ for the $\mathbb{P}^1$-model should be linearly independent solutions of the Bessel equation. By direct comparison of the series form for the partition functions, we find that

$$\begin{aligned} \mathscr{Z}_1 &= -\frac{\pi e^{-i\pi\nu/2}}{\sin(\pi\nu)}\,J_\nu(x), \\ \mathscr{Z}_2 &= +\frac{\pi e^{i\pi\nu/2}}{\sin(\pi\nu)}\,J_{-\nu}(x), \end{aligned} \tag{276}$$

where $J_\nu(x)$ is the Bessel function of the first kind and

$$\nu = \frac{2\widetilde{m}}{\epsilon}. \tag{277}$$

Before moving on, we note that in terms of $\mathbf{z} = e^{2\pi i\tau}$ the differential operator $d_\epsilon$ is given by

$$d_\epsilon = \epsilon^2\left(\mathbf{z}^2\partial_{\mathbf{z}}^2 - \mathbf{z}\partial_{\mathbf{z}}\right) - \widetilde{m}^2 - \mu^2\mathbf{z}. \tag{278}$$

By a small modification, corresponding to considering solutions of the form $\psi(\mathbf{z}) = \mathbf{z}^{\frac{1}{2}}\,\widetilde{\psi}(\mathbf{z})$, this Schrödinger equation is brought into the more familiar form

$$d_\epsilon = \epsilon^2\partial_{\mathbf{z}}^2 - \frac{\left(\widetilde{m}^2 - \frac{\epsilon^2}{4}\right)}{\mathbf{z}^2} - \frac{\mu^2}{\mathbf{z}}, \tag{279}$$

which you may recognize as "half" of the $SL(2)$-oper connection corresponding to the four-dimensional $\mathcal{N} = 2$ pure $SU(2)$ theory (see for instance §4 of [9] or §5.4 of [6]).

### 4.4.4 Remark: difference equations on the Coulomb branch

We have just learned that the Higgs branch partition functions $(\mathcal{Z}_\alpha)_\alpha$ are linearly independent solutions of the differential equation

$$d_\epsilon \mathcal{Z}_\alpha(\mathbf{z}, \epsilon) = 0, \tag{280}$$

which reduces in the semi-classical limit $\epsilon \to 0$ to the twisted chiral ring equation. In §5 we will find out that the differential operator $d_\epsilon$ transforms as a so-called **oper connection** on $C_\mathbf{z}$.

Here we instead want to follow up on remark §4.3.3 on exponential networks, or rather, on the different descriptions of the GLSM on the $\mathbf{z}$-plane and the $\sigma$-plane. Indeed, you may wonder why vortex counting corresponds to the canonical quantization of the spectral curve in which

$$\hat{\sigma} = \frac{\epsilon}{2\pi i} \, \partial_\tau \quad \text{and} \quad \hat{\tau} = \tau, \tag{281}$$

rather than the opposite choice

$$\hat{\tau} = \frac{\epsilon}{2\pi i} \, \partial_\sigma \quad \text{and} \quad \hat{\sigma} = \sigma. \tag{282}$$

The reason for this is that the vortex partition function is naturally an object on the Higgs branch of the theory, which is parametrized by $\mathbf{z} = e^{2\pi i \tau}$. Indeed, since it is defined as a power series in $\mathbf{z}$, see equation (233), it naturally has good asymptotics when $\mathbf{z} \to 0$.

Nonetheless, the alternative choice of quantization is interesting as well. This transforms the twisted chiral ring into a difference operator $D_\epsilon$ (or better: a difference oper connection). For instance, in the abelian Higgs model we obtain the difference equation

$$D_\epsilon \widetilde{\mathcal{Z}} = \left(e^{\epsilon \partial_\sigma} - \sigma\right) \widetilde{\mathcal{Z}} = 0. \tag{283}$$

Its solution

$$\widetilde{\mathcal{Z}}(\sigma, \epsilon) = \epsilon^{\frac{\sigma}{\epsilon}} \, \Gamma\left(\frac{\sigma}{\epsilon}\right) \tag{284}$$

should be interpreted as the GLSM partition function on the $\sigma$-plane. From now on, we refer to this partition function as the Coulomb branch partition function and denote it by $\widetilde{\mathcal{Z}}^{\text{Coulomb}}(\sigma, \epsilon)$.

The Higgs branch partition function for the abelian Higgs model is simply equal to its vortex partition function (248). Hence, the Higgs and Coulomb branch partition functions are related by a Fourier transform

$$\widetilde{\mathcal{Z}}^{\text{Coulomb}}(\sigma, \epsilon) = \epsilon^{\frac{\sigma}{\epsilon}} \, \Gamma\left(\frac{\sigma}{\epsilon}\right) = \int_0^\infty d\mathbf{z} \, \mathbf{z}^{\frac{\sigma}{\epsilon} - 1} \, e^{\frac{\mathbf{z}}{\epsilon}} = 2\pi \int_{-\infty}^\infty d\widetilde{\tau} \, e^{\frac{2\pi \widetilde{\tau} \sigma}{\epsilon}} \, e^{\frac{\mathbf{z}}{\epsilon}} =$$
$$= 2\pi \int_{-\infty}^\infty d\widetilde{\tau} \, e^{\frac{2\pi \widetilde{\tau} \sigma}{\epsilon}} \, \mathcal{Z}^{\text{vortex}}(\mathbf{z}, \epsilon), \tag{285}$$

where we defined $\tilde{\tau} = i\tau$.

A similar remark holds for the Higgs branch partition function of any GLSM. For instance, the Higgs branch partition function (267) for the $\mathbb{P}^{N-1}$-model can be re-expressed as a contour integral through the identity

$$\sum_{\mathfrak{m}=0}^{\infty} \frac{(-1)^{\mathfrak{m}}}{\mathfrak{m}!} f(-\mathfrak{m}) = \int_C \frac{d\sigma}{2\pi i} \, \Gamma(\sigma) \, f(\sigma), \tag{286}$$

where $C$ is a contour encircling all poles of the gamma function $\Gamma(\sigma)$ on the negative real axis. We then find that

$$\mathcal{Z}_\alpha(\mathbf{z}, \epsilon) = e^{\frac{2\pi i \widetilde{m}_\alpha \tau}{\epsilon}} \int_C \frac{d\sigma}{2\pi i \epsilon} \, e^{\frac{2\pi i \sigma \tau}{\epsilon}} \left( \frac{\epsilon}{\mu} \right)^{\frac{N\sigma}{\epsilon}} \prod_{\beta=1}^{N} \Gamma\left( \frac{\widetilde{m}_\alpha - \widetilde{m}_\beta + \sigma}{\epsilon} \right) =$$
$$= \int_{C_\alpha} \frac{d\sigma}{2\pi i \epsilon} \, e^{\frac{2\pi i \sigma \tau}{\epsilon}} \left( \frac{\epsilon}{\mu} \right)^{\frac{N}{\epsilon}(\sigma - \widetilde{m}_\alpha)} \prod_{j=1}^{N} \Gamma\left( \frac{\sigma - \widetilde{m}_j}{\epsilon} \right). \tag{287}$$

Indeed, the Coulomb branch partition function

$$\widetilde{\mathcal{Z}}^{\text{Coulomb}}(\sigma, \epsilon) = \left( \frac{\epsilon}{\mu} \right)^{\frac{N\sigma}{\epsilon}} \prod_{j=1}^{N} \Gamma\left( \frac{\sigma - \widetilde{m}_j}{\epsilon} \right) \tag{288}$$

is annihilated by the difference operator

$$D_\epsilon = \mu^N e^{\epsilon \partial_\sigma} - \prod_{j=1}^{N} (\sigma - \widetilde{m}_j). \tag{289}$$

You may recognise the partition function of $N$ free chiral fields in (288), with twisted masses $\sigma - \widetilde{m}_j$. Indeed, each such chiral field is known to have a partition function proportional to the gamma function[34]

$$\Gamma\left( \frac{\sigma - \widetilde{m}_j}{\epsilon} \right) = \exp\left( \frac{1}{\epsilon} \left( \widetilde{W}_{\text{eff}}^{\text{chiral}}(\sigma - \widetilde{m}_j) + \mathcal{O}(\epsilon) \right) \right) \tag{290}$$

where

$$\widetilde{W}_{\text{eff}}^{\text{chiral}} = (\sigma - \widetilde{m}_j) \left( \log\left( \frac{\sigma - \widetilde{m}_j}{\epsilon} \right) - 1 \right) \tag{291}$$

is the effective twisted superpotential for a free chiral field with twisted mass $\sigma - \widetilde{m}_j$ at energy scale $\mu = \epsilon$. In particular, note that the leading contribution in $\epsilon$ of the logarithm of the integrand in (287) computes the effective twisted superpotential (179) of the full GLSM on the Coulomb branch, at energy scale $\mu = \epsilon$.

---

[34]This partition function can be computed as a 1-loop determinant in the 2d $\Omega$-background, see for instance [55] or [56].

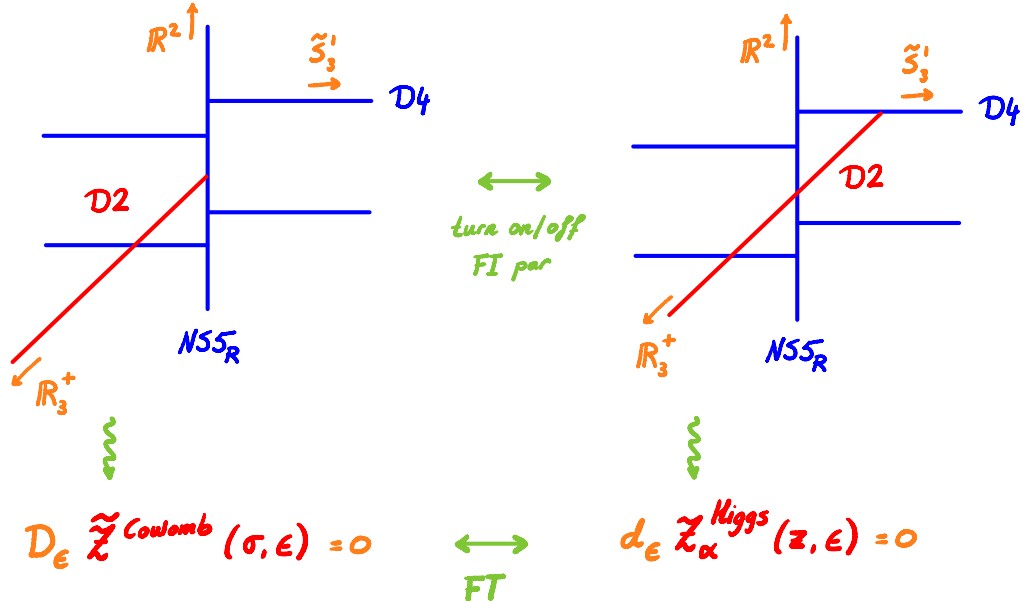

Figure 27: Brane setup that computes the Coulomb versus the Higgs branch partition functions. Note that the left and right picture differ in the fact that the D2-branes have moved off the NS5-brane onto a D4-brane. The directions (in orange) are part of the gauge origami notation (299) used in §4.4.5.

More precisely, we can rephrase the expression (287) as the partition function of the UV mirror to the GLSM (as introduced in §4.2.5). Indeed, if we substitute the definition

$$\Gamma(\sigma) = \int_{-\infty}^{\infty} dY \, e^{\sigma Y - e^Y}, \quad \text{Re}(\sigma) > 0, \tag{292}$$

into equation (287), we find (after some manipulations) that

$$\mathcal{Z}_\alpha = \int_{C_\alpha} \frac{d\sigma}{2\pi i \epsilon} \int_{\mathbb{R}^N} dY_1 \cdots dY_N \, \exp\left(-\frac{1}{\epsilon} \widetilde{W}_{\text{exact}}(\sigma, Y_j)\right), \tag{293}$$

in terms of *exact* twisted superpotential (see also (192))

$$\widetilde{W}_{\text{exact}}(\sigma, Y_j) = -2\pi i \sigma \tau + \sum_{j=1}^{N} \left((\sigma - \widetilde{m}_j) Y_j + \mu e^{-Y_j}\right) \tag{294}$$

of the UV Landau-Ginzburg mirror, at the energy scale $\mu = \epsilon$ [60]. This suggests that the Higgs branch partition function $\mathcal{Z}_\alpha$ for the $\mathbb{P}^{N-1}$-model can be obtained through a localization computation in the $\Omega$-deformed mirror theory.[35]

---

[35]We discuss partition functions for LG models in the 2d $\Omega$-background later in §5.6.

### 4.4.5 Remark: partition functions from gauge origami

Recall from §4.3.4 that GLSM's can be embedded in string theory using a system of D2-branes ending on a combination of D4 and NS5-branes. Also, remember that the movement of the D2's along the D4-branes parameterizes the Higgs branch of the GLSM, whereas the movement of the D2's along the NS5$_R$-brane parametrizes its Coulomb branch. We may thus anticipate that the Higgs and Coulomb branch partition functions can be computed as D2-brane partition functions relative to the above D-brane setups, as illustrated in Figure 27.

To make this precise, we need to turn on the $\Omega$-background in the D-brane system. This can be accomplished through the gauge origami, introduced by Nikita in [63] (as well as other papers in the same series). That is, we start in Type IIB with the ten-dimensional background

$$\text{IIB}: \quad X \times T^2, \tag{295}$$

where $X$ is a Calabi-Yau 4-fold. In fact, think of this Calabi-Yau simply as the product $X \sim \mathbb{C}_1 \times \mathbb{C}_2 \times \mathbb{C}_3 \times \mathbb{C}_4$.

Now T-dualize along both cycles of $T^2$, and deform the resulting geometry

$$\text{IIA}: \quad X \times \widetilde{T}^2 \tag{296}$$

into a $X$-fibration over $\widetilde{T}^2$, such that $X$ is rotated by the $U(1)^3 \subset SU(4)$ isometry along the cycles of $\widetilde{T}^2$ with parameters $\epsilon_{1,2,3,4}$ for which $\sum_i \epsilon_i = 0$. In the limit $\widetilde{T}^2 \to 0$ this yields an 8-dimensional generalization of the $\Omega$-background introduced in §4.4.1. The background

$$\text{IIB}: \quad X_{\epsilon_1,\epsilon_2,\epsilon_3,\epsilon_4} \times \mathbb{R}^2 \tag{297}$$

that we obtain after T-dualizing back, where we may think of the $\Omega$-deformed Calabi-Yau as the product $X_{\epsilon_1,\epsilon_2,\epsilon_3,\epsilon_4} \sim \mathbb{C}_{\epsilon_1} \times \mathbb{C}_{\epsilon_2} \times \mathbb{C}_{\epsilon_3} \times \mathbb{C}_{\epsilon_4}$, is called the gauge origami [63]. We can add to this background various kinds of D-branes that preserve the $U(1)^3$ isometry.

The Type IIB gauge origami background can be related to the Type IIA Hanany-Hori setup through a few additional string dualities. To obtain the brane configuration from §4.3.4 we start with the origami background

$$\text{IIB}: \quad X \times \mathbb{R}^2 = \mathbb{C}_1 \times \mathbb{C}_2 \times (\mathbb{C}_3 \times \mathbb{C}_4)/\mathbb{Z}_2 \times \mathbb{R}^2, \tag{298}$$

and introduce $N$ additional D3-branes inserted at the origin of $(\mathbb{C}_3 \times \mathbb{C}_4)/\mathbb{Z}_2$ while wrapping $\mathbb{C}_1 \times \mathbb{C}_2$. After a T-duality along the (degenerate) $S_3^1 \subset \mathbb{C}_3$, we end up with the type IIA background

$$\text{IIA}: \quad \mathbb{C}_1 \times \mathbb{C}_2 \times (\mathbb{R}_3^+ \times \widetilde{S}_3^1) \times \mathbb{C}_4 \times \mathbb{R}^2 \tag{299}$$

in which the D3-branes have turned into D4-branes wrapped along the dual $\widetilde{S}_3^1$. Moreover, because the original $S_3^1$ degenerates at the origin of $(\mathbb{C}_3 \times \mathbb{C}_4)/\mathbb{Z}_2$, two

additional NS5-branes emerge at antipodal points along the dual $\widetilde{S}^1_3$. These NS5-branes wrap $\mathbb{C}_1 \times \mathbb{C}_2 \times \mathbb{R}^2$.

Decompactifying the $\widetilde{S}^1_3$ means that we loose one of the NS5-branes and end up with a Hanany-Hori setup similar to the one discussed in §4.3.4. That is, if we identify $\mathbb{C}_1 \times \mathbb{C}_2$ with the 0189-directions in §4.3.4, and $\widetilde{S}^1_3$ with the 7-direction, we see that we have obtained a Type IIA configuration with the D4-branes and the $\mathrm{NS5_R}$ brane from §4.3.4. To be precise, we have ended up with a Hanany-Hori configuration with the same number of upper and lower D4-branes, but this can be modified.

The only ingredients that are still missing are the $\mathrm{NS5_L}$-brane and the D2-branes! In fact, we will ignore the $\mathrm{NS5_L}$-brane as it is not relevant for the following discussion. The D2-branes can be introduced by inserting D3-branes in the gauge origami at the origin of $\mathbb{C}_2 \times \mathbb{C}_4$ while wrapping $\mathbb{C}_1 \times \mathbb{C}_3$. The resulting D2-branes end on the NS5-brane, and we thus expect their partition function to compute the Coulomb branch partition function. The final setup is illustrated in Figure 27.

Indeed, the brane partition function of the resulting setup can be computed using the same equivariant localization techniques that we introduced in §4.4.1. More precisely, the partition function will be an equivariant integral of an equivariant Euler class of a certain vector bundle over the moduli space of D(-1)-instantons that are dissolved in the brane background. These D(-1)-instantons are called spiked instantons by Nikita, and their moduli space has an ADHM-like quiver description.

In fact, this gauge origami partition function contains some more information than we are interested in. In particular, we want to set $\epsilon_1 = \epsilon$ and set $\epsilon_2 = \epsilon_3 = 0$ to relate it to the Coulomb branch partition function. The resulting partition function is called the $Q$-**observable** by Nikita (because they have the same form as the $Q$-operators in the Baxter $TQ$-relation) and it indeed reproduces the Coulomb partition functions such as the Coulomb partition function (288) for the $\mathbb{P}^{N-1}$-model.

By turning on the complexified FI parameter in the underlying GLSM, the gauge theory is brought in the NLSM phase, and the D2-branes move onto one of the $N$ D4-branes. The resulting Higgs branch partition function can be written as an integral over the $Q$-observable, and is sometimes called the **canonical surface defect** partition function (following [39]). The Fourier transform (285) in this setting is studied in detail, for instance, in reference [64].

### 4.4.6   Remark: supersymmetric localization

So far, we have described the computation of vortex partition function in the two-dimensional $\Omega$-background using topological localization techniques. More recent is the computation of the exact partition functions for 2d $\mathcal{N} = (2, 2)$ theory on certain symmetric spaces using **supersymmetric localization** techniques. An example of such space is the squashed two-sphere, which is embedded in $\mathbb{R}^3$ by the equation

$$b^2(x_1^2 + x_2^2) + x_3^2 = R^2 \tag{300}$$

with squashing parameter $b$ and radius $R$, and similarly the squashed hemisphere. See, for instance, the review [65] for an introduction into supersymmetric localiza-

tion, the papers [66, 56] for more details on the supersymmetric two-sphere partition function, and the papers [67, 68, 69] for more details on the supersymmetric hemisphere partition function.

Supersymmetric localization relies on the existence of a Killing spinor in the curved background. Such a Killing spinor parametrizes the preserved supersymmetry in the curved background. The idea of supersymmetric localization is to employ the corresponding supercharge to simplify the path integral to a finite-dimensional integral over saddle points, just as in the topological localization method.

It turns out that the supersymmetric partition function on the squashed two-sphere (and similar for the hemi-sphere) is independent of the squashing parameter $b$ and localizes to an integral over vortex configurations on the north pole and anti-vortex configurations on the south pole. This implies that the two-sphere partition function can be factorized into a product of the vortex partition function on one hemisphere and the anti-vortex partition function on the other, with the deformation parameter $\epsilon$ equal to the inverse radius $\frac{1}{R}$ [66, 56]. The precise relation between the hemi-sphere partition function and the vortex partition function is detailed in [67, 68, 69]. Essentially, the hemi-sphere partition function, with standard boundary conditions, is equal to the sum over all vacua $\alpha$ of the Higgs branch partition function $\mathcal{Z}_\alpha$, where $\epsilon = \frac{1}{R}$.

The relation between supersymmetric and topological localisation is explained in [59] as follows. To find Killing spinors on the two-sphere, it is crucial to turn on a background gauge field for the $U(1)$ R-symmetry. Now, this background field reduces to (minus) half the spin connection near the north (south) pole of the two-sphere, and, as explained in §4.4.1, is thus equivalent to considering the (anti) A-twist in this region.

# 5 Non-perturbative partition functions and exact WKB analysis

In this section, we construct a 2d $\mathcal{N} = (2, 2)$ partition function that is non-perturbative in the $\Omega$-background parameter $\epsilon$ and captures the full BPS soliton spectrum. We relate this partition function to the exact WKB analysis and compute it in various examples, including Landau-Ginzburg models and GLSM's.

## 5.1 Half-BPS boundary conditions

The partition function that we want to introduce in this section will be defined on a two-dimensional background with boundary. Let us therefore brush up what we know about supersymmetric boundary conditions in two dimensions.

First, we emphasize that it is impossible to preserve the full $\mathcal{N} = (2, 2)$ algebra at a spatial boundary because the spatial translation symmetry $P$ is broken. The next best thing we can hope to preserve is an $\mathcal{N} = 2$ subalgebra with only time

translations. Recall that we encountered two candidates in §2.2.1: the A-type and the B-type subalgebras, both indexed by a phase $\xi$.

Suppose that we are in the setting of Landau-Ginzburg models, with superpotential $W$ and target manifold $X$, on a two-dimensional space-time with boundary. Just as $\mathcal{N} = (2,2)$ supersymmetry forces the target $X$, which the worldsheet is mapped into, to be a Kähler manifold, the boundary condition $\Gamma \subset X$, which the worldsheet boundary is mapped into, must have special geometric properties to preserve half of the supersymmetry. By imposing standard Dirichet/Neumann boundary conditions on the bulk LG action, one finds that [38, 23]:

- The A-type $\mathcal{N} = 2$ subalgebra (with phase $\xi = -\zeta^{-1}$) is preserved when $\Gamma$ is a **Lagrangian** submanifold of $X$, and such that $W(\Gamma)$ is a straight line with angle $\pm\zeta$. The cycle $\Gamma$ is said to support an **A-brane**.

- The B-type $\mathcal{N} = 2$ subalgebra is preserved when $\Gamma$ is a **complex** submanifold of $X$, such that $W(\Gamma)$ is a constant. This cycle $\Gamma$ is said to support an **B-brane**.

It is furthermore possible to couple the worldsheet boundary to a gauge field living on the brane. Such a configuration preserves supersymmetry whenever the gauge field is **flat** on an A-brane, and whenever it defines a **holomorphic** line bundle on a B-brane. These deformations will not play an important role in these notes.

Even more generally, the A-type boundary conditions may be relaxed by adding suitable boundary interactions to the LG Lagrangian (see for instance §7.1 in [70] and §11.2 in [35]). One then finds that any Lagrangian submanifold $L \subset X$ can serve as the support of an A-brane. Moreover, it turns out that Hamiltonian deformations of the Lagrangian $L$ are induced by ($Q$-exact) boundary D-terms. If $X$ is non-compact, one needs to supplement the Lagrangian condition with a boundary condition at infinity of the Lagrangian submanifold. In the presence of the superpotential $W$ one may require that $\text{Im}(\zeta^{-1}W) \to \infty$ at infinity of $L$.

Remember that we defined the left/right Lefschetz thimbles $J^{\zeta}_{\alpha,L} = J^{-\zeta}_{\alpha,R} \subset X$ as the union of all solutions to the $\zeta$-soliton equation (103) with left/right boundary condition given by the vacuum $\alpha$, and that they may be obtained through the upward Morse flow generated by the Morse function

$$\mathbf{h} = \text{Im}(\zeta^{-1}W). \tag{301}$$

These thimbles therefore clearly have the right properties to support A-branes.[36] The D-brane charge of such an A-brane is given by its (middle-dimensional) homology

---

[36]Considering A-type boundary conditions in an LG mdoel may seem counter-intuitive to someone who is familiar with the LG model in the B-twist. Indeed, the B-twist is the most common twist to study LG models, because it picks out chiral operators. Also, in this twist the natural boundary conditions are given by B-branes, for instance, studied in [71]. Yet, at the moment we do not want to restrict ourselves to a twisted sector. We are merely analysing boundary conditions that preserve half of the supersymmetry in the **physical** $\mathcal{N} = (2,2)$ theory. In this context, it is actually more natural to consider A-type boundary conditions, because these boundary conditions preserve the axial R-symmetry that is compatible with $Z \neq 0$.

class

$$[J_\alpha^\zeta] \in H_n(X, B), \tag{302}$$

where the subspace $B$ was defined below equation (124). In fact, since the collection of Lefschetz cycles $(J_\alpha^\zeta)_\alpha$ span the homology group $H_n(X, B)$, for any fixed choice of $\zeta$ and under certain genericity conditions, we may view $H_n(X, B)$ as the charge lattice of these A-branes.

As an example, suppose that we consider the LG model on the strip [38]

$$I_\sigma \times \mathbb{R}_\tau. \tag{303}$$

with boundary condition at the left and right end of the interval $I_\sigma$ implemented by an A-brane wrapping the Lefschetz cycles $J_{\alpha,L}^\zeta$ and $J_{\beta,R}^\zeta$, respectively. This setup describes the propagation of an open string stretched between the two A-branes, and preserves the A-type subalgebra with phase $\xi = -\zeta^{-1}$.

**Supersymmetric ground states** of the LG model on the strip are in 1-1 correspondence with the $\zeta$-solitons with central charge $Z_{\alpha\beta}$. In the $W$-plane they project to trajectories that are orthogonal to the Lefschetz thimbles. Indeed, whereas the energy in a system without boundary conditions is minimized by a straight trajectory in the $W$-plane connecting the critical points $W(\alpha)$ and $W(\beta)$, this is not the case any longer when we introduce boundary conditions, in which case the trajectory can start at any point of the image of the Lefschetz thimble. The **index** of the supersymmetric ground states at phase $\zeta$ is then equal to the oriented intersection number

$$J_\alpha^{\zeta e^{i\epsilon}} \circ J_\beta^{\zeta e^{-i\epsilon}}, \tag{304}$$

where we have slightly rotated the phase of the Lefschetz thimbles to make the thimbles intersect transversally.

Mathematically, we find that the boundary conditions for an LG model form a category, which is known as the **Fukaya-Seidel category**. Given some physicality conditions on $W$ (mathematically we require that it defines a Lefschetz fibration), this category is generated by the Lefschetz cycles $J_\alpha^\zeta$ indexed by the vacua $\alpha$, whereas the morphisms are given by the Floer homology groups[37]

$$\mathrm{HF}_W^*(J_{\alpha,L}^\zeta, J_{\beta,R}^\zeta) \tag{305}$$

that "count" the intersection points of the left and right Lefschetz cycles $J_{\alpha,L}^\zeta$ and $J_{\beta,L}^\zeta$, when they are slightly rotated away from the critical soliton phase.

Remember that the Lefschetz cycles jump as

$$[J_{\alpha,L}^{\zeta'}] = [J_{\alpha,L}^\zeta] \pm \mu_{\alpha\beta} [J_{\beta,L}^\zeta]. \tag{306}$$

---

[37]To be precise, as we will expand on in §5.9, the morphisms of the Fukaya-Seidel category are given by local boundary operators. One requires the operator-state correspondence to relate these boundary operators to states on the interval.

across the critical phases $\zeta_{\alpha\beta} = \arg(Z_{\alpha\beta})$, as we saw before in equation (125). In the context of branes, these jumps may be interpreted as what is known as **brane creation** [48]. That is, an A-brane that wraps the cycle $J_{\alpha,L}^{\zeta}$ will only depend continuously on the phase $\zeta$ if a new brane $J_{\beta,L}^{\zeta}$ is created at the phase $\zeta_{\alpha\beta}$.

Finally, let us comment on half-BPS boundary conditions for more general $\mathcal{N} = (2,2)$ theories such as GLSM's. Recall from §4.2.2 that GLSM's have a *twisted chiral* LG model description on the Coulomb branch in terms of twisted chiral fields $\Sigma$ coupled by an effective twisted superpotential $\widetilde{W}_{\text{eff}}(\Sigma)$. We can thus simply adapt the previous discussion regarding *chiral* LG models to find half-BPS boundary conditions for GLSM's in their low-energy limit. Note that the *twisted chiral* LG model preserves the vector (instead of the axial) R-symmetry, which is compatible with $\tilde{Z} \neq 0$ (instead of $Z \neq 0$), and A-type and B-type boundary conditions are therefore reversed compared to boundary conditions for a *chiral* LG model.

We may also try to construct half-BPS boundary conditions through their microscopic description. Such UV boundary conditions must similarly preserve an $\mathcal{N} = 2$ subalgebra indexed by a phase $\xi = -\zeta^{-1}$. They can generically be defined as combinations of Neumann and Dirichlet boundary conditions on the GLSM multiplets, possibly combined with additional boundary interactions. The "basic boundary condition" from [67] imposes a Dirichlet boundary condition on $\text{Re}(\sigma)$ and the component $A_{\perp}$ of the gauge field perpendicular to the boundary, and a Neumann boundary condition on $\text{Im}(\sigma)$ and the component of the gauge field $A_{\parallel}$ parallel to the boundary. For the chiral fields there are two possibilities: one can either impose a Neumann or Dirichlet boundary condition on the complex scalar $\phi$.

The forth-coming analysis (see §5.8) suggests that the basic boundary conditions map in the IR to the collection of Lefschetz thimbles

$$(J_{\alpha,L}^{\pm\zeta_{\text{FN}}})_{\alpha}, \tag{307}$$

at a special phase $\zeta_{\text{FN}}$ corresponding to the presence of self-solitons. IR boundary conditions for any other value of $\zeta$ may be reconstructed in the UV description by coupling additional one-dimensional matter to the boundary.

## 5.2  Cigar partition function

We could try to define the $\mathcal{N} = (2,2)$ theory on any Riemann surface $\Sigma$ with boundary circles. In general, this can only be achieved by twisting. Yet, let us zoom in on the description of the $\mathcal{N} = (2,2)$ theory in the neighborhood of the boundary circles. If we choose a metric on $\Sigma$ that turns an open region near each boundary circle into a flat cylinder $I_{\sigma} \times S_{\tau}^{1}$, the original theory is well-defined in each such open region, and we might impose either the A-type or B-type boundary conditions at the boundary circles.

We focus again on the class of LG models, where we may consider A-branes supported on the Lefschetz thimbles $J_{\alpha}^{\zeta} \subset X$. If we make the same choice of phase $\zeta$ at each boundary circle, the total boundary condition preserves the $\mathcal{N} = 2$ subalgebra of type A with phase $\xi = -\zeta^{-1}$.

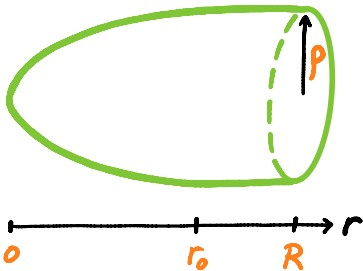

Figure 28: The cigar background $D_R$ with asymptotic radius $\rho$.

Let us consider in detail the special case where $\Sigma$ is topologically a disc. As a metric on $\Sigma$ we choose

$$ds^2 = dr^2 + f(r)\,d\tau^2, \tag{308}$$

where $f(r)$ can be any function on the interval $r \in [0, R]$, for given $R \gg 0$, such that

$$\begin{aligned} f(r) &\sim r^2 \quad \text{as } r \to 0, \\ f(r) &= \rho^2 \quad \text{when } r \in [r_0, R], \end{aligned} \tag{309}$$

where $0 < r_0 \ll R$ and the **asymptotic radius** $\rho \ll R$ are fixed constants. This metric is known as the cigar metric, and the resulting geometry as the **cigar background** $D_R$. The cigar geometry is illustrated in Figure 28.

Choose the Lefschetz thimble $J_\alpha^\zeta$ as a half-BPS boundary condition. If we were able to extend the LG model on the boundary cylinder to the entire cigar geometry, we would be able to compute a cigar partition function $\mathcal{Z}_\alpha^\zeta$. Another way to phrase this is to say that the LG model on $\Sigma$ would determine a state $|\mathcal{Z}\rangle$ in the Hilbert space of the LG model, and that the Lefschetz thimble would determine a boundary state $\langle \alpha; \zeta |$ such that

$$\mathcal{Z}_\alpha^\zeta = \langle \alpha; \zeta | \mathcal{Z} \rangle. \tag{310}$$

Such a **cigar partition function** was studied through the tt$^*$ geometry in [72] (see for instance [19, 20] for well-written reviews). In §5.4 we construct a related cigar partition function

$$\mathcal{Z}_\alpha^\zeta(\mathbf{z}, \epsilon) \tag{311}$$

by placing the LG model $T_{\mathbf{z}}$ in the $\Omega$-background with parameter $\epsilon$, following techniques developed in [62].[38]

The cigar partition function acquires the same transformation properties as the boundary condition $J_\alpha^\zeta$ when $\zeta$ crosses a critical phase. Indeed, if $\zeta$ crosses $\zeta_{\alpha\beta}$ then

$$\begin{aligned} \mathcal{Z}_\alpha^\zeta &\mapsto \mathcal{Z}_\alpha^\zeta \pm \mu_{\alpha\beta} \mathcal{Z}_\beta^\zeta, \\ \mathcal{Z}_\gamma^\zeta &\mapsto \mathcal{Z}_\gamma^\zeta, \end{aligned} \tag{312}$$

---

[38] Another method to define such a partition function is through supersymmetric localisation, as for instance studied in [67, 68, 69] (at least for specific phases $\zeta$).

for any $\gamma \neq \alpha$.

In this setup we may implement the morphisms of the Fukaya-Seidel category as half-BPS **domain walls** that capture the BPS solitons with central charge $Z_{\alpha\beta}$. Indeed, after inserting this domain wall on the disc with boundary condition labeled by $\alpha$, we obtain the disc with boundary condition labeled by $\beta$. This is illustrated in Figure 29.

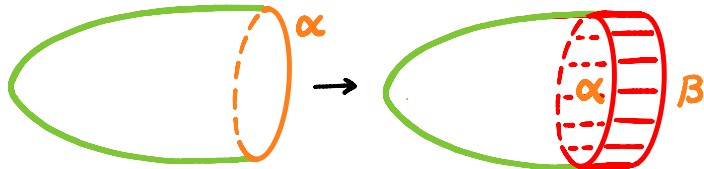

Figure 29: Inserting an $\alpha\beta$-domain wall in the cigar background $D_R$ changes the boundary condition from $\alpha$ to $\beta$.

Even though the discussion in this subsection has focused on LG models, its conclusions may be extended to any 2d $\mathcal{N} = (2,2)$ theory with a preserved $U(1)_R$-symmetry. That is, for any such theory we may consider a cigar partition function $\mathcal{Z}_\alpha^\zeta$ that jumps whenever $\zeta$ crosses the phase of a BPS soliton with central charge $Z_{\alpha\beta}$ (or $\widetilde{Z}_{\alpha\beta}$, depending on whether it is the axial or vector R-symmetry that is preserved).

## 5.3 $\Omega$-deformation of the cigar geometry

In this section we consider 2d GLSM's in the cigar geometry $D_R$, as defined around equation (308), while turning on the $\Omega$-**background** with parameter $\epsilon$.

Recall from §4.4.1 the description of the $\Omega$-deformation of the 2d space-time $\mathbb{C}_z$, which was defined in terms of a 4d twisted fibration of $\mathbb{C}_z$ over an auxiliary torus $T_w^2$, with 4d metric

$$ds^2 = |dz - iz(\epsilon dw + \bar{\epsilon} d\overline{w})|^2 + |dw|^2. \tag{313}$$

This description can be generalised to any 2d space-time that admits a Killing vector field $V^\mu \partial_\mu$. The corresponding 4d metric on the twisted fibration over the auxiliary torus $T_w^2$ reads

$$ds^2 = (dx^\mu - V^\mu(\epsilon dw + \bar{\epsilon} d\bar{w})))^2 + |dw|^2. \tag{314}$$

Specialising to the cigar background $D_R$, which admits the Killing vector field $\partial_\tau$, we recover the 4d metric

$$ds^2 = dr^2 + f(r)\left(d\tau - (\epsilon dw + \bar{\epsilon} d\bar{w})\right)^2 + |dw|^2. \tag{315}$$

The non-trivial curvature of the $\Omega$-background makes it possible to preserve at most two out of the four supercharges, which we have identified as **A-type supercharges** $Q_A^\xi$, for any $\xi$, in §4.4. However, unlike in §4.4, the cigar partition function also requires the data of supersymmetric boundary conditions. In this section we

find the appropriate boundary conditions by **undoing** the $\Omega$-deformation away from the tip of the cigar, similar to the discussion in §3.2 of [73].

Our argument makes use of the 4d origin of 2d GLSMs: The 2d $\mathcal{N} = (2,2)$ GLSM labeled $T_{\mathbf{z}}$ can be obtained via a torus reduction of a 4d $\mathcal{N} = 1$ theory $\overline{T}_{\mathbf{z}}$. Then, the 2d $\Omega$-deformed theory $T_{\mathbf{z}}^{\epsilon}$ is obtained by placing the 4d theory $\overline{T}_{\mathbf{z}}$ in the deformed background (315), while sending the volume of the auxiliary torus $T_w^2$ to zero (just as in §4.4).

In the lift to four dimensions, the real and imaginary components of the complex scalar field $\sigma$ get identified with the components of the 4d gauge field $A$ in the auxiliary torus directions. We would like to know the associated gauge covariant derivatives. Note that the translation generator of the deformed metric (314) is no longer $\partial_w$ but rather $\partial_w + \epsilon V^\mu \partial_\mu$, so the corresponding gauge covariant derivative $D_w = \partial_w + \sigma$ is also deformed to

$$D_w \mapsto D_w + \epsilon V^\mu D_\mu. \tag{316}$$

Since we will compactify the 4d theory on the auxiliary torus $T_w^2$, we can assume that all 2d fields are constant along the torus directions. This implies that the $\Omega-$deformation is effectively a replacement of the scalar $\sigma$ by $\sigma \mapsto \sigma + \epsilon V^\mu D_\mu$, which becomes

$$\sigma \mapsto \sigma + \epsilon D_\tau \tag{317}$$

in the deformed cigar background (315). Note that this replacement should be understood as a deformation of the action, and is meaningless otherwise!

For ease in notation, let us restrict to $r > r_0$ where $f(r) = \rho^2$ and consider the 2d GLSM Lagrangian on the original cigar $D_R$. The only terms relevant for this argument are

$$\mathcal{L}_{\text{bos}} = \frac{1}{e^2} \left( \frac{1}{2} F_{\mu\nu} F_{\mu\nu} + D_\mu \sigma D_\mu \sigma^\dagger \right). \tag{318}$$

Note that while $D_\tau \sigma D_\tau \sigma^\dagger$ is invariant under the $\Omega$-deformation, the Yang-Mills term is not. Yet, it turns out that the $\Omega$-deformation can be nullified in this region, up to $\mathcal{O}(\epsilon^2)$ terms, by the field redefinition

$$A_\tau \mapsto A_\tau - \frac{1}{2}(\bar{\epsilon}\sigma + \epsilon\sigma^\dagger). \tag{319}$$

This is perhaps clearest from the four-dimensional perspective, from which the $\Omega$-deformation (317) and the field redefinition (319) combine into the infinitesimal rotation

$$\begin{pmatrix} D_\tau \\ D_2 \\ D_3 \end{pmatrix} \mapsto \begin{pmatrix} 1 & -\mathrm{Re}(\epsilon) & -\mathrm{Im}(\epsilon) \\ \mathrm{Re}(\epsilon) & 1 & 0 \\ \mathrm{Im}(\epsilon) & 0 & 1 \end{pmatrix} \begin{pmatrix} D_\tau \\ D_2 \\ D_3 \end{pmatrix}, \tag{320}$$

of the $\tau 23-$subspace, where we have used the notation $w = w_2 + iw_3$.

Indeed, it is possible to extend the infinitesimal rotation (320) to a finite rotation that eliminates all orders of $\epsilon$ in the Lagrangian. Let us define

$$X = \rho^{-1} D_\tau \tag{321}$$

to be the covariant derivative of correct mass dimensions. Then, one can check that the following combination of rotation and field redefinitions

$$X \mapsto X' = \frac{1}{\sqrt{1 + \rho^2 |\epsilon|^2}} \left( X - \rho \, \mathrm{Re}(\epsilon \sigma^\dagger) \right)$$

$$\sigma \mapsto \sigma' = \frac{\epsilon}{\sqrt{1 + \rho^2 |\epsilon|^2}} \left( \frac{\mathrm{Re}(\epsilon \sigma^\dagger)}{|\epsilon|^2} + \rho X \right) - \frac{i \mathrm{Im}(\epsilon \sigma^\dagger)}{\bar{\epsilon}}, \tag{322}$$

preserves the bosonic Lagrangian up to all orders in $\epsilon$.

There is a subtlety, though, in scaling the field $X$; since the theory is not conformal, we are not allowed to simply scale derivatives. Nevertheless, if we scale the radius and coupling constant at the same time, as in

$$\rho \mapsto \rho' = \frac{\rho}{\sqrt{1 + \rho^2 |\epsilon|^2}}$$

$$e^2 \mapsto e'^2 = \frac{e^2}{\sqrt{1 + \rho^2 |\epsilon|^2}}, \tag{323}$$

the combined operation is well-defined and preserves the action. Indeed, we check that rescaling the radius changes the covariant derivative

$$X = \rho^{-1} D_\tau \mapsto \sqrt{1 + \rho^2 |\epsilon|^2} \, \rho^{-1} D_\tau = \sqrt{1 + \rho^2 |\epsilon|^2} \, X, \tag{324}$$

which is cancelled by the coefficient that appears in $X'$. However, rescaling the radius also changes the space-time measure $\rho \, dr d\tau$. We absorb this factor into the coupling constant $e^2$ to keep the action invariant. We have thus shown that the $\Omega-$deformation can be absorbed into a series of field redefinitions.

In fact, this series of transformations is really a space-time rotation in disguise. To see this, we will rewrite them in terms of the real fields $\sigma = A_2 - iA_3$. Let us first consider a change of basis $A \mapsto A^\vartheta$ with

$$\begin{pmatrix} A_2^\vartheta \\ A_3^\vartheta \end{pmatrix} = \begin{pmatrix} \cos \vartheta & -\sin \vartheta \\ \sin \vartheta & \cos \vartheta \end{pmatrix} \begin{pmatrix} A_2 \\ A_3 \end{pmatrix}, \tag{325}$$

where we have defined $\zeta = \epsilon / |\epsilon| = \cos \vartheta + i \sin \vartheta$. Let us also define the angular variable $\chi$ as

$$\cos \chi = \frac{1}{\sqrt{1 + \rho^2 |\epsilon|^2}}. \tag{326}$$

In these conventions, the field rotation (322) can be written concisely as

$$\begin{pmatrix} X' \\ A_2^{\vartheta'} \end{pmatrix} = \begin{pmatrix} \cos \chi & -\sin \chi \\ \sin \chi & \cos \chi \end{pmatrix} \begin{pmatrix} X \\ A_2^\vartheta \end{pmatrix}, \quad \text{while} \quad A_3^{\vartheta'} = A_3^\vartheta. \tag{327}$$

The corresponding action on the fermionic fields is given by

$$\Psi \mapsto \Psi' = \exp\left(\frac{\vartheta}{2}\Gamma^{23}\right)\exp\left(-\frac{\chi}{2}\Gamma^{12}\right)\exp\left(-\frac{\vartheta}{2}\Gamma^{23}\right)\Psi, \tag{328}$$

in terms of the four-dimensional gamma matrices $\Gamma^{12}$ and $\Gamma^{23}$. Conjugation by $\Gamma^{23}$ implements the change of basis (325). Applying this rotation to the two-dimensional supercharges, we find the primed supercharges to be

$$\begin{aligned}
Q'_- &= Q_- \cos\frac{\chi}{2} - \zeta Q_+ \sin\frac{\chi}{2}, \\
Q'_+ &= Q_+ \cos\frac{\chi}{2} + \zeta^{-1} Q_- \sin\frac{\chi}{2}, \\
\overline{Q}'_- &= \overline{Q}_- \cos\frac{\chi}{2} - \zeta^{-1}\overline{Q}_+ \sin\frac{\chi}{2}, \\
\overline{Q}'_+ &= \overline{Q}_+ \cos\frac{\chi}{2} + \zeta\overline{Q}_- \sin\frac{\chi}{2}.
\end{aligned} \tag{329}$$

The primed supercharges in particular obey the relation

$$\{Q'_\pm, \overline{Q}'_\pm\} = (H \pm P\cos\chi) \pm \mathrm{Re}(\zeta^{-1}\widetilde{Z})\sin\chi. \tag{330}$$

This shows that the primed supersymmetry algebra, valid away from the tip of the cigar, is a rotated version by an angle $\chi$ of the standard supersymmetry algebra with central charge $\zeta^{-1}\widetilde{Z}$.

### 5.3.1   Large $\rho$ limit

The previous analysis drastically simplifies the limit in which $\rho \to \infty$. In this limit, the theory becomes very weakly coupled since $e'^2 \sim 1/\rho$. But also the transformations (327) become much simpler, merely boiling down to a swap

$$\begin{pmatrix} X' \\ A_2^{\vartheta'} \end{pmatrix} = \begin{pmatrix} 0 & -1 \\ 1 & 0 \end{pmatrix}\begin{pmatrix} X \\ A_2^\vartheta \end{pmatrix}. \tag{331}$$

The rewritten action in terms of the primed fields carries no memory of the deformation. In addition, the asymptotic cylindrical region of the cigar effectively becomes a flat strip in the large radius limit.

This implies that we are in the familiar territory of flat space supersymmetry and supersymmetric boundary conditions, albeit with the rotated supercharges (329):

$$\begin{aligned}
Q'_- &= Q_- - \zeta\, Q_+, & Q'_+ &= Q_+ + \zeta^{-1}Q_-, \\
\overline{Q}'_- &= \overline{Q}_- - \zeta^{-1}\overline{Q}_+, & \overline{Q}'_+ &= \overline{Q}_+ + \zeta\,\overline{Q}_-.
\end{aligned} \tag{332}$$

Each pair $\{Q'_\pm, \overline{Q}'_\pm\}$ thus generates a **B-type subalgebra** with phase $\pm\zeta^{-1}$ respectively. Following the discussion in §5.1, we thus find that the half-BPS boundary conditions for the cigar partition function in the $\Omega$-background are labeled by $B$-type branes with phase $\zeta = \epsilon/|\epsilon|$. We will see in §5.5 that the latter relation between $\zeta$ and $\epsilon$ is also natural from the perspective of the exact WKB analysis.

## 5.4 Non-perturbative cigar partition function

We are finally ready to define the non-perturbative partition function $\mathcal{Z}_\alpha^\zeta(\mathbf{z}, \epsilon)$.

Consider any 2d $\mathcal{N} = (2, 2)$ theory $T_\mathbf{z}$. We define its **non-perturbative partition function** $\mathcal{Z}_\alpha^\zeta(\mathbf{z}, \epsilon)$ as the cigar partition function in the $\Omega$-background, with respect to a half-BPS boundary condition specified by the vacuum $\alpha$ and the phase $\zeta$. This is illustrated in Figure 30. The non-perturbative partition function $\mathcal{Z}_\alpha^\zeta(\mathbf{z}, \epsilon)$ is locally constant with respect to the phase $\zeta$ and jumps whenever $\zeta$ crosses a BPS phase $\zeta_{\alpha\beta}$ as[39]

$$\mathcal{Z}_\alpha^\zeta(\mathbf{z}, \epsilon) \mapsto \mathcal{Z}_\alpha^\zeta(\mathbf{z}, \epsilon) \pm \mu_{\alpha\beta} \, \mathcal{Z}_\beta^\zeta(\mathbf{z}, \epsilon). \tag{333}$$

The non-perturbative partition function is also locally analytic in $\mathbf{z}$, but can jump across walls of marginal stability on $C$. As before, it is useful to consider the complete vector $(\mathcal{Z}_\alpha^\zeta(\mathbf{z}, \epsilon))_\alpha$.

We may think of the phase $\zeta$ as being independent of $\epsilon$. However, in light of the discussion in §5.3, as well as the forthcoming relation to the exact WKB analysis, it is helpful to think of $\zeta$ as specifying a phase of $\epsilon$. We may compute the partition function with respect to this phase, and analytically continue the result in $\epsilon$ afterwards.

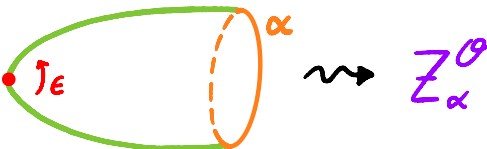

Figure 30: The non-perturbative partition function $\mathcal{Z}_\alpha^\vartheta$ is defined as the 2d $\mathcal{N} = (2, 2)$ partition function in the 2d $\Omega$-background, with $\vartheta = \arg(\epsilon)$, and with boundary condition given by the vacuum $\alpha$.

In subsections 5.6, 5.7 and 5.8 we describe the non-perturbative partition function in detail for Landau-Ginzburg models as well as for GLSMs. . In particular, we will see in detail that the non-perturbative (Higgs branch) partition function for a GLSM not only encodes vortices but also the complete BPS soliton spectrum.

## 5.5 Exact WKB analysis and open discs

Let us, however, first explain the appearance of the non-perturbative partition function $Z_\alpha^\zeta(\mathbf{z}, \epsilon)$ in the context of the **exact WKB analysis**, introduced in Kohei Iwaki's lectures as well in Marcos Mariño's book [74]. We restrict ourselves to the case when the moduli space $C_\mathbf{z}$ of deformations of the 2d theory $T_\mathbf{z}$ is complex 1-dimensional, since the exact WKB analysis is best developed then.

So let us fix a (possibly punctured) Riemann surface $C$ together with a spin structure (i.e. a choice of square-root of the canonical bundle $K_C$ on $C$), as well

---

[39]More precisely, we will see in §5.5 that equation (333) describes the jumps locally in $C$, relative to a particular choice of gauge. Globally, the jumps are described by the coefficients $\mathbf{S}_{\alpha\beta}$ in equation (348).

as a **Schrödinger operator** $d_\epsilon$ on $C$. The Schrödinger operator is a linear scalar differential operator, or more precisely, an $SL(N)$-**oper connection** on $C$. That is, the Schrödinger operator $d_\epsilon$ can locally be written in the form

$$d_\epsilon = \epsilon^N \partial_{\mathbf{z}}^N + q_2(\mathbf{z}, \epsilon)\,\epsilon^{N-2}\partial_{\mathbf{z}}^{N-2} + \ldots + q_N(\mathbf{z}, \epsilon), \qquad (334)$$

acting not on functions, but rather on $-\frac{(N-1)}{2}$-differentials on $C$. This ensures that the corresponding differential equation is globally well-defined, after specifying the transformation laws for the coefficients. Note that in the $SL(N)$ case the Schrödinger operator must have a vanishing $(N-1)$th order term in the local form.

The coefficients $q_j$ have a holomorphic dependence on $\epsilon$ and a meromorphic dependence on $\mathbf{z}$ – they are allowed to have poles at the punctures of $C$. The coefficient $q_2$ are required to transform as a projective connection on $C$, whereas we can find linear combinations $t_k$ of the $q_j$ (with $j \leq k$) and their derivatives, such that the $t_k$ transform as $k$-differentials.[40]

Traditionally, one would only consider differential operators of degree 2, but the idea can easily be extended to any degree[41] as well as to difference operators[42]. In terms of the 2d theory $T_{\mathbf{z}}$ the Schrödinger operator $d_\epsilon$ describes the quantization of the spectral curve $\Sigma$ in the $\Omega$-background with parameter $\epsilon$.

The exact WKB method is a scheme for studying the monodromy (or more generally Stokes data) of the Schrödinger operator $d_\epsilon$. One of the outputs of the exact WKB analysis is a distinguished set of local solutions $\psi_\alpha^\zeta(\mathbf{z}, \epsilon)$, labeled by the vacua $\alpha$ as well as an additional phase $\zeta$, that are analytic in $\epsilon$ and locally constant in the phase $\zeta$. Our goal in this section is to interpret these solutions physically as the non-perturbative partition functions $\mathcal{Z}_\alpha^\zeta(\mathbf{z}, \epsilon)$.

Suppose we fix a contractible open set $U \subset C$, a local coordinate $\mathbf{z}$ on $U$ and a point $z_0 \in U$. The **WKB ansatz** tells us to build formal series solutions of the form

$$\psi^{\mathrm{formal}}(\mathbf{z}; z_0, \epsilon) = \exp\left( \sum_{k=-1}^{\infty} \epsilon^k \int_{z_0}^{\mathbf{z}} S_k(z)\,dz \right). \qquad (335)$$

to the Schrödinger equation $d_\epsilon \psi(\mathbf{z}) = 0$. By plugging this formal solution into the Schrödinger equation, we immediately find that the leading exponent

$$x(\mathbf{z}) := S_{-1}(\mathbf{z}) \qquad (336)$$

of $\psi^{\mathrm{formal}}(\mathbf{z})$ is forced to obey the equation

$$x^N + p_2(\mathbf{z})\,x^{N-2} + \ldots + p_N(\mathbf{z}) = 0, \qquad (337)$$

---

[40]More details can for instance be found in §8 of [8].

[41]See [9] for many more references. Theorems, in particular regarding Borel resummability of the perturbative solutions, might be a different matter. Yet, there is a lot of numerical evidence. See for instance [8, 9, 75].

[42]The study of difference equations is a popular topic at the moment. See for instance [76, 11, 77] for connections with the exact WKB analysis.

where $p_j(\mathbf{z}) := q_j(\mathbf{z}, 0)$, which defines a spectral covering $\Sigma \subset T^*C$ with tautological 1-form $\lambda = x(\mathbf{z})\, d\mathbf{z}$.

Consistent with the notation in previous sections, we use the labels $\alpha$ to represent the sheets of $\Sigma$ and we write

$$x_\alpha(\mathbf{z}) = S_{-1}^{(\alpha)}(\mathbf{z}) \tag{338}$$

for $x(\mathbf{z})$ restricted to sheet $\alpha$ of the spectral cover $\Sigma$. The higher order expansion of

$$S_\alpha^{\text{formal}}(\mathbf{z}; \epsilon) := \sum_{k=-1}^{\infty} \epsilon^k \, S_k^{(\alpha)}(\mathbf{z}) \tag{339}$$

is then uniquely fixed.

This formal series is not convergent though, and it turns out that the best one can do is to ask for an actual solution $S_\alpha^\vartheta(\mathbf{z}, \epsilon)$ which has the expansion

$$S_\alpha^\vartheta(\mathbf{z}; \epsilon) \sim S_\alpha^{\text{formal}}(\mathbf{z}; \epsilon) \quad \text{as} \quad \epsilon \to 0, \tag{340}$$

while staying within the closed half-plane

$$\mathbb{H}_\vartheta = \{\mathrm{Re}(e^{-i\vartheta}\epsilon) \geq 0\}. \tag{341}$$

Such solutions conjecturally exist, with proofs available in the rank 2 case[43], and can be computed as the **Borel sum** of the formal solution in the direction $\vartheta$.[44] That is,

$$S_\alpha^\vartheta(\mathbf{z}; \epsilon) = B_\vartheta \, S_\alpha^{\text{formal}}(\mathbf{z}; \epsilon). \tag{342}$$

An important disclaimer is that the solutions only exist away from the $\alpha\beta$-trajectories of the **spectral network** $\mathcal{W}^\vartheta$, which is also known as the Stokes graph in the context of the exact WKB analysis. Remember that these trajectories are defined through the constraint

$$(\lambda_\alpha - \lambda_\beta)(v) \in e^{i\vartheta}\, \mathbb{R}_{>0}, \tag{343}$$

with $\lambda_\alpha = S_{-1}^{(\alpha)}(\mathbf{z})\, d\mathbf{z}$, for any tangent vector $v$ to the trajectory. Along any such trajectory, the formal solutions

$$\psi_\gamma^{\text{formal}}(\mathbf{z}; z_0, \epsilon) = \exp\left(\int_{z_0}^{\mathbf{z}} S_\gamma^{\text{formal}}(z, \epsilon)\, dz\right) \tag{344}$$

may have divergent asymptotics when we restrict the phase of $\epsilon$ to $\arg(\epsilon) = \vartheta$. Indeed, if $e^{-i\vartheta}\lambda_\alpha > 0$ while $e^{-i\vartheta}\lambda_\beta < 0$ in the direction of the $\alpha\beta$-trajectory, the formal solution

---

[43]The existence of these solutions had been a folk-theorem for some time, under some genericity assumptions, and a proof was announced by Koike-Schäfke. Nikita Nikolaev recently published a proof in fully geometric terms [78].

[44]The Borel sum $B_\vartheta$ has been introduced in previous lectures in the school, basically as an inverse Laplace transform. We won't repeat these details here, but refer to for instance §4.4 of [6] for an introduction.

$\psi_\alpha^{\text{formal}}(\mathbf{z})$ is asymptotically large along the trajectory while $\psi_\beta^{\text{formal}}(\mathbf{z})$ is asymptotically small.[45]

The name **Stokes graph** originates from the transformation properties of the Borel summed solutions $\psi_\gamma^\vartheta(\mathbf{z})$ across the $\alpha\beta$-trajectories. Indeed, suppose that we choose the branch-point $b$ at the origin of an $\alpha\beta$-trajectory as the base-point $z_0$. Then the Borel summed solutions to the left and the right of this trajectory (when they are analytically continued across the trajectory) are related by the **Stokes jump**

$$\psi_\alpha^R(\mathbf{z}; b, \epsilon) = \psi_\alpha^L(\mathbf{z}; b, \epsilon) \pm \mu_{\alpha\beta}\,\psi_\beta^L(\mathbf{z}; b, \epsilon), \tag{345}$$
$$\psi_\gamma^R(\mathbf{z}; b, \epsilon) = \psi_\gamma^L(\mathbf{z}; b, \epsilon) \quad (\text{for } \gamma \neq \alpha),$$

where $\mu_{\alpha\beta}$ is the 2d BPS index from (120). It has to be, since with this choice of Stokes factor we precisely reproduce the Cecotti-Vafa wall-crossing formula by imposing that the total monodromy of the solutions $\psi_\alpha(\mathbf{z})$ around the branch-point is the identity (in a similar computation as in §3.3). The jumps of the exact WKB solutions thus encode the 2d BPS solitons!

Comparing equation (345) with equation (333), we conclude that we may identify

$$\mathcal{Z}_\alpha^\vartheta(\mathbf{z}; \epsilon) = \psi_\alpha^\vartheta(\mathbf{z}; b, \epsilon), \tag{346}$$

if we make the specific choice $z_0 = b$ for the base-point.

It is easy to see though how to modify the Stokes transformation when we move the base point. Indeed, moving the base-point from $b$ to $z_0$ changes the exact WKB solutions $\psi_\alpha^\vartheta(\mathbf{z}; b, \epsilon)$ into

$$\psi_\alpha^\vartheta(\mathbf{z}; z_0, \epsilon) = g_\alpha\,\psi_\alpha^\vartheta(\mathbf{z}; b, \epsilon) \quad \text{with } g_\alpha = \exp\left(\frac{1}{\epsilon} \int_{z_0}^b S_\alpha^\vartheta(z, \epsilon)\,dz\right), \tag{347}$$

and thus the Stokes jump would change into

$$\mu_{\alpha\beta} \mapsto \mathbf{s}_{\alpha\beta} := g_\alpha\,\mu_{\alpha\beta}\,g_\beta^{-1}. \tag{348}$$

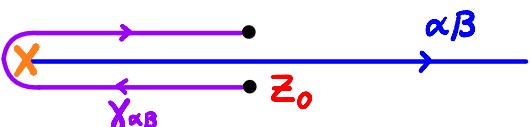

Figure 31: Projection of the detour path $\gamma_{\alpha\beta}$ on $C$.

If we would thus place the base-point $z_0$ along the $\alpha\beta$-trajectory at the point $\mathbf{z}$, as illustrated in Figure 31, we find

$$\mathbf{s}_{\alpha\beta} = \mu_{\alpha\beta}\,\exp\left(\frac{1}{\epsilon} \int_{\gamma_{\alpha\beta}} \lambda_\alpha^\vartheta\right) \quad \text{with } \lambda_\alpha^\vartheta = S_\alpha^\vartheta(z, \epsilon)\,dz, \tag{349}$$

---

[45]And in the situation that $e^{-i\vartheta}\lambda_\alpha > e^{-i\vartheta}\lambda_\beta > 0$ there has to be another overlapping trajectory, since $\sum_\alpha \lambda_\alpha = 0$.

where $\gamma_{\alpha\beta}$ is the **detour path** that starts at the lift of $z_0$ to sheet $\alpha$, runs back along the $\alpha\beta$-trajectory to the branch-point $b$, encircles the branch-point, and returns along the $\alpha\beta$-trajectory to the lift of $z_0$ to sheet $\beta$. That is, the leading contribution to the Stokes jump $\mathbf{S}_{\alpha\beta}$, in the limit $\epsilon \to 0$, is given by the volume of the special Lagrangian discs from §3.4!

### 5.5.1 Reformulation in terms of W-abelianization

The exact WKB method can be formulated geometrically through the framework of $\mathcal{W}$-**abelianization** [9]. In this perspective we interpret the Schrödinger equation $d_\epsilon$ as a flat $\epsilon$-connection $\nabla_\epsilon^{\text{flat}}$ on $C$, and the exact WKB solutions $(\psi_\alpha^\vartheta(\mathbf{z}, \epsilon))_\alpha$ as a basis of local sections that diagonalize $\nabla_\epsilon^{\text{flat}}$ in the complement of the spectral network $\mathcal{W}$. The $\mathcal{W}$-abelianization method [7, 9] then abelianizes this data into diagonal sections $\psi_\alpha^\vartheta(\mathbf{z}, \epsilon)$ for the $\mathbb{C}^*$-connection

$$\nabla_\vartheta^{\text{ab}} = \partial_z - \frac{1}{\epsilon}\lambda_\alpha^\vartheta(z) \tag{350}$$

on the spectral cover $\Sigma$. One of the advantages is that this removes the dependence on the arbitrary choice of base-point $z_0$ in the exact WKB analysis: the factors $g_\alpha$ now simply parametrize abelian gauge transformations.

We also note that the $\mathcal{W}$-nonabelianization map [2, 7], which turns "almost-flat" connections $\nabla^{\text{ab}}$ on the spectral cover $\Sigma$ into non-abelian flat connections $\nabla_\vartheta^{\text{flat}}$ on $C$, may be formulated in the language of Floer theory. In particular, Yoon Jae Nho proves in [79] that the $\mathcal{W}$-nonabelianization of the local system $\mathcal{L}$ defined by $\nabla^{\text{ab}}$ is isomorphic to a Floer cohomology local system $\text{HF}_\delta(\Sigma, \mathcal{L}; \mathbb{C})$, for small enough $\delta$.

## 5.6 Example: non-perturbative Landau-Ginzburg models

So far, we have not yet explicitly studied partition functions for the Landau-Ginzburg models with A-type boundary conditions. Such partition functions have a long history. In §3.3 of [38] it was shown that there is a holomorphic limit in which the partition function (at least when the target $X$ is Calabi-Yau) is simply given by an integral of the exponential of the holomorphic superpotential $W$ over the Lefschetz thimble $J_\alpha$, and obey the so-called $tt^*$ equation [72] (see also [19]). A similar statement was known from the theory of topological minimal models and their relation to integrable hierarchies (see for instance [80]). More recently, such statements have been verified using the method of supersymmetric localization (see for instance [59]).

These arguments may be adapted to write down the non-perturbative partition function $\mathcal{Z}_\alpha^\vartheta(\mathbf{z}, \epsilon)$ for any **Landau-Ginzburg model** (whose target $X$ is Calabi-Yau) as

$$\mathcal{Z}_\alpha^\vartheta(\mathbf{z}, \epsilon) = \int_{J_\alpha^\zeta(\mathbf{z})} d\phi \, \exp\left(\frac{iW(\phi; \mathbf{z})}{\epsilon}\right), \tag{351}$$

where the Lefschetz thimble $J_\alpha^\zeta(\mathbf{z})$ is defined as the upward flow from the critical point labelled by $\alpha$ with respect to the Morse function

$$\mathbf{h} = \mathrm{Im}(e^{-i\vartheta}W), \tag{352}$$

and where we initially assume $\arg(\epsilon) = \vartheta$ before analytically continuing in $\epsilon$.

For instance, the non-perturbative partition function for the cubic LG model with superpotential $W(\phi; \mathbf{z}) = \frac{\phi^3}{3} + \mathbf{z}\,\phi$ is a solution to the Schrödinger equation

$$(\epsilon^2 \partial_{\mathbf{z}}^2 - \mathbf{z})\, \mathcal{Z}_\pm^\vartheta(\mathbf{z}, \epsilon) = 0, \tag{353}$$

which is also known as the **Airy equation**, where $d_\epsilon = \epsilon^2 \partial_{\mathbf{z}}^2 - \mathbf{z}$ is the naive quantization of the chiral ring equation $\phi^2 = z$. The non-perturbative partition function can therefore be formulated as the Airy integral

$$\mathcal{Z}_\pm^\vartheta(\mathbf{z}, \epsilon) = \int_{J_\pm^\zeta(\mathbf{z})} d\phi \, \exp \frac{i(\phi^3 + 3\,\mathbf{z}\,\phi)}{3\epsilon}. \tag{354}$$

Let us verify that the non-perturbative LG partition function (351) is well-defined and has the correct properties. The integral in equation 351 is convergent when its integrand is bounded and decays exponentially when $\phi \to \infty$ along the contour $J_\alpha^\zeta(\mathbf{z})$. This is indeed the case since, for any choice of $\epsilon$ with $\arg(\epsilon) = \vartheta$ and any $z$, the real part of $iW(\phi; \mathbf{z})/\epsilon$, which equals the imaginary part of $-W(\phi; \mathbf{z})/\epsilon$, is bounded above by its value at the critical point and tends to minus infinity when $\phi \to \infty$.

Furthermore, something special happens when $\mathbf{z}$ and $\vartheta$ are chosen such that $\mathbf{z}$ lies on a trajectory of the spectral network $\mathcal{W}^\vartheta$, i.e. when

$$\mathrm{Im}\left( e^{-i\vartheta} \int^{\mathbf{z}} (\lambda_\alpha - \lambda_\beta) \right) = 0. \tag{355}$$

If we fix $\mathbf{z}$, the critical values for $\vartheta$ are the phases $\vartheta_{\alpha\beta}(\mathbf{z})$ for which there exists a BPS soliton in the 2d theory $T_\mathbf{z}$ with central charge $\arg(Z_{\alpha\beta}) = \vartheta_{\alpha\beta}(\mathbf{z})$. If we fix $\vartheta$, on the other hand, this happens for any $\mathbf{z} \in \mathcal{W}^\vartheta \subset \mathbb{C}_\mathbf{z}$. If $\mathbf{z}$ is part of a $\alpha\beta$-trajectory this implies that $\vartheta = \vartheta_{\alpha\beta}(\mathbf{z})$.

For any critical combination of $\mathbf{z}$ and $\vartheta$ the Lefschetz thimbles $(J_\alpha^\zeta(\mathbf{z}))_\alpha$ are not disjoint. If $\vartheta$ equals the critical phase $\vartheta_{\alpha\beta}(\mathbf{z})$, the thimble $J_\alpha^\zeta(\mathbf{z})$ contains the BPS soliton with central charge $Z_{\alpha\beta}$. If we fix $\mathbf{z}$ and vary the phase $\vartheta$ across the critical phase $\vartheta_{\alpha\beta}(\mathbf{z})$, the Lefschetz cycle $J_\alpha^\zeta(\mathbf{z})$ jumps as

$$J_\alpha^\zeta(\mathbf{z}) \mapsto J_\alpha^\zeta(\mathbf{z}) \pm \mu_{\alpha\beta} J_\beta^\zeta(\mathbf{z}), \tag{356}$$

whereas the Lefschetz cycles $J_\gamma^\zeta(\mathbf{z})$, for $\gamma \neq \alpha$, stay invariant. This implies that the collection of integrals $(\mathcal{Z}_\alpha^\vartheta(\mathbf{z}, \epsilon))_\alpha$ jumps precisely as in equation (333),

$$\mathcal{Z}_\alpha^\zeta(\mathbf{z}, \epsilon) \mapsto \mathcal{Z}_\alpha^\zeta(\mathbf{z}, \epsilon) \pm \mu_{\alpha\beta}\, \mathcal{Z}_\beta^\zeta(\mathbf{z}, \epsilon), \tag{357}$$

$$\mathcal{Z}_\gamma^\zeta(\mathbf{z}, \epsilon) \mapsto \mathcal{Z}_\gamma^\zeta(\mathbf{z}, \epsilon), \quad \text{for } \gamma \neq \alpha.$$

and therefore indeed has the properties we require for the non-perturbative partition function of the Landau-Ginzburg model.[46]

Note that the Stokes jumps (357) for the Airy integral $Z_\pm^\vartheta$ from equation (354) is the prototypical example of the Stokes phenomenon, describing the re-organization of dominant and sub-dominant integrals at the Stokes lines.[47]

### 5.6.1 W-abelianization and wall-crossing

Given an oper connection (or Schrödinger operator) $d_\epsilon$ and a phase $\zeta = e^{i\vartheta}$, the collection of integrals $Z_\alpha^\zeta(\mathbf{z}, \epsilon)$ may be re-interpreted in terms of $\mathcal{W}^\vartheta$-abelianization. The spectral network $\mathcal{W}^\vartheta$ is defined by the semi-classical limit of $d_\epsilon$, which is the limit $\epsilon \to 0$ with $\epsilon\partial_\mathbf{z} \to x$, at the phase $\zeta$.

To explain this, we need to introduce a few additional details about $\mathcal{W}^\vartheta$-abelianization. $\mathcal{W}^\vartheta$-abelianization requires a choice of $\mathcal{W}^\vartheta$-**framing** of the flat connection $\nabla$. This is a discrete choice associated with the flat connection $\nabla$. When the spectral covering is of degree 2, and the phase $\zeta$ is generic, the corresponds to a choice of eigenline of the local monodromy at each regular puncture of $C_\mathbf{z}$ as well as a choice of asymptotically small section at each marked point (labeling a Stokes sector) associated to an irregular puncture of $C_\mathbf{z}$.

If the flat connection $\nabla$ is in fact an oper connection $d_\epsilon$, the framing corresponds to local solutions $\psi$ of the oper equation $d_\epsilon\psi = 0$ with the correct properties. For instance, in the Airy example we would need to choose an asymptotically decreasing solution (for fixed phase $\zeta$) along each of the Stokes lines. The Airy integrals $Z_\pm^\vartheta$ thus constitute a natural choice of $\mathcal{W}^\vartheta$-framing. With this choice of framing, $\mathcal{W}^\vartheta$-abelianization of $d_\epsilon$ turns out to be equivalent to the exact WKB analysis of $d_\epsilon$ [9].

A $\mathcal{W}^\vartheta$-framed flat connection $\nabla$ can be $\mathcal{W}^\vartheta$-abelianized if one is able to bring the non-abelian gauge transformations across the $\alpha\beta$-trajectories in a unipotent form [7]. For an oper connection $d_\epsilon$, with the framing chosen as above, the non-abelian gauge transformations are then equivalent to the Stokes jumps (357).

Remember though that the formulae (357) are only valid locally, when we choose the base-point $z_0$ of the exact WKB analysis at the branch-point $b$ where the $\alpha\beta$-trajectory originates from. When moving around the base-point (which is needed for a global analysis), the solutions $\mathcal{Z}_\alpha^\zeta$ get multiplied by a factor $g_\alpha$, whereas the indices $\mu_{\alpha\beta}$ get replaced by the factors $\mathbf{s}_{\alpha\beta} = g_\alpha\mu_{\alpha\beta}g_\beta^{-1}$, as in equation (348). In terms of $\mathcal{W}^\vartheta$-abelianization, the factors $g_\alpha$ correspond to local abelian gauge transformations, whereas the factors $\mathbf{s}_{\alpha\beta}$ specify the unipotent gauge transformations across trajectories and can be written in terms of Wronskians of the solutions, as in

$$\mathcal{Z}_\alpha^\zeta \pm \mathbf{s}_{\alpha\beta}\mathcal{Z}_\beta^\zeta = \frac{[\mathcal{Z}_\alpha^\zeta, \mathcal{Z}_\beta^\zeta]}{[\mathcal{Z}_\gamma^\zeta, \mathcal{Z}_\beta^\zeta]}\mathcal{Z}_\gamma^\zeta \tag{358}$$

---

[46]Uniqueness statements in the exact WKB analysis imply that $\mathcal{Z}_\alpha^\zeta(\mathbf{z}, \epsilon)$ can also be written as the Borel sum in the direction $\vartheta$ of its asymptotic expansion. Details of the Borel sum for the Airy function (and its deformations) can for instance be found in [81].

[47]Be aware that Stokes lines are sometimes (confusingly) called anti-Stokes lines.

for $\gamma \neq \alpha, \beta$.

The **Cecotti-Vafa wall-crossing formula** (137) can now be obtained from the statement that the oper connection $d_\epsilon$ has trivial monodromy around an intersection point of $\alpha\beta$-trajectories. In fact, this is the geometric content of the Cecotti-Vafa wall-crossing formula, and the origin of the derivation presented in §3.3 and illustrated in Figure 12.

## 5.7   Example: non-perturbative abelian Higgs model

Remember from §4.4.2 that the abelian Higgs model is equal to the massless $\mathbb{P}^0$-model. If we turn on a twisted mass $\widetilde{m}$ for the chiral multiplet, its spectral curve is simply carved out by the equation

$$\Sigma : \quad \sigma - \widetilde{m} = \mathbf{z}, \tag{359}$$

and embedded in $T^*\mathbb{C}^*_{\mathbf{z}}$ with 1-form

$$\lambda = \frac{(\mathbf{z} + \widetilde{m})}{2\pi} \, d \log \mathbf{z}. \tag{360}$$

Its vortex partition function is given by

$$\mathcal{Z}^{\mathrm{vortex}}(\mathbf{z}, \epsilon) = \sum_{\mathfrak{m}=0}^{\infty} \mathbf{z}^{\mathfrak{m}} \prod_{k=1}^{\mathfrak{m}} \frac{1}{\widetilde{m} - k\epsilon}, \tag{361}$$

and the Schrödinger operator $d_\epsilon$, which annihilates the vortex partition function, is given by

$$d_\epsilon = -\epsilon \, \mathbf{z} \partial_{\mathbf{z}} + \mathbf{z} + \widetilde{m} \tag{362}$$

Even though the spectral covering $\Sigma \to \mathbb{C}^*_{\mathbf{z}}$ is single-sheeted, the abelian Higgs model does admit BPS solitons. At any point $\mathbf{z} \in \mathbb{C}^*_{\mathbf{z}}$ there is a family of BPS self-solitons, indexed by $n \in \mathbb{Z}$, with twisted central charge

$$\widetilde{Z}_n = n \int_\gamma \lambda = \frac{n\widetilde{m}}{2\pi} \int_\gamma d \log \mathbf{z} = in\widetilde{m} \tag{363}$$

where the 1-cycle $\gamma$ is represented by the path $\gamma(t) = \mathbf{z} \, e^{it}$ in the $\mathbf{z}$-plane. That is, there are no walls of marginal stability in this model, but there are BPS walls at the phases

$$\vartheta_{\mathrm{BPS}} = \arg(\widetilde{m}) - \frac{\pi}{2} \quad \text{and} \quad \vartheta_{\mathrm{BPS}} + \pi. \tag{364}$$

Even though you might think that (361) is the complete answer, we will see in the next subsection that there is a natural way to encode the BPS self-solitons in a non-perturbative partition function that jumps across the phases $\vartheta_{\mathrm{BPS}}$ and $\vartheta_{\mathrm{BPS}} + \pi$.

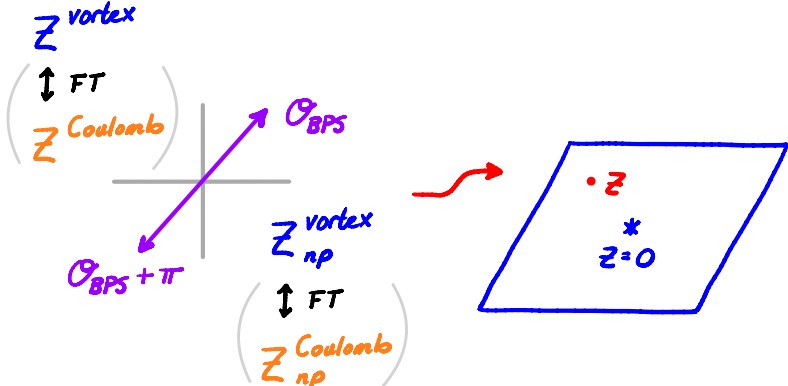

Figure 32: The partition functions that fully characterise the non-perturbative structure of the abelian Higgs model. On the left: the central charge plane with the two active BPS rays corresponding to the family of BPS self-solitons. On the right: the z-plane. Wall-crossing only happens in the central charge plane in this example.

### 5.7.1 Remark: dual Coulomb branch description

Remember from §4.4.4 that the Schrödinger operator on the $\sigma$-plane is instead given by the difference operator

$$D_\epsilon = e^{\epsilon \partial_\sigma} - \sigma + \widetilde{m}, \tag{365}$$

and the corresponding Coulomb branch partition function by

$$\widetilde{\mathcal{Z}}^{\text{Coulomb}}(\sigma, \epsilon) = \epsilon^{\frac{\sigma - \widetilde{m}}{\epsilon}} \, \Gamma\left(\frac{\sigma - \widetilde{m}}{\epsilon}\right). \tag{366}$$

Also remember that this partition function is related to the vortex partition function (361) by the Fourier transform (285).

In this dual description it is easy to see that there is another solution to the difference equation $D_\epsilon \widetilde{Z} = 0$, given by

$$\widetilde{\mathcal{Z}}^{\text{Coulomb}}_{\text{np}}(\sigma, \epsilon) = \left(-\frac{\epsilon}{\mu}\right)^{\frac{\sigma - \widetilde{m}}{\epsilon}} \frac{2\pi i}{\Gamma\left(1 - \frac{\sigma - \widetilde{m}}{\epsilon}\right)}. \tag{367}$$

The Coulomb branch partition functions (366) and (367) are related by the jump

$$\frac{\widetilde{\mathcal{Z}}^{\text{Coulomb}}(\sigma, \epsilon)}{\widetilde{\mathcal{Z}}^{\text{Coulomb}}_{\text{np}}(\sigma, \epsilon)} = \frac{e^{\frac{\pi i (\sigma + \widetilde{m})}{\epsilon}}}{2i \sin\left(\frac{\pi(\sigma - \widetilde{m})}{\epsilon}\right)} = \left(1 - e^{-\frac{2\pi i (\sigma - \widetilde{m})}{\epsilon}}\right)^{-1}, \tag{368}$$

where we have used the reflection equation for the gamma-function in the first equality. In [21] we show that this factor is due to self-solitons in the $\sigma$-plane with central charge

$$Z_n = n \int_{\widetilde{\gamma}} \widetilde{\lambda} = n \int_{\widetilde{\gamma}} \frac{\log \mathbf{z}}{2\pi} \, d\sigma = n \int_{\widetilde{\gamma}} \frac{\log(\sigma - \widetilde{m})}{2\pi} \, d\sigma = in(\sigma - \widetilde{m}), \tag{369}$$

where $n \in \mathbb{Z}^*$ and the 1-cycle $\widetilde{\gamma}$ is represented by the path $\widetilde{\gamma}(t) = (\sigma - \widetilde{m})\, e^{it}$ in the $\sigma$-plane. More specifically, the jump (368) is a Stokes jump due to the family of self-solitons with $n > 0$ when $|Z_n| < 1$ and $n < 0$ when $|Z_n| > 1$.

Note that the self-solitons with $n < 0$ have phase

$$\widetilde{\vartheta}_{\text{BPS}} = \arg Z_n = \arg(\sigma - \widetilde{m}) - \frac{\pi}{2}, \tag{370}$$

whereas the self-solitons with $n > 0$ have phase $\widetilde{\vartheta}_{\text{BPS}} + \frac{\pi}{2}$. The partition function $\widetilde{\mathcal{Z}}^{\text{Coulomb}}$ and $\widetilde{\mathcal{Z}}^{\text{Coulomb}}_{\text{np}}$ can be obtained as the Borel sums in the direction $\vartheta$ with

$$\widetilde{\vartheta}_{\text{BPS}} \lessgtr \vartheta \lessgtr \widetilde{\vartheta}_{\text{BPS}} + \pi, \tag{371}$$

respectively.[48] Since we have that $\arg(Z_n/\epsilon) = \pi$ for $n > 0$ at phase $\vartheta = \arg \epsilon = \widetilde{\vartheta}_{\text{BPS}}$, whereas $\arg(Z_n/\epsilon) = \pi$ for $n < 0$ at phase $\vartheta = \arg \epsilon = \widetilde{\vartheta}_{\text{BPS}} + \pi$, the two Borel sums are related by the Stokes jump (368) across the critical phase $\widetilde{\vartheta}_{\text{BPS}}$. A similar statement with $n < 0$ holds at phase $\widetilde{\vartheta}_{\text{BPS}} + \pi$.

Taking the Fourier transform gives the corresponding statement on the z-plane. Defining $x = (\sigma - \widetilde{m})/\epsilon$, we find

$$
\begin{aligned}
\frac{1}{2\pi} \widetilde{\mathcal{Z}}^{\text{Coulomb}}_{\text{np}}(\sigma, \epsilon) &= \frac{1}{2\pi} \left( 1 - e^{\frac{2\pi i x}{\epsilon}} \right) \widetilde{\mathcal{Z}}^{\text{Coulomb}}(\sigma, \epsilon) = \\
&= \int_{-\infty}^{\infty} d\widetilde{\tau} \left( 1 - e^{\frac{2\pi i x}{\epsilon}} \right) e^{\frac{2\pi \widetilde{\tau} \sigma}{\epsilon}} \mathcal{Z}^{\text{vortex}}(\mathbf{z}, \epsilon) = \\
&= \int_{-\infty}^{\infty} d\widetilde{\tau}\, e^{\frac{2\pi \widetilde{\tau} \sigma}{\epsilon}} \mathcal{Z}^{\text{vortex}}(\mathbf{z}, \epsilon) - e^{-\frac{2\pi i \widetilde{m}}{\epsilon}} \int_{-\infty}^{\infty} d\widetilde{\tau}\, e^{\frac{2\pi (\widetilde{\tau}+i)\sigma}{\epsilon}} \mathcal{Z}^{\text{vortex}}(\mathbf{z}, \epsilon) = \\
&= \int_{-\infty}^{\infty} d\widetilde{\tau}\, e^{\frac{2\pi \widetilde{\tau} \sigma}{\epsilon}} \left( 1 - e^{-\frac{2\pi i \widetilde{m}}{\epsilon}} \right) \mathcal{Z}^{\text{vortex}}(\mathbf{z}, \epsilon),
\end{aligned}
\tag{372}
$$

since $\mathbf{z}$ stays invariant under $\tau \mapsto \tau - 1$. That is,

$$\mathcal{Z}^{\text{vortex}}_{\text{np}}(\mathbf{z}, \epsilon) = \left( 1 - e^{-\frac{2\pi i \widetilde{m}}{\epsilon}} \right) \mathcal{Z}^{\text{vortex}}(\mathbf{z}, \epsilon), \tag{373}$$

or inversely,

$$\mathcal{Z}^{\text{vortex}}(\mathbf{z}, \epsilon) = \left( \sum_{n>0} e^{-\frac{2\pi i n \widetilde{m}}{\epsilon}} \right) \mathcal{Z}^{\text{vortex}}_{\text{np}}(\mathbf{z}, \epsilon), \tag{374}$$

at the critical phase $\vartheta_{\text{BPS}} + \pi$, when $\arg(in\widetilde{m}/\epsilon) = 0$. That is, $\mathcal{Z}^{\text{vortex}}_{\text{np}}(\mathbf{z}, \epsilon)$ only differs from $\mathcal{Z}^{\text{vortex}}(\mathbf{z}, \epsilon)$ by a constant in $\widetilde{m}/\epsilon$ – which is all that it could differ by, given that both satisfy the same differential equation (362) – which encodes the family of self-solitons with central charge $\widetilde{Z}_n$ for $n \lessgtr 0$. The final picture is illustrated in Figure 32.

---

[48]See for instance §4.4 of [6] for more details regarding Borel sums of the gamma-function.

## 5.8 Example: non-perturbative $\mathbb{P}^1$- model

In §4.4.3 we computed the Higgs branch partition function $\mathcal{Z}_\alpha(\mathbf{z}, \epsilon)$ for the $\mathbb{P}^1$-model and observed that it was annihilated by the Schrödinger operator

$$d_\epsilon = \epsilon^2 \left( \mathbf{z}^2 \partial_\mathbf{z}^2 - \mathbf{z}\partial_\mathbf{z} \right) - \widetilde{m}^2 - \mu^2 \, \mathbf{z}. \tag{375}$$

Moreover, we found that the Schrödinger operator $d_\epsilon$ reduces in the semi-classical approximation to the spectral curve $\Sigma$. Previously, in §4.3.2, we plotted the family of spectral networks $\mathcal{W}^\vartheta$ for the $\mathbb{P}^1$-model, and used this to derive its spectrum of BPS solitons across the parameter space $C = \mathbb{C}^*_\mathbf{z}$. In this section we combine both ingredients to determine the non-perturbative partition function $\mathcal{Z}_\alpha^\vartheta(\mathbf{z}, \epsilon)$ and to show its relation to the Higgs branch partition function.

We work in the abelianised setting. This means that we treat the exact WKB solutions $\psi_\alpha^\vartheta(\mathbf{z}; b, \epsilon)$ as local sections of a $\mathbb{C}^*$-bundle on the $\alpha$th sheet of the spectral curve, in the complement of $\pi^{-1}(\mathcal{W})$ where $\pi : \Sigma \to C$ is the spectral covering map. We work on $C$ by introducing a branch-cut between the branch-point $\mathbf{z} = -\widetilde{m}^2$ and $\mathbf{z} = \infty$, trivialising the spectral covering (i.e. fixing the labelling of the sheets at any given point $\mathbf{z}$). Across this branch-cut, the local solutions must be exchanged as

$$\begin{pmatrix} \psi_1 \\ \psi_2 \end{pmatrix} \mapsto \begin{pmatrix} 0 & 1 \\ -1 & 0 \end{pmatrix} \begin{pmatrix} \psi_1 \\ \psi_2 \end{pmatrix}. \tag{376}$$

As the $\mathbb{C}^*$-connection on $\Sigma$ abelianises an $SL(2,\mathbb{C})$ connection on $C$, abelian parallel transport (as well as abelian gauge transformations) act diagonally on the local solutions as

$$\begin{pmatrix} \psi_1 \\ \psi_2 \end{pmatrix} \mapsto \begin{pmatrix} \eta & 0 \\ 0 & \eta^{-1} \end{pmatrix} \begin{pmatrix} \psi_1 \\ \psi_2 \end{pmatrix} \tag{377}$$

for some $\eta \in \mathbb{C}^*$. Note that, as the $\mathbb{C}^*$-connection is flat, the abelian holonomy is only non-trivial around the singularities $\mathbf{z} = 0$ and $\mathbf{z} = \infty$. To keep track of this, we also introduce a holonomy-cut in the $\mathbf{z}$-plane.

Finally, local solutions on either side of an $\alpha\beta$-trajectory are related by a Stokes jump (or S-matrix)

$$S_{12} = \begin{pmatrix} 1 & \pm\mathbf{s}_{12} \\ 0 & 1 \end{pmatrix} \quad \text{or} \quad S_{21} = \begin{pmatrix} 1 & 0 \\ \pm\mathbf{s}_{21} & 1 \end{pmatrix}, \tag{378}$$

where, as explained in §5.5, $s_{\alpha\beta}$ is the product of the abelian holonomy along the associated detour path $\gamma_{\alpha\beta}$ and the corresponding BPS index $\mu_{\alpha\beta}$.

### 5.8.1 Fenchel-Nielsen phase

To start with, we consider the spectral network $\mathcal{W}^\vartheta$ at phases

$$\vartheta_{\text{FN}} = \arg Z_1^+ = \arg(i\widetilde{m}) \quad \text{or} \quad \vartheta_{\text{FN}} + \pi. \tag{379}$$

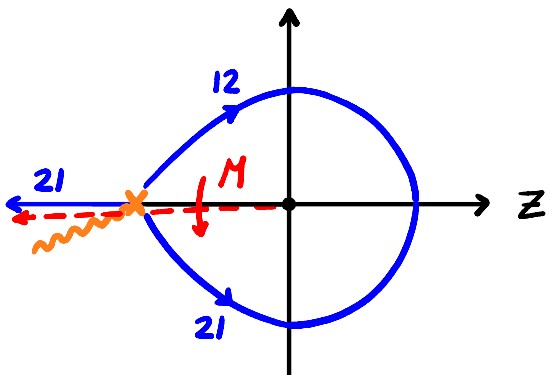

Figure 33: The spectral network $\mathcal{W}^{\vartheta_{\mathrm{FN}}}$ (in red) for the $\mathbb{P}^1$-model at $\widetilde{m} = 1$, so that $\vartheta_{\mathrm{FN}} = \pi/2$, together with a choice of a branch-cut (in orange) and a choice of a monodromy/holonomy cut (in red).

These networks encode BPS self-solitons with central charges $Z^+_{\alpha\alpha}$. Because they contain a maximal number of ring domains, they are sometimes called of Fenchel-Nielsen type [7]. Note that the two spectral networks are related by swapping the labels $12 \leftrightarrow 21$ of all trajectories. For concreteness, we will focus on the network at phase $\vartheta_{\mathrm{FN}}$ below. This network is illustrated in Figure 33.

The jumping behaviour of the local solutions $\psi_\alpha$ across $\mathcal{W}$-trajectories and (branch and holonomy) cuts is highly constraining and, without actually computing any Borel sum (!), we will argue below that it in fact determines them uniquely (up to abelian gauge transformations) across the entire $\mathbf{z}$-plane.

Define the counter-clockwise monodromy operator $M$ as

$$M : \mathbf{z} \mapsto e^{2\pi i}\, \mathbf{z}. \tag{380}$$

and consider the solutions $\psi^{\vartheta_{\mathrm{FN}}}_{\alpha,\mathrm{weak}}$ in the weak-coupling regime – remember that this is the region inside the saddle trajectory in Figure 33. Since there are no trajectories crossing the weak-coupling region, the monodromy must act diagonally on the local solutions $\psi^{\vartheta_{\mathrm{FN}}}_{\alpha,\mathrm{weak}}$ as

$$M : \begin{pmatrix} \psi_{1,\mathrm{weak}} \\ \psi_{2,\mathrm{weak}} \end{pmatrix} \mapsto \begin{pmatrix} \lambda & 0 \\ 0 & \lambda^{-1} \end{pmatrix} \begin{pmatrix} \psi_{1,\mathrm{weak}} \\ \psi_{2\mathrm{weak}} \end{pmatrix}, \tag{381}$$

where $\lambda^{\pm 1}$ are the eigenvalues of the Schrödinger operator (375). That is, the monodromy $M$ acts just as the abelian holonomy around $\mathbf{z} = 0$.

Recall that we have met solutions with these properties before. Indeed, the Higgs branch partition functions (276) of the $\mathbb{P}^1$-model are proportional to **Bessel functions** of the first kind

$$\mathcal{Z}_1(\mathbf{z}, \epsilon) = \left( -\frac{\pi e^{-i\pi\nu/2}}{\sin(\pi\nu)} \right) J_\nu(x), \quad \mathcal{Z}_2(\mathbf{z}, \epsilon) = \left( \frac{\pi e^{i\pi\nu/2}}{\sin(\pi\nu)} \right) J_{-\nu}(x), \tag{382}$$

where

$$x = \frac{2i\,\mu\,\mathbf{z}^{1/2}}{\epsilon} \quad \text{and} \quad \nu = \frac{2\widetilde{m}}{\epsilon}, \tag{383}$$

and hence

$$\lambda^{\pm 1} = e^{\pm \pi i \nu} = e^{\pm \pi \frac{Z_1^+}{\epsilon}}. \tag{384}$$

Note that at phase $\arg(\epsilon) = \vartheta_{\mathrm{FN}}$ we find that

$$\frac{Z_1^+}{\epsilon} \in \mathbb{R}_{>0}, \tag{385}$$

so that at this phase the eigenvalue $\lambda^- = \frac{1}{\lambda_+}$ takes real values between 0 and 1.

This has important consequences. Since the Bessel functions $J_{\pm\nu}(x)$ are the only solutions to the Schrödinger equation (375) on which the monodromy operator $M$ acts diagonally, we must have that

$$\psi^{\vartheta_{\mathrm{FN}}}_{\alpha,\mathrm{weak}}(\mathbf{z}, \epsilon) = c_\alpha\,\mathcal{Z}_\alpha(\mathbf{z}, \epsilon) \tag{386}$$

for some $c_\alpha \in \mathbb{C}^*$. Moreover, the WKB solutions ought to form an $SL(2)$-basis, so it makes sense to normalise them such that their Wronskian[49]

$$[\psi^{\vartheta_{\mathrm{FN}}}_{1,\mathrm{weak}}(\mathbf{z}, \epsilon), \psi^{\vartheta_{\mathrm{FN}}}_{2,\mathrm{weak}}(\mathbf{z}, \epsilon)] = \frac{1}{\mathbf{z}}. \tag{387}$$

This fixes the coefficient $c_2$ in terms of $c_1$.

It follows from equation (385) that we have ordered the eigenvalues $\lambda^{\pm 1}$ such that $\psi_{2,\mathrm{weak}}(\mathbf{z}, \epsilon)$ is the asymptotically small WKB solution, and $\psi_{1,\mathrm{weak}}(\mathbf{z}, \epsilon)$ the asymptotically large one. It is also important to note that equation (384) implies that there are no higher $\epsilon$-corrections to the abelian holonomy around $\mathbf{z} = 0$, i.e. the semi-classical value $\pi i \nu = \frac{2\pi i \widetilde{m}}{\epsilon}$ is exact!

Next, we move to the strong-coupling region; since it is crossed by a trajectory, the non-abelian monodromy operator $M$ does *not* act diagonally on a basis of local solutions $\psi^{\vartheta_{\mathrm{FN}}}_{\alpha,\mathrm{strong}}(\mathbf{z}, \epsilon)$. The abelian holonomy, however, does act diagonally as

$$\begin{pmatrix} \psi^{\vartheta_{\mathrm{FN}}}_{1,\mathrm{strong}} \\ \psi^{\vartheta_{\mathrm{FN}}}_{2,\mathrm{strong}} \end{pmatrix} \mapsto \begin{pmatrix} \lambda & 0 \\ 0 & \lambda^{-1} \end{pmatrix} \begin{pmatrix} \psi^{\vartheta_{\mathrm{FN}}}_{1,\mathrm{strong}} \\ \psi^{\vartheta_{\mathrm{FN}}}_{2,\mathrm{strong}} \end{pmatrix}. \tag{388}$$

With the choices of branch and holonomy cuts made in Figure 33, we find that the total action of the monodromy operator $M$ on the local solutions $\psi^{\vartheta_{\mathrm{FN}}}_{\alpha,\mathrm{strong}}(\mathbf{z}, \epsilon)$ is given by the product

$$M = B_r H^{\mathrm{ab}} S_{21} = \begin{pmatrix} 0 & -1 \\ 1 & 0 \end{pmatrix} \begin{pmatrix} \lambda & 0 \\ 0 & \lambda^{-1} \end{pmatrix} \begin{pmatrix} 1 & (1 + \lambda^{-2}) \\ 0 & 1 \end{pmatrix} =$$
$$= \begin{pmatrix} 0 & -\lambda^{-1} \\ \lambda & (\lambda + \lambda^{-1}) \end{pmatrix}, \tag{389}$$

---

[49]Note that the factor $\mathbf{z}^{-1}$ gets cancelled when we rescale the solutions by a factor $\sqrt{\mathbf{z}}$, as we do in equation (279) to turn the differential operator $d_\epsilon$ into an $SL(2)$-oper.

of the branch-cut matrix $B_r$, the abelian holonomy $H^{\mathrm{ab}}$ and the S-matrix $S_{21}$ associated with crossing the 21-trajectory.

Note that the non-vanishing 12-entry in the S-matrix $S_{21}$ is consistent with $\psi_{2,\mathrm{strong}}$ being the asymptotically small WKB solution. Furthermore, the 1's in the diagonal of the S-matrix $S_{21}$ imply that we have fixed the abelian gauge such that the WKB solutions $\psi_{\alpha,\mathrm{strong}}^{\vartheta_{\mathrm{FN}}}$ have their base-point at the branch-point. And finally, note that the off-diagonal entry $(1 + \lambda^{-2})$ in the $S$-matrix is a sum of two contributions. This is because the 21-trajectory crossing the strong-coupling region is doubled. One trajectory can be traced back to the branch-point directly, and contributes as 1, whereas the other winds around the singularity before g etting back to the branch-point, and thus contributes as $\lambda^{-2}$.

The resulting monodromy $M$ is precisely that of the **Hankel functions** $H_\nu^\alpha(x)$. These are other well-known solutions of the Schrödinger equation (375). Their asymptotics

$$H_\nu^\alpha(x) \sim -\frac{2}{\pi x} e^{\pm i\left(x - \frac{1}{2}\pi\nu - \frac{1}{4}\pi\right)},\tag{390}$$

when $x \to \infty$, are the asymptotics we are looking for: when $\mu > 0$ and $\arg(\epsilon) = \vartheta_{\mathrm{FN}}$, they grow/decay fastest along the ray with

$$\arg \mathbf{z} = 2\vartheta_{\mathrm{FN}}.\tag{391}$$

The WKB solutions $\psi_{\alpha,\mathrm{strong}}^{\vartheta_{\mathrm{FN}}}(\mathbf{z}, \epsilon)$ must thus be identified with the Hankel functions

$$\psi_{\alpha,\mathrm{strong}}^{\vartheta_{\mathrm{FN}}}(\mathbf{z}, \epsilon) = \frac{b\left(e^{i\pi\nu} - 1\right)}{i\pi} H_\nu^\alpha(x)\tag{392}$$

for some $b \in \mathbb{C}^*$, which is again determined by requiring that the Wronskian of this basis is equal to $1/\mathbf{z}$.

We have thus determined expressions for the local solutions $\psi_\alpha^{\vartheta_{\mathrm{FN}}}$ in the weak, as well as the strong coupling regions, up to the overall coefficient $c_1$. We did this solely by considering the action of the monodromy in each region. We should now compare the solutions on either side of the wall of marginal stability. The well-known relations between the Bessel and Hankel functions,

$$\begin{pmatrix} J_\nu \\ J_{-\nu} \end{pmatrix} = \frac{1}{2} \begin{pmatrix} 1 & 1 \\ e^{i\pi\nu} & e^{-i\pi\nu} \end{pmatrix} \begin{pmatrix} H_\nu^1 \\ H_\nu^2 \end{pmatrix}.\tag{393}$$

imply that

$$\begin{pmatrix} \psi_{1,\mathrm{weak}}^{\vartheta_{\mathrm{FN}}} \\ \psi_{2,\mathrm{weak}}^{\vartheta_{\mathrm{FN}}} \end{pmatrix} = C_{\mathrm{sw}} \begin{pmatrix} \psi_{1,\mathrm{strong}}^{\vartheta_{\mathrm{FN}}} \\ \psi_{2,\mathrm{strong}}^{\vartheta_{\mathrm{FN}}} \end{pmatrix} \quad \text{with} \quad C_{\mathrm{sw}} = \frac{1}{b} \begin{pmatrix} -c_1 & -c_1 \\ c_2\, e^{2i\pi\nu} & c_2 \end{pmatrix}.\tag{394}$$

Since the local WKB bases abelianise an $SL(2)$ flat connection, the non-abelian parallel transport matrix $C_{\mathrm{sw}}$ ought to be an $SL(2)$-matrix. This implies that the constants $b, c_1, c_2$ are related as

$$\frac{c_1 c_2}{b^2} = -\frac{1}{1 - e^{2\pi i\nu}}\tag{395}$$

and thus determines $c_1$ in terms of the already fixed $c_2$ and $b$.

Furthermore, the expression (394) must be consistent with crossing the wall of marginal stability (i.e. the saddle trajectory) up to a gauge transformation on each side. We note that the equation

$$\frac{1}{b}\begin{pmatrix} \rho_1 & 0 \\ 0 & 1/\rho_1 \end{pmatrix}\begin{pmatrix} -c_1 & -c_1 \\ c_2\,e^{2\pi i\nu} & c_2 \end{pmatrix}\begin{pmatrix} \rho_2 & 0 \\ 0 & 1/\rho_2 \end{pmatrix} = \\ = \begin{pmatrix} 1 & 0 \\ A & 1 \end{pmatrix}\begin{pmatrix} 1 & B \\ 0 & 1 \end{pmatrix}, \tag{396}$$

with $c_1$, $c_2$ and $b$ all previously fixed, has the general solution

$$A = \frac{\rho_2^2}{(1 - e^{-2\pi i\nu})}, \qquad B = -\frac{1}{\rho_2^2},$$
$$\rho_1\rho_2 = -\frac{c_1 e^{-\pi i\nu}}{b(1 - e^{2\pi i\nu})^2}. \tag{397}$$

That is, all coefficients $A$, $B$ and $\rho_1$ can be fixed in terms of $\rho_2$.

This implies that the rescaled bases of strong and weak-coupling WKB solutions can be related

$$\begin{pmatrix} \rho_1^{-1}\psi_{1,\text{weak}}^{\vartheta_{\text{FN}}} \\ \rho_1\psi_{2,\text{weak}}^{\vartheta_{\text{FN}}} \end{pmatrix} = S_{12}\,S_{21}\begin{pmatrix} \rho_2^{-1}\,\psi_{1,\text{strong}}^{\vartheta_{\text{FN}}} \\ \rho_2\,\psi_{2,\text{strong}}^{\vartheta_{\text{FN}}} \end{pmatrix} \tag{398}$$

through the product $S_{12}\,S_{21}$ of S-matrices

$$S_{12} = \begin{pmatrix} 1 & 0 \\ A & 1 \end{pmatrix} \quad \text{and} \quad S_{21} = \begin{pmatrix} 1 & B \\ 0 & 1 \end{pmatrix}, \tag{399}$$

where $\rho_1$ and $\rho_2$ are related through equations (397).

Remember that we explicitly placed the base-point for the strong-coupling WKB basis at the branch-point. Yet, so far we did not discuss the base-point for the weak-coupling WKB basis. Relation (398) with $\rho_2 = 1$ tells us that the weak-coupling basis with base-point at the branch-point is given by

$$\begin{pmatrix} \rho & 0 \\ 0 & \rho^{-1} \end{pmatrix}\begin{pmatrix} \psi_{1,\text{weak}}^{\vartheta_{\text{FN}}} \\ \psi_{2,\text{weak}}^{\vartheta_{\text{FN}}} \end{pmatrix} \tag{400}$$

with $\rho = -\frac{b}{c_1}\,e^{\pi i\nu}\,(1 - e^{2\pi i\nu})^2$. Hence, equation (398), modified to

$$\begin{pmatrix} \rho & 0 \\ 0 & \rho^{-1} \end{pmatrix}\begin{pmatrix} \psi_{1,\text{weak}}^{\vartheta_{\text{FN}}} \\ \psi_{2,\text{weak}}^{\vartheta_{\text{FN}}} \end{pmatrix} = \begin{pmatrix} \eta^{-1} & 0 \\ 0 & \eta \end{pmatrix} S_{12}\,S_{21}\begin{pmatrix} \eta & 0 \\ 0 & \eta^{-1} \end{pmatrix}\begin{pmatrix} \psi_{1,\text{strong}}^{\vartheta_{\text{FN}}} \\ \psi_{2,\text{strong}}^{\vartheta_{\text{FN}}} \end{pmatrix}, \tag{401}$$

shows the relation between the strong and weak-coupling WKB bases at any point along the wall of marginal stability, where $\text{diag}(\eta, \eta^{-1})$ and its inverse encode the required parallel transport. This is illustrated in Figure 34.

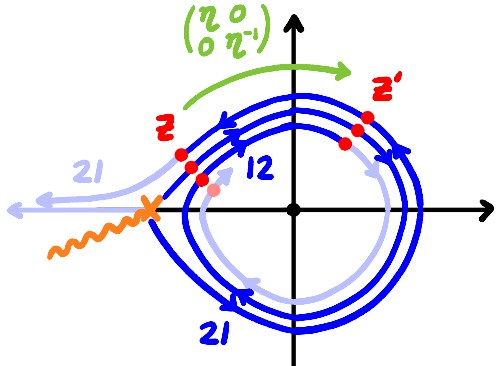

Figure 34: Cartoon of the spectral network $\mathcal{W}^{\vartheta^-_{\text{FN}}}$, when the 12-trajectory unwinds into an infinite spiral. The product of S-matrices $S_{12}S_{21}$ describes the wall-crossing across the double trajectory at the point $z$ (which is meant to be just above the branch-point), whereas the sandwiched product $\text{diag}(\eta^{-1}, \eta)\, S_{12}S_{21}\, \text{diag}(\eta, \eta^{-1})$ describes the wall-crossing at the point $z'$.

On top of that, the S-matrix entries $A$ and $B$ are related through the equation

$$AB = -\frac{1}{(1 - e^{-2\pi i \nu})}. \tag{402}$$

We interpret this relation in terms of the spectral network at the phase $\vartheta = \vartheta^-_{\text{FN}}$, i.e. just below its critical value, where the strong and the weak-coupling region are connected by crossing a single 21-trajectory before traversing an infinite family of 12-trajectories. The coefficient

$$B = -\frac{1}{\rho_2^2} = -\eta^2 \tag{403}$$

encodes the abelian holonomy along the detour path $\gamma^+_{21}$, whereas the coefficient $A$ can be expanded as

$$A = \rho_2^2 \sum_{k=0}^{\infty} e^{-2\pi i k \nu} = \eta^{-2} \sum_{k=0}^{\infty} e^{-2\pi i k \nu} \tag{404}$$

and encodes the abelian holonomy along the infinite family of detour paths $\gamma^k_{12}$, illustrated back in Figure 22, that wind $k$ times around the puncture at $\mathbf{z} = 0$. Indeed, we check that the (22)-coefficient of the matrix $S_{12}\, S_{21}$ is given by

$$1 + AB = -\sum_{k=1}^{\infty} e^{-2\pi i k \nu}, \tag{405}$$

and therefore encodes the family of self-solitons with central charge $Z_2^+ = -2i\widetilde{m}$.

Finally, remember from equation (349) that $A$ and $B$ are in fact equal to plus or minus the abelian parallel transport times the corresponding BPS index $\mu_{\alpha\beta} \in \mathbb{Z}$.

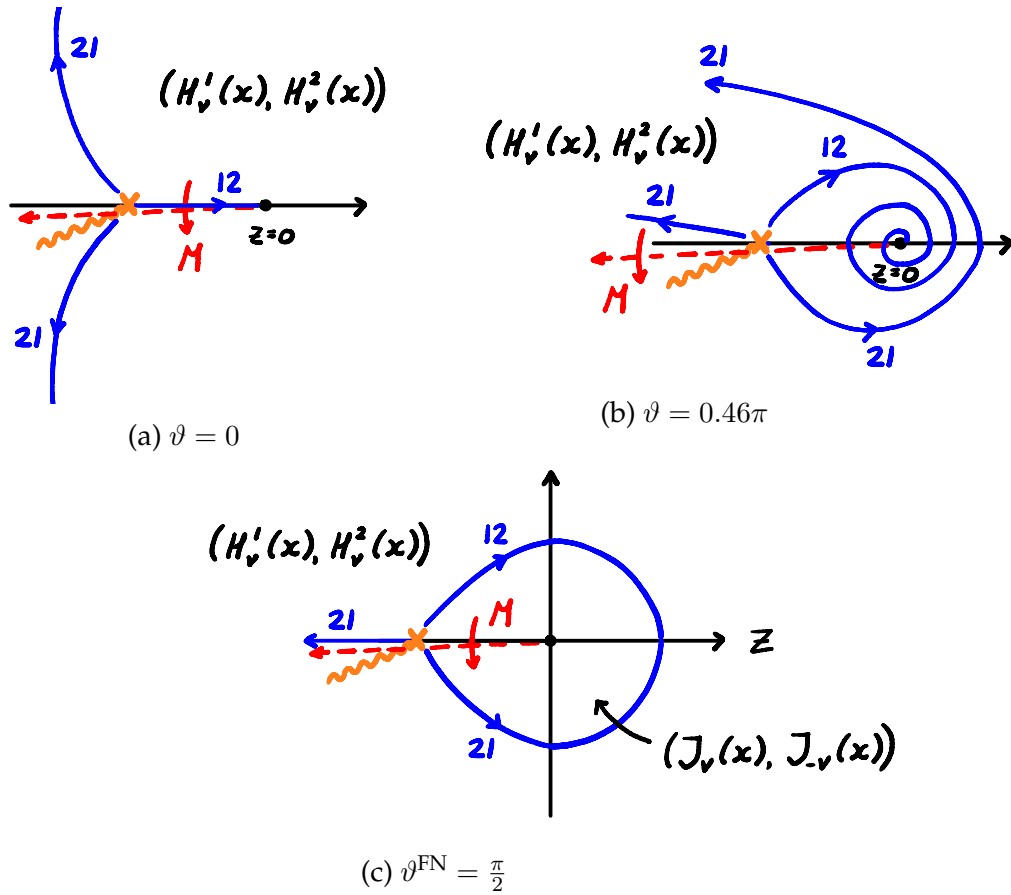

(a) $\vartheta = 0$

(b) $\vartheta = 0.46\pi$

(c) $\vartheta^{\text{FN}} = \frac{\pi}{2}$

Figure 35: Spectral networks $\mathcal{W}^\vartheta$ for the $\mathbb{P}^1$-model, with $\mu = \widetilde{m} = 1$, at a phases $\vartheta = 0$, $\vartheta = 0.46\pi$ and $\vartheta = \pi/2$, respectively. The non-perturbative Higgs branch partition function $\mathcal{Z}^\vartheta_\alpha(\mathbf{z}, \epsilon)$ is equal to the exact WKB solution $\psi^\vartheta_\alpha(\mathbf{z}, \epsilon)$ in the cell of $C \backslash \mathcal{W}^\vartheta$ that $\mathbf{z}$ is part of. The WKB solutions are proportional to Hankel or Bessel functions, as indicated above. In the cells where the WKB solutions are not explicitly written down, they may be obtained by a simple wall-crossing argument.

Now, relation (405) tells us that the product $\mu^+_{21}\mu^k_{12}$ of BPS indices is equal to $\pm 1$, and hence the individual indices $\mu^+_{21}$ and $\mu^k_{12}$ must also be equal to $\pm 1$. This is the best we can do currently, given that we have not been careful with these (somewhat subtle) signs throughout these notes.

   An analogous discussion holds at phase $\vartheta = \vartheta^+_{\text{FN}}$ where we first cross a single 12-trajectory before crossing infinite an family of 21-trajectories. This concludes our analysis at phase $\vartheta_{\text{FN}}$ (and similarly at $\vartheta_{\text{FN}} + \pi$), which is summarized in Figure 35(c).

### 5.8.2   Away from the Fenchel-Nielsen phase

If we move the phase $\vartheta$ away from the critical phase $\vartheta_{\text{FN}}$, the spectral network $\mathcal{W}^\vartheta$ starts to unwind, as illustrated again in Figure 35. The basis of WKB solutions stays the same at a given point $\mathbf{z} \in C$, as long as we do not cross the unfolding spectral

network $\mathcal{W}^\vartheta$. This implies that the strong-coupling basis

$$\psi^\vartheta_{\alpha,\text{strong}}(\mathbf{z},\epsilon) \sim H^\alpha_\nu(x). \tag{406}$$

remains the WKB basis in one of the cells in $C\backslash\mathcal{W}^\vartheta$ when $\vartheta$ changes.

In particular, at phase $\vartheta = \vartheta^{\text{FN}} - \pi/2$, when the network is in its simplest form, the strong-coupling basis (406) is also the WKB basis in the upper right domain, as illustrated in Figure 35(a). The WKB basis in any other cell of $C\backslash\mathcal{W}^\vartheta$ can be found by a fairly simple wall-crossing argument, similar to the discussion around equation (389). We thus conclude that we have determined the non-perturbative Higgs branch partition function $\mathcal{Z}^\vartheta_\alpha(\mathbf{z},\epsilon)$ across $C_\mathbf{z} \times [0, 2\pi]$!

The non-pertubative partition function can be defined as a Borel sum, in the direction $\vartheta$, of an asymptotic WKB solution in $\epsilon$, at the position $\mathbf{z}$, which is then analytically continued in $\mathbf{z}$ as well as in $\epsilon$. But as we found in this section, the non-perturbative partition function $\mathcal{Z}^\vartheta_\alpha(\mathbf{z},\epsilon)$ can also be computed without calculating any Borel transform, by simply bootstrapping it using the data of the spectral network $\mathcal{W}^\vartheta$. We may think of the resulting object $\mathcal{Z}^\vartheta_\alpha(\mathbf{z},\epsilon)$ as a basis of analytic functions assigned to each cell in the space $C_\mathbf{z} \times [0, 2\pi]$ with respect to the three-dimensional spectral network $\mathcal{W}^{3\text{d}}$, obtained by varying the two-dimensional spectral network $\mathcal{W}^\vartheta$ in the $\vartheta$-direction.[50]

We stress that even though we have attached the name Higgs branch to the non-perturbative partition function $\mathcal{Z}^\vartheta_\alpha$, it is really defined (through wall-crossing) across the whole $\mathbf{z}$-plane. A more appropriate name might therefore have been the $\mathbf{z}$-plane partition function.

Note that in the context of $\mathcal{W}^\vartheta$-abelianization, the basis (406) defines a choice of $\mathcal{W}^\vartheta$-framing at each of the two punctures of $C$. This choice of framing is called the **WKB framing** in [9]. In the conventions of Figure 35(a) the Hankel function $H^2_\nu(x)$, with $\arg\epsilon = 0$, is asymptotically small when approaching the puncture at $z = 0$ along the negative $\mathbf{z}$-axis, whereas the Hankel function $H^1_\nu(x)$, with $\arg\epsilon = 0$, is asymptotically small when approaching the puncture at $\mathbf{z} = \infty$ along the negative $\mathbf{z}$-axis. With this choice of framing data, $\mathcal{W}^\vartheta$-abelianization is equivalent to the exact WKB analysis [8].

Finally, let us remark that a similar analysis can be performed for the $\mathbb{P}^{N-1}$-model for any $N$. If we choose all masses $\widetilde{m}_j = \widetilde{m}$, or at least with the same phase, we find a Fenchel-Nielsen network at phase $\vartheta_{\text{FN}} = \arg(i\widetilde{m})$. The Higgs branch partition functions $\mathcal{Z}_\alpha(\mathbf{z},\epsilon)$ form a basis of exact WKB solutions at this phase in the cell enclosing the puncture at $\mathbf{z} = 0$. The non-perturbative Higgs branch partition function $\mathcal{Z}^\vartheta_\alpha(\mathbf{z},\epsilon)$ in other cells can be found by identifying solutions to the $\mathbb{P}^{N-1}$-differential equation (272) with the correct asymptotics at infinity in the $\mathbf{z}$-plane together with wall-crossing arguments.

---

[50]More about three-dimensional spectral networks can for instance be found in [82].

## 5.9 Remark: categorification

In the story so far we have studied the simplest $\mathcal{N} = (2,2)$ boundary conditions as possible: D-branes in either the A or B-twist labelled by a phase $\zeta$ and a vacuum $\alpha$. Yet, it is well-known by now that boundary conditions in a two-dimensional topological field theory (TFT) form a richer structure, namely that of a $\mathbb{C}$-**linear category**. That is, even TFT's in two dimensions can be extended. Let us explain this here briefly. We refer to, for instance, Kapustin's [83] for a great physics introduction to the concept of an **extended TFT**, as well as to many of the important mathematical references.

The objects in the category $\mathcal{C}$, associated to a 2d TFT, correspond to the different types of boundary conditions. These are for instance the A-branes (B-branes) in the A-twisted (B-twisted) version of a 2d $\mathcal{N} = (2,2)$ theory. The morphisms in the category $\mathcal{C}$ correspond to so-called **boundary-changing local operators**. That is, local operators $\mathcal{O}_{\alpha\beta}$ on the boundary that implement a change from boundary condition $\alpha$ to boundary condition $\beta$.

These objects may be argued to form a (graded) vector space, with identity given by the trivial operator. Indeed, by a change of perspective, where we replace the operator insertion $\mathcal{O}_{\alpha\beta}$ on a vertical (previously time-like) boundary component by a small semi-circle and where we consider the angle $\phi$ as the spatial coordinate and the radius $r$ as time, we may interpret the boundary-changing operator as specifying an initial condition, and the space of initial conditions in any quantum field theory is known to form a vector space [83]. This is illustrated in Figure 36.

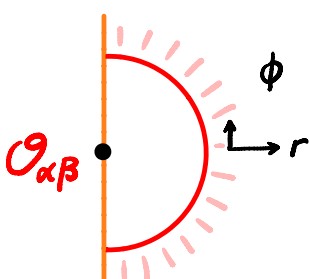

Figure 36: Boundary-changing local operators $\mathcal{O}_{\alpha\beta}$ form a vector space, since they can be interpreted as boundary conditions.

The resulting category $\mathcal{C}$, with composition of morphisms defined by the fusion product of the operators $\mathcal{O}_{\alpha\beta}$, is the category that the **extended 2d TFT** assigns to a point.

This category was constructed explicitly for two-dimensional Landau-Ginzburg models (in the A-twist) by Gaiotto, Moore and Witten [35] and expanded on by Khan and Moore [84, 31].[51] The upshot in their works is that BPS soliton solutions may be

---

[51]See Ahsan's thesis [85] for a great introduction to these works, and the related works [86, 87, 88] for many more advances.

rotated, or **boosted**, on the two-dimensional worldsheet by an angle $\varphi$. That is, the boosted BPS soliton $\phi_{\alpha\beta}^{\zeta'}(\sigma, \tau)$, interpolating between the vacua $\alpha$ and $\beta$, is a solution to the **boosted-soliton equation**

$$\left(\frac{d}{d\sigma} + i\frac{d}{d\tau}\right)\phi_{\alpha\beta}^{\zeta',i} = \frac{ie^{i\varphi}\zeta_{\alpha\beta}}{2}g^{i\bar{j}}\overline{\partial_j W}(\phi_{\alpha\beta}^{\zeta'}). \tag{407}$$

This equation preserves the supercharge $Q_A^{\zeta'}$ with

$$\zeta' = \zeta_{\alpha\beta}e^{i\varphi}. \tag{408}$$

Whereas a stationary soliton has a time-like **world-line** (the region where the solution is not exponentially close to either vacuum, and where its energy is localised), a boosted soliton has their world-line rotated by the angle $\varphi$. See Figure 37.

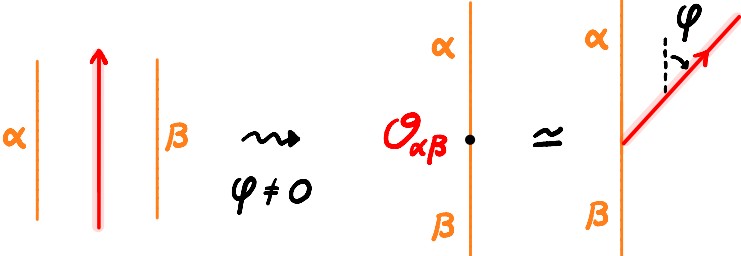

Figure 37: Stationary solitons have a time-like world-line (on the left), whereas boosted solitons have their worldline tilted by an angle $\varphi$ (on the right). Boosted soliton define boundary-changing local operators $\mathcal{O}_{\alpha\beta}$ (in the middle).

Given a time-like boundary component, we may now interpret the boosted soliton $\phi_{\alpha\beta}^{\zeta'}(\sigma, \tau)$, with $\varphi \neq 0$, as a boundary changing operator $\mathcal{O}_{\alpha\beta}^{\zeta'}$ implementing the change from the boundary condition specified by $(\alpha, \zeta')$ to the one specified by $(\beta, \zeta')$.[52] See Figure 37. In other words, the morphisms of the category of boundary conditions of Landau-Ginzburg models are implemented by the boosted solitons.

We may then generalize the cigar partition function $\mathcal{Z}_\alpha^\vartheta$ from §5.2, with boundary condition specified by a single vacuum $\alpha$, to a new **"categorified" cigar partition function** $\mathcal{Z}_{\mathtt{b}}^{\zeta'}$ with boundary condition $\mathtt{b}$ specified by a *collection* of vacua $\{\alpha\}$ and corresponding boundary-changing operators, all preserving the same supercharge $Q_A^{\zeta'}$. Note that, in contrast to the discussion earlier in this section, the new partition function $\mathcal{Z}_{\mathtt{b}}^{\zeta'}$ (at least for Landau-Ginzburg models) needs to the computed in the A-twist with respect to the supercharge $Q_A^{\zeta'}$. It therefore localizes on (and thus "counts") the solutions to the $\zeta'$-instanton equation

$$\left(\frac{d}{d\sigma} + i\frac{d}{d\tau}\right)\phi^i = \frac{ie^{i\varphi}\zeta}{2}g^{i\bar{j}}\overline{\partial_j W}(\phi), \tag{409}$$

---

[52] For $\varphi = 0$ we need to displace the two boundary components, so that the resulting configuration corresponds to the domain wall solution from §5.2.

with boundary conditions specified by the boundary data $\mathfrak{b}$. Examples of $\mathcal{Z}_{\mathfrak{b}}^{\zeta'}$ may be worked out using the web-based formalism of [35, 84, 31].

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
