# Peer review of "Les Houches lectures on non-perturbative Seiberg-Witten geometry"

_SciPost Physics_

## Round 1 · Referee Report · Anonymous (Referee 1) · 2025-10-2

The referee discloses that the following generative AI tools have been used in the preparation of this report:
Microsoft Copilot, powered by OpenAI's GPT-4 architecture, has been used to assist with language formulation of the report.
Strengths
Weaknesses
Report
These lecture notes explore the intricate relationship between BPS states in two-dimensional (2,2) supersymmetric quantum field theories and exact WKB analysis. This connection has gained prominence through the foundational work of Gaiotto, Moore, and Neitzke on wall-crossing and spectral networks.
The scope of the notes is impressively broad. They encompass key aspects of (2,2) QFTs, mirror symmetry, and Witten’s quantum mechanical Morse theory, while also touching on topics such as Cohomological Field Theory, Equivariant Localization, and even Gaiotto-Moore-Witten’s Categorification of 2d Wall-Crossing. This breadth makes the lectures intellectually stimulating and stylistically rich. However, the ambitious range of material occasionally detracts from the pedagogical clarity, which may pose challenges for readers seeking a more structured introduction.
The final section presents original contributions. The authors introduce non-perturbative partition functions for 2d (2,2) QFTs with boundary conditions—specifically, Omega-deformed cigar partition functions—and argue that these are governed by wall-crossing phenomena associated with solitons. The underlying mathematical framework is exact WKB analysis: the partition functions satisfy Schrödinger-type equations with potentials where the equivariant parameter plays the role of Planck’s constant. The discontinuous behavior of these functions is then interpreted via Stokes phenomena in exact WKB theory.
Overall, these lecture notes are carefully written and offer a wealth of material, including numerous references for further study. They serve as a valuable and timely review of the interplay between supersymmetric QFTs and exact WKB methods.
Suggestions for Improvement:
1) References: The bibliography is not comprehensive, particularly with respect to both the foundational literature and the diverse topics covered. Expanding the reference list would enhance the utility of the notes for readers seeking deeper engagement with the material.
2) Discussion of Novel Results: The main results of Section 5 are only briefly mentioned in the introduction, and there is no dedicated concluding discussion. A more detailed summary—either in the introduction or in a final section—would help readers appreciate the significance and novelty of the contributions without requiring a full read-through of the technical details.
Recommendation
Ask for minor revision

---

## Editorial Decision

unknown